# Turning Sand to Gold: Recycling Data to Bridge On-Policy and Off-Policy Learning via Causal Bound

**Tal Fiskus**
Department of Computer Science
Bar-Ilan University
Ramat Gan, Israel
fiskust@biu.ac.il

**Uri Shaham**
Department of Computer Science
Bar-Ilan University
Ramat Gan, Israel
uri.shaham@biu.ac.il

## Abstract

Deep reinforcement learning (DRL) agents excel in solving complex decision-making tasks across various domains. However, they often require a substantial number of training steps and a vast experience replay buffer, leading to significant computational and resource demands. To address these challenges, we introduce a novel theoretical result that leverages the Neyman-Rubin potential outcomes framework into DRL. Unlike most methods that focus on bounding the counterfactual loss, we establish a causal bound on the factual loss, which is analogous to the on-policy loss in DRL. This bound is computed by storing past value network outputs in the experience replay buffer, effectively utilizing data that is usually discarded. Extensive experiments across the Atari 2600 and MuJoCo domains on various agents, such as DQN and SAC, achieve *up to 383%* higher reward ratio, outperforming the same agents without our proposed term, and reducing the experience replay buffer size by *up to 96%*, significantly improving *sample efficiency at a negligible cost*[1].

## 1 Introduction

Deep reinforcement learning agents have demonstrated remarkable success in solving complex decision-making tasks across various areas of interest, such as gaming [1–5], robotics [6–8], healthcare [9–11], and autonomous driving [12–14]. It also has a vital role in enhancing large language models [15, 16]. Despite these achievements, DRL is still challenging due to its high computational and resource demands. For example, AlphaGo Zero was trained for 29 million self-play games [17], and the training experience for OpenAI's Rubik's Cube robotic hand is roughly equated to 13 thousand years [18].

In DRL, we can categorize agents into two types: on-policy agents, like A3C and PPO [19, 20], and off-policy agents, such as DQN and SAC [21, 22]. On-policy agents are generally simpler and often considered first. They interact with the environment, collect experiences using a behavior policy, and once they optimize the target policy, discard all previously collected experiences to ensure that the behavior and target policies remain identical. This consistency leads to more stable convergence but at the cost of sample inefficiency, as each optimization iteration requires collecting new data. Off-policy agents, on the other hand, allow the agent to learn from previously collected experiences, meaning it can have different behavior and target policies. This methodology improves sample efficiency by leveraging past experiences stored in a replay buffer. While this improves sample efficiency, it can also lead to reduced learning stability due to the mismatch between the behavior and target policies. As a result, improving the learning stability typically requires a substantial number of training steps and a vast experience replay buffer, leading to significant computational and resource demands.

---

[1]Our code is available at `https://shaham-lab.github.io/TurningSandToGold/`

39th Conference on Neural Information Processing Systems (NeurIPS 2025).

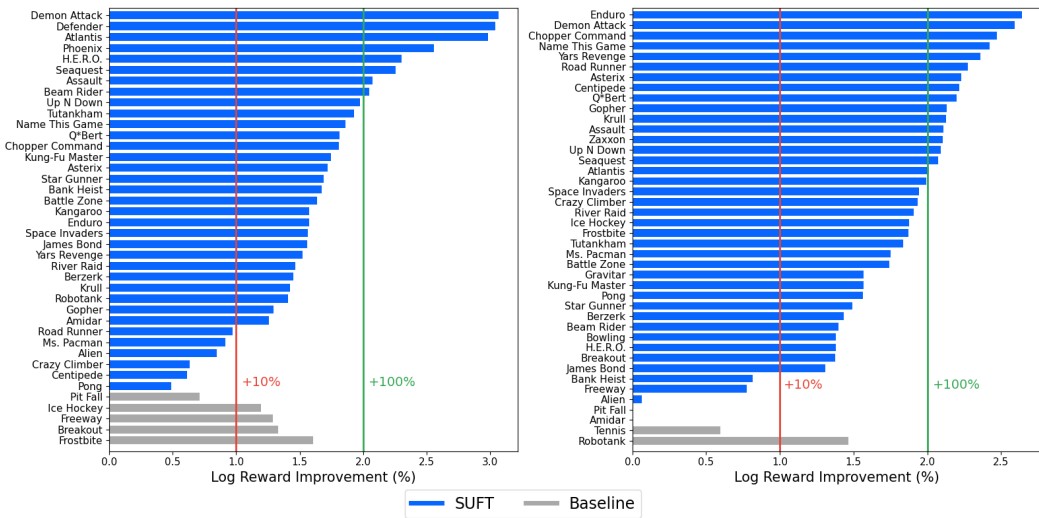

Figure 1: Log-scaled reward improvements comparison between agents using the additional SUFT OPE term and the baseline agents without it across 57 Atari games. The results demonstrate the superior performance of our method across the majority of games. The red line indicates a 10% improvement, and the green line represents a 100% improvement. Left: Double DQN SUFT outperforms the baseline agent in 35 out of the 40 valid games; Right: PPO SUFT outperforms the baseline agent in 39 out of the 42 valid games.

To mitigate these challenges, researchers have proposed numerous strategies designed to reduce computational demands and improve learning stability. A widely adopted approach focuses on enhancing the experience replay buffer sampling, aiming to improve convergence rates by optimizing the selection of experiences [23–26]. While this approach improves convergence, it still involves high computational resources and does not fully address the divergence between on-policy and off-policy methods. Other works have attempted to bridge the divergence by suggesting additional computable terms, e.g., [27]. Although such approaches have improved learning stability, they often require extensive data collection and involve complex techniques.

This work addresses these limitations by introducing a novel theoretical result in causal inference and incorporating it into the DRL framework. Specifically, we establish a causal upper bound on the unobserved factual loss using the observed counterfactual loss and the estimated treatment effect. Within the DRL framework, we incorporate this as an upper bound for the on-policy loss using the standard off-policy loss and an off-policy evaluation (OPE) term, which quantifies the estimated value discrepancy between the behavior policy and the target policy. By incorporating this term into the optimization process, our method explicitly accounts for the causal estimated treatment effect difference between the target policy and the behavior policy. This causal perspective enhances the agent's ability to reason about the differences between the target policy and the behavior policy used to generate the experiences, thereby mitigating the fundamental divergence between on-policy and off-policy learning. Importantly, our method reuses existing data of the behavior policy network values that are usually discarded to improve *sample efficiency at a negligible cost*.

We evaluate our proposed method, SUFT[2], on existing off-policy agents such as DQN and SAC, as well as on-policy agents like PPO, which benefit from our method due to the deviation between behavior and target policies during optimization. Throughout comprehensive experiments on the Atari 2600 and MuJoCo domains, agents using the SUFT OPE term outperform the same ones without it. Notably, we achieve *up to 383%* higher reward ratio, as shown in Figure 2, indicating an improvement in the training convergence rate compared to an identical agent without the additional term. In addition, we reduce the experience replay buffer size by *up to 96%* while outperforming the baseline agent that uses a *25 times larger buffer* by up to *437.05%*. These significant improvements highlight the potential of our method to make DRL more computationally efficient and accessible.

---

[2]We name this novel theoretical result the SUFT causal bound, representing the initials of Sample efficient, Upper bound, Factual loss, and Treatment effect.

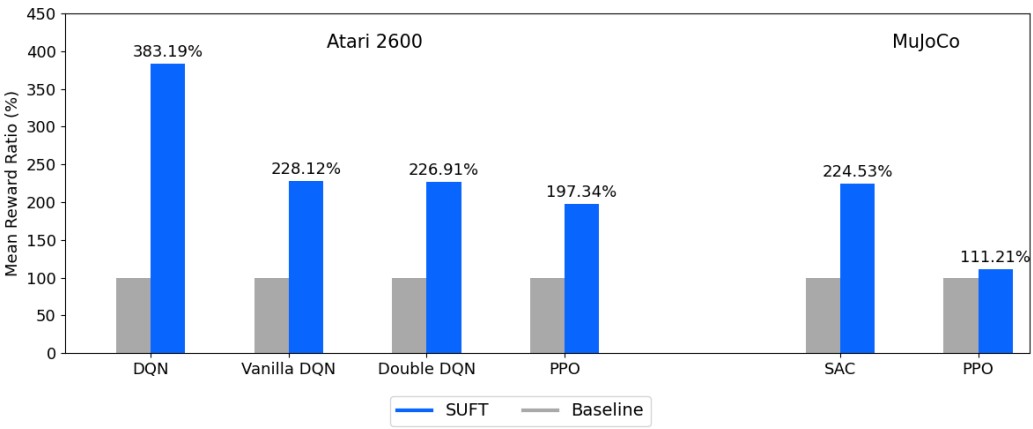

Figure 2: Mean reward ratio comparison between agents using the additional SUFT OPE term and the baseline agents without it across 57 Atari games and five MuJoCo environments, highlighting the profound reward gains across diverse agents and domains.

## 2 Related Work

### 2.1 Causal Reinforcement Learning

Causal reinforcement learning (CRL) is an emerging subfield of reinforcement learning (RL). CRL seeks to enhance agents' performance by incorporating causal inference methodologies into the learning process. The causal perspective allows the agent to learn cause-and-effect knowledge, resulting in more informed and effective decision-making [28, 29].

In causal inference, a treatment typically refers to performing a single action, commonly termed as performing an intervention [30, 31]. On the other hand, the majority of works in CRL define treatment as a sequence of actions determined by a dynamic treatment plan [32–35]. In this work, we adopt the latter approach and consider the treatment to be the agent's policy. This aligns with the RL framework, which focuses on evaluating the long-term effects of policies rather than the immediate impact of individual actions. Recent research in CRL covers a variety of areas, such as advancing generalizability [36], promoting safe agents [37, 38], enhancing sample efficiency [39–41], and addressing spurious correlations [42–44]. Our work falls in the field of CRL and primarily contributes to the latter two directions.

**CRL for Addressing Spurious Correlations:** Spurious correlations in off-policy RL often arise as a result of learning based on experiences from an old behavior policy. These experiences might not accurately reflect the states and actions the target policy is likely to encounter, resulting in deceptive associations between actions and outcomes that limit the agent's ability to make reliable decisions. Several works leverage OPE as a systematic way to evaluate the quality of a policy using off-policy data, which helps mitigate the impact of spurious correlations during policy optimization [45, 46].

**CRL for Addressing Sample Inefficiency:** Causality provides a valuable perspective for creating sample-efficient agents, with research focusing on three key directions: representation learning [47–49], directed exploration [50, 51], and data augmentation [52, 53]. While many works improve sample efficiency by generating new data, our method utilizes existing discarded data from past experiences, achieving *sample efficiency at a negligible cost*. This novel strategy sets our work apart, demonstrating that DRL agents can achieve sample efficiency with marginal computational overhead.

### 2.2 Potential Outcomes Causal Bounds

The Neyman-Rubin potential outcomes framework [54] is a cornerstone of causal inference, providing a robust theoretical basis for estimating treatment effects. Several works have utilized this framework to develop causal bounds [55, 56]. For instance, Shalit et al. [57] establish a generalization-error bound to estimate individual treatment effects from observational data, addressing unobserved

counterfactuals by formalizing a lemma that bounds this loss. Our work expands on this foundational framework and introduces several key changes.

First, while most works apply these bounds within general machine learning (ML) settings [58, 59], we redefine the terms to align with the DRL framework. Second, while other studies consider treatment as a single intervention [60, 61], we address treatments as a sequence of actions within a treatment plan. The primary and novel aspect of our work lies in reversing the traditional loss-bounding methodologies. Whereas existing methods typically bound the counterfactual loss [62, 63], our approach does the exact opposite and bounds the factual loss. This pivotal shift is feasible within the RL framework since off-policy methods use counterfactual data while factual data remains unobserved. Finally, whereas other approaches mainly focus on the scenario of a single target treatment and a single control treatment [64, 65], we scale this scenario to a single target treatment and $N$ control treatments. This multiple control treatments scenario allows us to leverage the replay buffer, which contains experiences generated from $N$ behavior policies, enabling effective evaluation of the target policy against a broader spectrum of previous behavior policies.

## 3   Preliminaries

In this section, we present some fundamental conceptual ideas necessary to understand our method.

**Causal and Counterfactual Reasoning:**   Causal reasoning refers to the process of identifying and understanding the relationships between causes and their effects. Counterfactual reasoning [66], a subfield of causal reasoning, goes a step further by imagining alternative scenarios to answer "what-if" questions, such as "What would have happened if a different treatment had been applied?" as Pearl & Mackenzie [67] elaborate in "The Book of Why". This form of reasoning enables the evaluation of potential outcomes conditioned under different treatments. In RL, incorporating counterfactual reasoning enhances agents' capabilities to evaluate and optimize policies in complex, dynamic environments. By retrospectively analyzing alternative policies and their outcomes, agents can improve decision-making and policy optimization. In this paper, counterfactual reasoning is implemented through the integration of a causal bound into the agent's loss function.

**Off-Policy Evaluation:**   Off-policy evaluation is a fundamental concept in RL [68], which aims to numerically evaluate the performance of a desired policy $\pi$ using historical data $D$ that was collected under different policies. As outlined in [69], the desired policy being optimized is referred to as the *target policy*, while the policies used to generate experiences are termed the *behavior policies*. OPE is particularly important in offline RL, where the agent learns the target policy solely from a static dataset $D$ without interacting with the environment. In contrast, this work focuses on applying OPE to online DRL agents, which continuously interact with the environment while updating the experience replay buffer $D$.

## 4   SUFT Causal Bound

In this section, we formalize our theoretical causal result and detail our method.

### 4.1   Rationale

To illustrate our method and provide a general intuition, we begin with a straightforward example of a greedy off-policy DQN agent. This agent observes the current state of the environment, uses its Q-network to compute Q-values, and selects the action with the highest Q-value for interacting with the environment. As the agent continuously interacts with the environment, it stores the experiences in a replay buffer and periodically updates its Q-network, which determines its target policy.

During Q-network updates, an ideal scenario would be to replace all experiences with fresh samples generated directly from the target policy without any additional cost. This would preserve the sample efficiency from the off-policy approach while resolving the divergence between the behavior and target policies. However, since such a scenario is impractical, we introduce a novel mathematical result and adapt it to the DRL framework to bridge between on-policy and off-policy methods.

Within the causal framework, the on-policy loss corresponds to the factual loss when the treatment being optimized matches the treatment applied for generating the observations. This is analogous to optimizing the target policy using experiences from an identical behavior policy. Similarly, the off-policy loss aligns with the counterfactual loss when the treatment being optimized differs from the one used to generate the observations. This is equivalent to optimizing the target policy using experiences from a different behavior policy.

In this work, we establish a causal upper bound on the unobserved factual loss using the observed counterfactual loss and an additional term, which is the estimated treatment effect of the target treatment compared to $N$ control treatments. After formalizing the upper bound within the causal framework, we seamlessly integrate it into the DRL framework as an upper bound for the on-policy loss using the standard off-policy loss and the estimated treatment effect, which serves as an OPE.

Throughout this paper, we demonstrate the causal definitions alongside corresponding DQN definitions to provide a clear and intuitive representation of the reduction from the causal framework to the DRL framework. Nevertheless, our method is applicable to any DRL agent with a V or Q-value network that influences its policy, such as DQN and Actor-Critic architectures, as long as a divergence exists between the target and behavior policies.

## 4.2 Setup and Definitions

Here, we present the necessary mathematical definitions required to formalize our causal upper bound. Comprehensive definitions and proofs, along with a detailed reduction from the causal framework to the DRL framework, are provided in Appendix B.

To simplify the equations, we present the binary treatment case, where there is a single target treatment, denoted by $t = 1$, and a single control treatment, denoted by $t = 0$. In practical DRL applications, we extend the binary case to a multiple control treatment case involving a single target policy and $N$ behavior policies, reflecting the existence of experiences generated from different behavior policies in the replay buffer. The proofs and definitions for the multiple control treatment case are in Appendix B.

We employ the Neyman-Rubin potential outcomes framework and define the following: $\mathcal{X} \subset \mathbb{R}^d$ is the observation space. $\mathcal{Y} \subset \mathbb{R}$ is the outcome space. $\mathcal{T} = \{0, 1\}$ is the binary treatment space. Under this framework, there are two potential outcomes depending on the treatment assignment: if $t = 1$, the observed outcome is $y = Y_1$, and if $t = 0$, the observed outcome is $y = Y_0$. We assume there exists a joint distribution $P(X, T, Y_0, Y_1)$ with the following components: $q_t := \Pr(T = t)$ is the treatment probability. $P_X^t := P(X|T = t)$ is the conditional distribution of observations. $P_{Y_1}^x := P(Y_1|X = x)$ is the conditional distribution of potential outcomes given the target treatment, and $P_{Y_0}^x := P(Y_0|X = x)$ is the conditional distribution of potential outcomes given the control treatment. We also assume that there are no hidden confounders in the observational data, which makes the estimated treatment effect identifiable. This is formalized by assuming that the standard strong ignorability condition holds: $(Y_0, Y_1) \perp\!\!\!\perp T|X$, and $0 < p(t = 1|x) < 1$ for all $x$. Strong ignorability is a sufficient condition for the estimated treatment effect to be identifiable [70, 57]. To integrate the potential outcomes framework into the ML context, we define a neural network $\phi(x; \theta_t) : \mathcal{X} \to \mathcal{Y}$, parameterized by weights $\theta_t$ corresponding to the performed treatment $t \in \mathcal{T}$.

We now present the analogous definitions within the DRL framework for a DQN agent, assuming the DRL framework is defined as a Markov decision process (MDP).

$\mathcal{X} := (\mathcal{S} \times \mathcal{A})$ is the state-action space. Potential outcome $y \sim P_{Y_t}^{(s,a)}$ is the accumulated reward over $n$ time steps, starting from state $s$, taking the initial action $a$, and following policy $\pi_t$. The hypothesis $\phi(x; \theta_t) := Q(s, a; \theta_t) : (\mathcal{S} \times \mathcal{A}) \to \mathcal{Y}$ is the Q-values network, which influences the policy $\pi_t$.

Let $\mathbf{L} : \mathbb{R} \times \mathbb{R} \to \mathbb{R}_{\geq 0}$ represent a loss function. We rely on the following inequality assumption, which is a variation of the inverse triangle inequality:

**Assumption 4.1.**

$$\mathbf{L}(x, y) - \mathbf{L}(x', y') \leq \mathbf{L}(x, x') + \mathbf{L}(y, y').$$

The proof for the $L_1$ loss is in Appendix B[3].

Now, we define the expected factual and counterfactual losses, which are used to formulate the bound.

**Definition 4.2.** The expected outcome loss:

$$\ell_\phi(x, t) := \mathbb{E}_{y \sim P_{Y_t}^x}[\mathbf{L}(y, \phi(x; \theta_t))].$$

The corresponding definition in DQN:

**Definition 4.3.** The expected temporal difference loss:

$$\ell_Q(s, a, t) := \mathbb{E}_{y \sim P_{Y_t}^{(s,a)}}[\mathbf{L}(y, Q(s, a; \theta_t))].$$

Note that in the DRL framework, the expected outcome loss refers to the temporal difference loss between the accumulated rewards and the Q-value estimation.

**Definition 4.4.** The expected factual outcome loss:

$$\epsilon_{F_\phi}^1 := \mathbb{E}_{x \sim P_X^1}[\ell_\phi(x, 1)].$$
$$\epsilon_{F_\phi}^0 := \mathbb{E}_{x \sim P_X^0}[\ell_\phi(x, 0)].$$
$$\epsilon_{F_\phi} := q_1 \epsilon_{F_\phi}^1 + q_0 \epsilon_{F_\phi}^0.$$

To formalize the corresponding definition in the DRL framework, consider a replay buffer $D_1$ containing samples produced by a behavior policy that matches the target policy. In this context, the on-policy loss is a special case of the expected factual loss, and for DQN, it is defined as:

**Definition 4.5.** The expected on-policy loss:

$$\epsilon_{\text{On-Policy}_Q} := \mathbb{E}_{(s,a) \sim D_1}[\ell_Q(s, a, t_{\text{target}})].$$

Note that this loss is the temporal difference loss with experiences collected on-policy, with identical target and behavior policies from $D_1$.

**Definition 4.6.** The expected counterfactual outcome loss:

$$\epsilon_{CF_\phi}^1 := \mathbb{E}_{x \sim P_X^0}[\ell_\phi(x, 1)].$$
$$\epsilon_{CF_\phi}^0 := \mathbb{E}_{x \sim P_X^1}[\ell_\phi(x, 0)].$$
$$\epsilon_{CF_\phi} := q_0 \epsilon_{CF_\phi}^1 + q_1 \epsilon_{CF_\phi}^0.$$

To establish the corresponding definition within the DRL framework, consider a replay buffer $D_0$ that stores experiences generated by a behavior policy that differs from the target policy. In this context, the off-policy loss is a special case of the expected counterfactual loss, and for DQN, it is defined as:

**Definition 4.7.** The expected off-policy loss:

$$\epsilon_{\text{Off-Policy}_Q} := \mathbb{E}_{(s,a) \sim D_0}[\ell_Q(s, a, t_{\text{target}})].$$

Note that this loss is the temporal difference loss with experiences gathered off-policy, with different target and behavior policies from $D_0$.

Additionally, in DRL applications, we consider the scenario of a replay buffer containing experiences generated from $N$ distinct behavior policies. The definitions for this case are in Appendix B.

We now define our method's additional estimated treatment effect term in both the causal and the DRL frameworks:

**Definition 4.8.** The estimated treatment effect loss:

$$\psi_\phi^1 := \mathbb{E}_{x \sim P_X^0}[\mathbf{L}(\phi(x; \theta_0), \phi(x; \theta_1))].$$
$$\psi_\phi^0 := \mathbb{E}_{x \sim P_X^1}[\mathbf{L}(\phi(x; \theta_1), \phi(x; \theta_0))].$$
$$\psi_\phi := q_0 \psi_\phi^1 + q_1 \psi_\phi^0.$$

---

[3]DRL agents are often trained using $L_2$ loss, for which Assumption 4.1 does not hold. While the assumption is needed to prove Theorem 4.10, similar experimental results were obtained with both $L_1$ and $L_2$ losses. To provide intuition for the observed $L_2$ performance, we empirically examined the validity of Assumption 4.1 using $L_2$. This analysis is detailed in Appendix D.

The interpretation for the estimated treatment effect in DRL is an OPE, and for DQN, it is defined as:

**Definition 4.9.** The SUFT OPE term:

$$\psi_{\text{SUFT}_Q} := \mathbb{E}_{(s,a) \sim D_0}[\mathbf{L}\left(Q(s, a; \theta_{\text{behavior}}), Q(s, a; \theta_{\text{target}})\right)].$$

This term aligns with the principles of OPE, focusing on quantifying the discrepancy between the target policy and the behavior policy using off-policy data. Once the data is stored in the buffer, it is used as observational data.

### 4.3 Causal Bound Theorem

Having established the key definitions, we now present our novel factual loss upper bound theorem:

**Theorem 4.10.** *The expected factual outcome loss, $\epsilon_{F_\phi}$, is bounded by the expected counterfactual outcome loss, $\epsilon_{CF_\phi}$, the estimated treatment effect loss, $\psi_\phi$, and a constant term $\delta$, independent of $\phi$:*

$$\epsilon_{F_\phi} \leq \epsilon_{CF_\phi} + \psi_\phi + \delta.$$

The proof appears in Appendix B.

The interpretation of Theorem 4.10 in the DRL framework establishes an upper bound to the on-policy loss, by the off-policy loss, the SUFT OPE term, and a constant term $\delta$. A certain difference exists in the observational data between the causal and the DRL framework, which is detailed in Appendix B.

$$\epsilon_{\text{On-Policy}_Q} \leq \epsilon_{\text{Off-Policy}_Q} + \psi_{\text{SUFT}_Q} + \delta.$$

Note that since $\delta$ is independent of $Q$, it can be omitted during the gradient-based optimization of $Q$.

To the best of our knowledge, we are the first to introduce such an upper bound within the causal framework on the factual loss, adapting it to the DRL framework, thereby bounding the on-policy loss. Importantly, based on our causal bound theorem, the adaptation to the DRL framework is not constrained to DQN agents but can be applied to every off-policy agent with V or Q-value networks that influence its policy. This scenario is thoroughly detailed in Appendix B. Furthermore, our method is also applicable to on-policy agents, such as PPO, which benefits from it due to the divergence between the behavior and target policies that arises during epoch optimization.

### 4.4 Recycling Value Network Outputs

To compute the SUFT OPE term in the causal bound, we need to calculate the Q-network values corresponding to both the target and the behavior policies. While the Q-values for the target policy can be easily computed using the current Q-network, the Q-network values for the old behavior policies are not addressable since storing the network weights for every behavior policy is impractical. Notably, during the agent's training process, the Q-values matching the behavior policies are already calculated when they are used to select the action to perform. These values are calculated for the purpose of exploitation and then discarded once the action is executed. The novel contribution of our approach lies in utilizing and reusing the value network outputs in order to compute the SUFT OPE term. This allows the reuse of data that is overlooked while enhancing the loss optimization with significant causal insights in the form of the estimated treatment effect. Our approach effectively transforms overlooked data into valuable causal insights, metaphorically *turning sand data into gold*.

To obtain value network outputs in practice, we store them in the experience replay buffer. For agents employing a Q-network, only a single Q-value corresponding to the behavior policy is stored, resulting in an experience tuple of the form $(s, a, r, s', Q(s, a; \theta_{\text{behavior}}))$, instead of the standard $(s, a, r, s')$. For agents using a V-network, the state value output is stored instead, yielding an experience tuple of the form $(s, a, r, s', V(s; \theta_{\text{behavior}}))$, instead of the standard $(s, a, r, s')$.

### 4.5 Universal Implementation of the SUFT OPE Term

Following the formalization of the SUFT causal bound, in Section 4.3, we adopt the well-established approach of optimizing a bound on the loss, a technique commonly employed in algorithms like expectation–maximization [71] and variational inference [72]. Consequently, rather than directly optimizing the on-policy loss, we focus on optimizing the SUFT causal bound. In addition, to improve

the flexibility and accuracy of the loss optimization, we incorporate a $\lambda_{TF}$ coefficient into the SUFT OPE additional term. In the case of DQN, the SUFT causal bound objective term is as follows:

$$\epsilon_{\text{Off-Policy}_Q} + \lambda_{\text{TF}} \cdot \psi_{\text{SUFT}_Q}.$$

In value-based methods such as DQN agents, the SUFT OPE term is seamlessly added to the standard loss function, while in Actor-Critic architectures, it is incorporated into the critic's loss. We provide a detailed pseudocode of our method algorithms for both DQN and PPO in Appendix C.

## 5 Experiments

In this section, we provide a comprehensive empirical analysis of our proposed method, emphasizing the impact of adding the SUFT OPE term to the agent's standard loss function across a broad and diverse range of environments. In this section, we present results obtained using $L_2$ loss, allowing us to compare our method to standard DRL agents. The results for $L_1$ loss are detailed in Appendix D.

### 5.1 Experimental Setup

We assess our method's performance across two distinct domains: The first consists of 57 Atari games with a discrete action space, which contain a range of decision-making scenarios, including immediate versus long-term strategies, varying image background complexities, and dynamic on-screen objects. The second includes five MuJoCo environments that simulate robotic motions with a continuous action space that provides a robust platform for dynamic physical interactions.

In the Atari domain, we perform a wide range of experiments using DQN [21], which serves as the foundational algorithm for numerous Q-value network agents, Vanilla DQN, and Double DQN [73], as well as PPO [20], which is a widely adopted agent that has become a standard in the field. The Vanilla DQN results are detailed in Appendix D. For the MuJoCo domain, we evaluate our method using SAC [22], a continuous control off-policy actor-critic agent known for its robust performance, and with PPO. Across these settings, we explore different values of $\lambda_{TF}$, resulting in more than 10K individual experiments. This extensive and versatile experimentation ensures statistically significant findings, providing confidence in the robustness of the results and enabling us to draw more reliable conclusions about our method's performance. More details about tuning $\lambda_{TF}$ are in Appendix D.

**Training Details:** To evaluate SUFT's sample efficiency, we conduct experiments with constrained environment interactions. For the off-policy agents, we use a relatively small experience replay buffer size of 4K. Each agent configuration is independently trained 10 times per environment, with the training process limited to 400K steps in the Atari domain and 1M steps in the MuJoCo domain. Performances are analyzed by taking the median smoothed training reward from the 10-trial set, ensuring a more reliable evaluation while mitigating the influence of randomness on the results. This constrained resources experimentation is designed to demonstrate our method's sample efficiency, and it is a common technique in DRL. [74] Results for a larger buffer size are in Appendix D.

**Implementation Details:** We use the Gymnasium environments API [75] to train on the Atari [76] (version v5) and MuJoCo [77] (version v4) domains. For training the agents, we are using the default architecture and hyperparameter configurations as implemented in the "Stable-Baselines3" DRL library [78]. In the Atari games, we apply standard preprocessing, which includes resizing observations to 84x84 grayscale images and stacking four consecutive frames. This preprocessing is implemented using the library's default Atari wrapper.

### 5.2 Controlled Comparison with Baseline Agents

We evaluate the impact of incorporating the SUFT OPE term by comparing agents that include our proposed term to baseline agents without it, ensuring both types share identical architecture and hyperparameters, where the presence of our proposed term is the only difference between them. This controlled experiment is designed to isolate the influence of our method, ensuring fair comparisons.

In addition, we conduct a $t$-test for each environment against a two-sided alternative using 10 different random seed runs, where the null hypothesis is the baseline result. The $p$-values provide robust statistical significance results for evaluating the effectiveness of our proposed method.

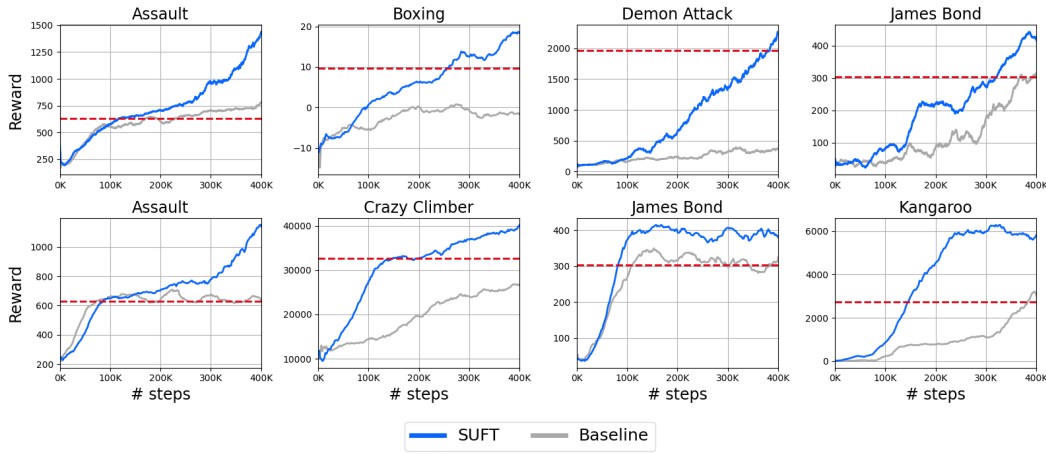

Figure 3: Learning curves comparison between agents using the SUFT OPE term and the baseline agents without it across selected Atari games. The red line indicates human-level performance, showing that SUFT not only surpasses the baseline agent but can even exceed human rewards. Top: Double DQN; Bottom: PPO.

The complete experimental results, the evaluation formulas, and more empirical analysis of our method are detailed in Appendix D.

**Atari:** On the Atari benchmark, SUFT demonstrates significant improvements for all of the agents. On the DQN, Double DQN, and PPO agents, SUFT surpasses the baseline agent in 52, 48, and 50 games out of the 57 games, respectively, with statistically significant gains ($p$-value $< 0.05$) in 49, 37, and 40 games.

We also analyze the results for the valid Atari games (according to the evaluation formula in Appendix D). For the DQN agent, SUFT outperforms the baseline agent in 28 out of the 29 valid games, achieving a reward improvement of over 100% in 48.3% of them and over 10% in 93.1% of them, as shown in Figure 11. For the Double DQN agent, SUFT outperforms the baseline agent in 35 out of the 40 valid games, achieving a reward improvement of over 100% in 20% of them and over 10% in 72.5% of them, as shown in Figure 1 (Left). For the PPO agent, SUFT surpasses the baseline agent in 39 out of the 42 valid games, achieving a reward improvement of over 100% in 38.1% of them and over 10% in 83.3% of them, as shown in Figure 1 (Right).

Additionally, when evaluating the mean reward ratio, SUFT demonstrates profound gains relative to the baseline agents. For the DQN, Vanilla DQN, Double DQN, and PPO agents, SUFT achieves a mean reward ratio of 383.19%, 228.12%, 226.91%, and 197.34%, respectively, as shown in Figure 2.

Moreover, in several Atari games, SUFT not only surpasses the baseline agent but also achieves higher rewards than an average human, as shown in Figure 3. This achievement highlights the potential of our method to boost agents' performance even above the human level, while keeping constrained environment interactions and resource demands. The human rewards are sourced from [79].

**MuJoCo:** For the MuJoCo benchmark, SUFT demonstrates improvements in the SAC and PPO agents. For the SAC agent, SUFT surpasses the baseline agent in 4 out of 5 environments, with statistically significant gains in 3 environments. For the PPO agent, SUFT surpasses the baseline agent in 4 out of 5 environments (in the other environment, the score is the same), with statistically significant gains in 2 environments. With regard to the mean reward ratio, SUFT achieves 224.53% for the SAC agent and 111.21% for the PPO agent, as shown in Figure 2.

### 5.3 Buffer Size Reduction

We also evaluate the sample efficiency of SUFT by comparing its performance to an identical DQN agent utilizing a larger 100K experience replay buffer size. Remarkably, SUFT achieves a mean reward ratio gain of 437.05%, outperforming an identical agent that uses a *25 times larger* buffer size.

Table 1: An ablation study on the DQN agent comparing the SUFT additional term impact against enlarging the buffer size, adding a Vanilla DQN, and adding a Double DQN. The DQN, Vanilla DQN, Double DQN, and SUFT DQN use a 4K buffer size. All agents are using $L_2$ loss.

| Agent | Human Normalized Mean (%) |
|---|---|
| DQN | 25.04% |
| DQN, buffer 100K | 29.29% |
| Vanilla DQN | 41.26% |
| Double DQN | 42.31% |
| **SUFT DQN** | **63.45%** |

These results underscore the potential of our approach to obtain high-performance DRL agents with significantly smaller buffers, paving the way for more computationally efficient and accessible DRL methods by enhancing *sample efficiency at a negligible cost*.

### 5.4 Ablation Study

We conduct an ablation study to isolate the contribution of the SUFT OPE term to DQN performance. As shown Table 1, we compare SUFT DQN against several alternatives: increasing the buffer size, a Vanilla DQN, and a Double DQN. While enlarging the buffer or adopting more complex DQN variants leads to improvements, incorporating the SUFT OPE term yields the highest Human Normalized Mean of **63.45%**, outperforming all other baselines that require higher resources and computational costs. These results highlight the potential of our method to improve the agent's performance by enhancing *sample efficiency at a negligible cost*, providing a simple yet highly effective method for improving agent performance at a *negligible cost*.

### 5.5 Causal Bound Optimization vs Inertia Regularizer

To isolate the effect of our causal mechanism and ensure that the significant improvements of our method stem from the causal bound optimization rather than merely from a regularization effect, we conduct a controlled experiment. Specifically, we evaluate a double DQN agent that adds an inertia regularizer to the standard temporal difference loss function in the form of:

$$\left(Q_{\text{current network}} - Q_{\text{target network}}\right)^2$$

Where $Q_{\text{target network}}$ denotes the slowly updated target network, as done in a standard double DQN. The experiment uses the same configurations as both the SUFT double DQN and the baseline double DQN, with the additional regularization term being the only difference from the baseline agent. The results in Table 11 show that the inertia regularization significantly degrades double DQN performance. Resulting in an 83.21% mean reward ratio. This provides evidence that the gains from our method, which result in a 226.91% mean reward ratio improvement, arise from the causal bound optimization, where the old Q-network values used for the OPE term correspond to the behavior policy that generated the experiences, thereby mitigating the divergence between the on-policy and off-policy methods. These results indicate that our method functions not merely as a stability-promoting regularizer, but as a principled causal bound optimization.

## 6 Conclusion

This paper presents SUFT, a causal upper-bound loss optimization method that enhances DRL sample efficiency and reduces computational demands at a *negligible cost*. We begin by establishing a provable bound on the factual loss within the Neyman-Rubin potential outcomes framework and seamlessly adapting it to the DRL framework, showing that on-policy loss is bounded by every agent's standard off-policy loss and an additional SUFT OPE term. This term is computable by storing past value network outputs in the experience replay buffer, thereby reusing discarded data to improve agents' performance. Experimental results across diverse environments and agents show that our method leads to significant improvements in convergence rates and reward outcomes. Looking ahead to future work, we see physical robotics as a particularly promising domain, where SUFT's reduced-resource method could make DRL more practical under real-world constraints.

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

# A  Causal Bound Notation Summary

$\mathbf{L}(\cdot, \cdot) : \mathbb{R} \times \mathbb{R} \to \mathbb{R}_{\geq 0}$ : loss function.
$q_t$ : the probability of treatment $t$.
$P_X^t$ : the conditional distribution of observations.
$P_{Y_1}^x$ : the conditional distribution of potential outcomes given the target treatment.
$P_{Y_0}^x$ : the conditional distribution of potential outcomes given the control treatment.
$\phi(x; \theta_t) : \mathcal{X} \to \mathcal{Y}$ : neural network hypothesis parameterized by weights $\theta_t$.
$\ell_\phi(x, t)$ : the expected outcome loss.
$\epsilon_{F_\phi}$ : the expected factual outcome loss.
$\epsilon_{CF_\phi}$ : the expected counterfactual outcome loss.
$\psi_\phi$ : The estimated treatment effect loss.
$\delta$ : a constant term independent of $\phi$.

# B  Theoretical Result

## B.1  Definitions and Proofs

In this section, we present the complete definitions and proofs to establish our theoretical result within the causal framework. Let $\mathbf{L} : \mathbb{R} \times \mathbb{R} \to \mathbb{R}_{\geq 0}$ represent a loss function.

**Assumption B.1.**

$$\mathbf{L}(x, y) - \mathbf{L}(x', y') \leq \mathbf{L}(x, x') + \mathbf{L}(y, y').$$

**Proposition B.2.** *Assumption B.1 holds for $L_1$ loss.*

*Proof.*

$$\mathbf{L}(x, y) - \mathbf{L}(x', y') \leq |\mathbf{L}(x, y) - \mathbf{L}(x', y')| \tag{1}$$
$$= ||x - y| - |x' - y'|| \tag{2}$$
$$\leq |x - y - (x' - y')| \tag{3}$$
$$= |x - x' + y' - y)|$$
$$\leq |x - x'| + |y' - y| \tag{4}$$
$$= \mathbf{L}(x, x') + \mathbf{L}(y, y') \tag{5}$$

Inequality 1 is by $x \leq |x|$, equality 2 is by the definition of $L_1$ loss, inequality 3 is by reverse triangle inequality $||x_1| - |x_2|| \leq |x_1 - x_2|$, inequality 4 is by triangle inequality $|x_1 + x_2| \leq |x_1| + |x_2|$, equality 5 is by absolute symmetry $|a| = |-a|$, and the definition of $L_1$ loss. $\square$

We start with the binary treatment case and employ the Neyman-Rubin potential outcomes framework to define the following: $\mathcal{X} \subset \mathbb{R}^d$ is the observation space. $\mathcal{Y} \subset \mathbb{R}$ is the outcome space. $\mathcal{T} = \{0, 1\}$ is the binary treatment space. Under this framework, there are two potential outcomes depending on the treatment assignment: if $t = 1$, the observed outcome is $y = Y_1$, and if $t = 0$, the observed outcome is $y = Y_0$.

We assume a joint distribution $P(X, T, Y_0, Y_1)$ with the following components: $q_t := \Pr(T = t)$ is the treatment probability. $P_X^t := P(X|T = t)$ is the conditional distribution of observations. $P_{Y_1}^x := P(Y_1|X = x)$ is the conditional distribution of potential outcomes given the target treatment, and $P_{Y_0}^x := P(Y_0|X = x)$ is the conditional distribution of potential outcomes given the control treatment. We also assume that there are no hidden confounders in the observational data, which makes the estimated treatment effect identifiable. This is formalized by assuming that the standard strong ignorability condition holds: $(Y_0, Y_1) \perp\!\!\!\perp T|X$, and $0 < p(t = 1|x) < 1$ for all $x$. Strong ignorability is a sufficient condition for the estimated treatment effect to be identifiable [70, 57].

We now present the definitions used to formulate the causal upper bound.

Let $t \in \mathcal{T}$ be a treatment.

**Definition B.3.** $\phi(x; \theta_t) : \mathcal{X} \to \mathcal{Y}$ is a neural network hypothesis parameterized by weights $\theta_t$ corresponding to the performed treatment $t$.

**Definition B.4.** The expected outcome loss:

$$\ell_\phi(x, t) := \mathbb{E}_{y \sim P_{Y_t}^x}[\mathbf{L}(y, \phi(x; \theta_t))].$$

**Definition B.5.** The expected factual outcome loss:

$$\epsilon_{F_\phi}^1 := \mathbb{E}_{x \sim P_X^1}[\ell_\phi(x, 1)].$$
$$\epsilon_{F_\phi}^0 := \mathbb{E}_{x \sim P_X^0}[\ell_\phi(x, 0)].$$

**Definition B.6.** The expected combined factual outcome loss:

$$\epsilon_{F_\phi} := q_1 \epsilon_{F_\phi}^1 + q_0 \epsilon_{F_\phi}^0.$$

**Definition B.7.** The expected counterfactual outcome loss:

$$\epsilon_{CF_\phi}^1 := \mathbb{E}_{x \sim P_X^0}[\ell_\phi(x, 1)].$$
$$\epsilon_{CF_\phi}^0 := \mathbb{E}_{x \sim P_X^1}[\ell_\phi(x, 0)].$$

**Definition B.8.** The expected combined counterfactual outcome loss:

$$\epsilon_{CF_\phi} := q_0 \epsilon_{CF_\phi}^1 + q_1 \epsilon_{CF_\phi}^0.$$

**Definition B.9.** The estimated treatment effect loss:

$$\psi_\phi^1 := \mathbb{E}_{x \sim P_X^0}[\mathbf{L}(\phi(x; \theta_0), \phi(x; \theta_1))].$$
$$\psi_\phi^0 := \mathbb{E}_{x \sim P_X^1}[\mathbf{L}(\phi(x; \theta_1), \phi(x; \theta_0))].$$

**Definition B.10.** The combined estimated treatment effect loss:

$$\psi_\phi := q_0 \psi_\phi^1 + q_1 \psi_\phi^0.$$

## B.2 Causal Bound Theorem

**Lemma B.11.**

$$\mathbb{E}_{y_1 \sim P_{Y_1}^x}[\mathbb{E}_{y_0 \sim P_{Y_0}^x}[\mathbf{L}(y_1, \phi(x; \theta_1))]] = \mathbb{E}_{y_1 \sim P_{Y_1}^x}[\mathbf{L}(y_1, \phi(x; \theta_1))].$$

*Similarly, for the opposite case:*

$$\mathbb{E}_{y_0 \sim P_{Y_0}^x}[\mathbb{E}_{y_1 \sim P_{Y_1}^x}[\mathbf{L}(y_0, \phi(x; \theta_0))]] = \mathbb{E}_{y_0 \sim P_{Y_0}^x}[\mathbf{L}(y_0, \phi(x; \theta_0)))].$$

*Proof.*

$$\mathbb{E}_{y_1 \sim P_{Y_1}^x}[\mathbb{E}_{y_0 \sim P_{Y_0}^x}[\mathbf{L}(y_1, \phi(x; \theta_1))]] = \int_{\mathcal{Y}} \left( \int_{\mathcal{Y}} \mathbf{L}(y_1, \phi(x; \theta_1)) \cdot p(y_0|x) dy_0 \right) \cdot p(y_1|x) dy_1$$

$$= \int_{\mathcal{Y}} \mathbf{L}(y_1, \phi(x; \theta_1)) \cdot \left( \int_{\mathcal{Y}} p(y_0|x) dy_0 \right) \cdot p(y_1|x) dy_1$$

$$= \int_{\mathcal{Y}} \mathbf{L}(y_1, \phi(x; \theta_1)) \cdot p(y_1|x) dy_1$$

$$= \mathbb{E}_{y_1 \sim P_{Y_1}^x}[\mathbf{L}(y_1, \phi(x; \theta_1))]$$

The proof for the opposite case is the same, replacing $y_0$ and $y_1$. $\qquad\square$

**Theorem B.12.** *The expected factual outcome loss, $\epsilon_{F_\phi}$, is bounded by the expected counterfactual outcome loss, $\epsilon_{CF_\phi}$, the estimated treatment effect loss, $\psi_\phi$, and a constant term $\delta$, independent of $\phi$:*

$$\epsilon_{F_\phi} \leq \epsilon_{CF_\phi} + \psi_\phi + \delta.$$

*Where $\delta$ is the following:*

$$\delta^1 := \mathbb{E}_{x \sim P_X^0}[\mathbb{E}_{y_0 \sim P_{Y_0}^x}[\mathbb{E}_{y_1 \sim P_{Y_1}^x}[\mathbf{L}(y_0, y_1)]]].$$
$$\delta^0 := \mathbb{E}_{x \sim P_X^1}[\mathbb{E}_{y_1 \sim P_{Y_1}^x}[\mathbb{E}_{y_0 \sim P_{Y_0}^x}[\mathbf{L}(y_1, y_0)]]].$$
$$\delta := q_0 \delta^1 + q_1 \delta^0.$$

*Proof.* We begin by placing the loss definitions:

$$\epsilon_{F_\phi} - \epsilon_{CF_\phi} = q_1 \epsilon_{F_\phi}^1 + q_0 \epsilon_{F_\phi}^0 - q_0 \epsilon_{CF_\phi}^1 - q_1 \epsilon_{CF_\phi}^0 \tag{6}$$

$$= q_1 \mathbb{E}_{x \sim P_X^1}[\ell_\phi(x,1)] + q_0 \mathbb{E}_{x \sim P_X^0}[\ell_\phi(x,0)] \tag{7}$$
$$- q_0 \mathbb{E}_{x \sim P_X^0}[\ell_\phi(x,1)] - q_1 \mathbb{E}_{x \sim P_X^1}[\ell_\phi(x,0)]$$

$$= q_1 \mathbb{E}_{x \sim P_X^1}[\mathbb{E}_{y_1 \sim P_{Y_1}^x}[\mathbf{L}(y_1, \phi(x;\theta_1))]] + q_0 \mathbb{E}_{x \sim P_X^0}[\mathbb{E}_{y_0 \sim P_{Y_0}^x}[\mathbf{L}(y_0, \phi(x;\theta_0))]] \tag{8}$$
$$- q_0 \mathbb{E}_{x \sim P_X^0}[\mathbb{E}_{y_1 \sim P_{Y_1}^x}[\mathbf{L}(y_1, \phi(x;\theta_1))]] - q_1 \mathbb{E}_{x \sim P_X^1}[\mathbb{E}_{y_0 \sim P_{Y_0}^x}[\mathbf{L}(y_0, \phi(x;\theta_0))]]$$

$$= q_1 (\mathbb{E}_{x \sim P_X^1}[\mathbb{E}_{y_1 \sim P_{Y_1}^x}[\mathbf{L}(y_1, \phi(x;\theta_1))]] - \mathbb{E}_{x \sim P_X^1}[\mathbb{E}_{y_0 \sim P_{Y_0}^x}[\mathbf{L}(y_0, \phi(x;\theta_0))]])$$
$$+ q_0 (\mathbb{E}_{x \sim P_X^0}[\mathbb{E}_{y_0 \sim P_{Y_0}^x}[\mathbf{L}(y_0, \phi(x;\theta_0))]] - \mathbb{E}_{x \sim P_X^0}[\mathbb{E}_{y_1 \sim P_{Y_1}^x}[\mathbf{L}(y_1, \phi(x;\theta_1))]])$$

Now we will use the linearity of expectation and Lemma B.11:

$$\epsilon_{F_\phi} - \epsilon_{CF_\phi} = q_1 (\mathbb{E}_{x \sim P_X^1}[\mathbb{E}_{y_1 \sim P_{Y_1}^x}[\mathbf{L}(y_1, \phi(x;\theta_1))] - \mathbb{E}_{y_0 \sim P_{Y_0}^x}[\mathbf{L}(y_0, \phi(x;\theta_0))]]) \tag{9}$$
$$+ q_0 (\mathbb{E}_{x \sim P_X^0}[\mathbb{E}_{y_0 \sim P_{Y_0}^x}[\mathbf{L}(y_0, \phi(x;\theta_0))] - \mathbb{E}_{y_1 \sim P_{Y_1}^x}[\mathbf{L}(y_1, \phi(x;\theta_1))]])$$

$$= q_1 (\mathbb{E}_{x \sim P_X^1}[\mathbb{E}_{y_1 \sim P_{Y_1}^x}[\mathbb{E}_{y_0 \sim P_{Y_0}^x}[\mathbf{L}(y_1, \phi(x;\theta_1)) - \mathbf{L}(y_0, \phi(x;\theta_0))]]]) \tag{10}$$
$$+ q_0 (\mathbb{E}_{x \sim P_X^0}[\mathbb{E}_{y_0 \sim P_{Y_0}^x}[\mathbb{E}_{y_1 \sim P_{Y_1}^x}[\mathbf{L}(y_0, \phi(x;\theta_0)) - \mathbf{L}(y_1, \phi(x;\theta_1))]]])$$

According to Assumption B.1 we will get:

$$\epsilon_{F_\phi} - \epsilon_{CF_\phi} \le q_1 (\mathbb{E}_{x \sim P_X^1}[\mathbb{E}_{y_1 \sim P_{Y_1}^x}[\mathbb{E}_{y_0 \sim P_{Y_0}^x}[\mathbf{L}(y_1, y_0) + \mathbf{L}(\phi(x;\theta_1), \phi(x;\theta_0))]]]) \tag{11}$$
$$+ q_0 (\mathbb{E}_{x \sim P_X^0}[\mathbb{E}_{y_0 \sim P_{Y_0}^x}[\mathbb{E}_{y_1 \sim P_{Y_1}^x}[\mathbf{L}(y_0, y_1) + \mathbf{L}(\phi(x;\theta_0), \phi(x;\theta_1))]]])$$

$$= q_1 (\mathbb{E}_{x \sim P_X^1}[\mathbb{E}_{y_1 \sim P_{Y_1}^x}[\mathbb{E}_{y_0 \sim P_{Y_0}^x}[\mathbf{L}(y_1, y_0)]]] + \mathbb{E}_{x \sim P_X^1}[\mathbf{L}(\phi(x;\theta_1), \phi(x;\theta_0))]) \tag{12}$$
$$+ q_0 (\mathbb{E}_{x \sim P_X^0}[\mathbb{E}_{y_0 \sim P_{Y_0}^x}[\mathbb{E}_{y_1 \sim P_{Y_1}^x}[\mathbf{L}(y_0, y_1)]]] + \mathbb{E}_{x \sim P_X^0}[\mathbf{L}(\phi(x;\theta_0), \phi(x;\theta_1))])$$

$$= q_0 \mathbb{E}_{x \sim P_X^0}[\mathbf{L}(\phi(x;\theta_0), \phi(x;\theta_1))] + q_1 \mathbb{E}_{x \sim P_X^1}[\mathbf{L}(\phi(x;\theta_1), \phi(x;\theta_0))]$$
$$+ q_0 \mathbb{E}_{x \sim P_X^0}[\mathbb{E}_{y_0 \sim P_{Y_0}^x}[\mathbb{E}_{y_1 \sim P_{Y_1}^x}[\mathbf{L}(y_0, y_1)]]] + q_1 \mathbb{E}_{x \sim P_X^1}[\mathbb{E}_{y_1 \sim P_{Y_1}^x}[\mathbb{E}_{y_0 \sim P_{Y_0}^x}[\mathbf{L}(y_1, y_0)]]]$$

$$= q_0 \psi_\phi^1 + q_1 \psi_\phi^0 + q_0 \delta^1 + q_1 \delta^0 \tag{13}$$

$$= \psi_\phi + \delta \tag{14}$$

Finally, we are getting the causal bound:

$$\epsilon_{F_\phi} \le \epsilon_{CF_\phi} + \psi_\phi + \delta$$

Equality 6 is by Definitions B.6 and B.8, equality 7 is by Definitions B.5 and B.7, equality 8 is by Definition B.4, equality 9 is by linearity of expectations, equality 10 is by linearity of expectations and by Lemma B.11, inequality 11 is by Assumption B.1, equality 12 is by linearity of expectations and because $\phi$ is a constant on the expectation of $Y_0, Y_1$, equality 13 is by Definition B.9 and $\delta^1, \delta^0$ notation, equality 14 is by Definition B.10 and $\delta$ notation. $\qquad\square$

## B.3 Causal Bound for $N$ Control Treatments

We now adjust the definitions and Theorem B.12 proof from the binary treatment scenario to a single target treatment and N control treatments scenario.

We first set the definitions of the Neyman-Rubin potential outcome framework for the $N$ control treatments scenario: $\mathcal{T} = \{1, \ldots, N+1\}$ is the treatments space, where $t = j : 2 \le j \le N+1$ is the control treatments, and $t = 1$ is the target treatment compared against each control treatment. Under this framework, there are two types of potential outcomes depending on the treatment assignment: if $t = 1$, the observed outcome $y = Y_1$ is from the target treatment, and $\forall j \in \{2, \ldots, N+1\}$ if $t = j$, the observed outcome $y = Y_j$ is from a control treatment.

We assume joint distribution $P(X, T, Y_1, Y_2, \ldots, Y_{N+1})$ with the following components: $q_t := \Pr(T = t)$ is the treatment probability. $P_X^t := P(X|T = t)$ is the conditional distribution of observations. $P_{Y_1}^x := P(Y_1|X = x)$ is the conditional distribution of potential outcomes given the target treatment, and $\forall j \in \{2, \ldots, N+1\}$ $P_{Y_j}^x := P(Y_j|X = x)$ is the conditional distribution of potential outcomes given a control treatment $t = j$.

**Definition B.13.** The expected factual outcome loss for $N$ control treatments:

$$\epsilon_{F_\phi}^1 := \mathbb{E}_{x \sim P_X^1}[\ell_\phi(x, 1)].$$

$\forall j \in \{2, \ldots, N+1\}:$

$$\epsilon_{F_\phi}^j := \mathbb{E}_{x \sim P_X^j}[\ell_\phi(x, j)].$$

**Definition B.14.** The expected combined factual outcome loss for $N$ control treatments:

$$\epsilon_{F_\phi} := q_1 \epsilon_{F_\phi}^1 + \sum_{j=2}^{N+1} q_j \epsilon_{F_\phi}^j.$$

**Definition B.15.** The expected counterfactual outcome loss for $N$ control treatments:
$\forall j \in \{2, \ldots, N+1\}:$

$$\epsilon_{CF_\phi}^{1,j} := \mathbb{E}_{x \sim P_X^j}[\ell_\phi(x, 1)].$$

$$\epsilon_{CF_\phi}^{j} := \mathbb{E}_{x \sim P_X^1}[\ell_\phi(x, j)].$$

**Definition B.16.** The expected combined counterfactual outcome loss for $N$ control treatments:

$$\epsilon_{CF_\phi} := \sum_{j=2}^{N+1} \left( q_j \epsilon_{CF_\phi}^{1,j} + \frac{1}{N} q_1 \epsilon_{CF_\phi}^{j} \right).$$

**Definition B.17.** The estimated treatment effect loss for $N$ control treatments:
$\forall j \in \{2, \ldots, N+1\}:$

$$\psi_\phi^{1,j} := \mathbb{E}_{x \sim P_X^j}[\mathbf{L}(\phi(x; \theta_j), \phi(x; \theta_1))].$$

$$\psi_\phi^{j} := \mathbb{E}_{x \sim P_X^1}[\mathbf{L}(\phi(x; \theta_1), \phi(x; \theta_j))].$$

**Definition B.18.** The combined estimated treatment effect loss for $N$ control treatments:

$$\epsilon_{\psi_\phi} := \sum_{j=2}^{N+1} \left( q_j \psi_\phi^{1,j} + \frac{1}{N} q_1 \psi_\phi^{j} \right).$$

All the other definitions and proofs are the same as the binary case.

We now generalize Theorem B.12 from a single control treatment scenario to the $N$ control treatments scenario as follows:

**Theorem B.19.**

$$q_1 \epsilon_{F_\phi}^1 + \sum_{j=2}^{N+1} q_j \epsilon_{F_\phi}^j \leq \sum_{j=2}^{N+1} \left( q_j \epsilon_{CF_\phi}^{1,j} + \frac{1}{N} q_1 \epsilon_{CF_\phi}^{j} + q_j \psi_\phi^{1,j} + \frac{1}{N} q_1 \psi_\phi^{j} + q_j \delta^{1,j} + \frac{1}{N} q_1 \delta^{j} \right).$$

Where $\delta$ is the following:

$$\delta^{1,j} := \mathbb{E}_{x \sim P_X^j}[\mathbb{E}_{y_j \sim P_{Y_j}^x}[\mathbb{E}_{y_1 \sim P_{Y_1}^x}[\mathbf{L}(y_j, y_1)]]].$$

$$\delta^{j} := \mathbb{E}_{x \sim P_X^1}[\mathbb{E}_{y_1 \sim P_{Y_1}^x}[\mathbb{E}_{y_j \sim P_{Y_j}^x}[\mathbf{L}(y_1, y_j)]]].$$

$$\delta := \sum_{j=2}^{N+1} \left( q_j \delta^{1,j} + \frac{1}{N} q_1 \delta^{j} \right).$$

*Proof.* by induction.

Step 1: Base Case

For a single target treatment and a single control treatment, $N = 1$, the following inequality holds:

$$q_1 \epsilon_{F_\phi}^1 + q_2 \epsilon_{F_\phi}^2 - q_2 \epsilon_{CF_\phi}^{1,2} - q_1 \epsilon_{CF_\phi}^2 \leq q_2 \psi_\phi^{1,2} + q_1 \psi_\phi^2 + q_2 \delta^{1,2} + q_1 \delta^2$$

As we prove in Theorem B.12 where the control treatment is $t = 2$ instead of $t = 0$.

Step 2: Inductive Hypothesis

Assume that for $N - 1 \geq 2$ control treatments, the following inequality holds:

$$\sum_{j=2}^{N} \left( \frac{1}{N-1} q_1 \epsilon_{F_\phi}^1 + q_j \epsilon_{F_\phi}^j - q_j \epsilon_{CF_\phi}^{1,j} - \frac{1}{N-1} q_1 \epsilon_{CF_\phi}^j \right) \leq$$

$$\sum_{j=2}^{N} \left( q_j \psi_\phi^{1,j} + \frac{1}{N-1} q_1 \psi_\phi^j + q_j \delta^{1,j} + \frac{1}{N-1} q_1 \delta^j \right)$$

Step 3: Inductive Step

We will now prove the inequality for $N$ control treatments:

$$\sum_{j=2}^{N+1} \left( \frac{1}{N} q_1 \epsilon_{F_\phi}^1 + q_j \epsilon_{F_\phi}^j - q_j \epsilon_{CF_\phi}^{1,j} - \frac{1}{N} q_1 \epsilon_{CF_\phi}^j \right) \leq \sum_{j=2}^{N+1} \left( q_j \psi_\phi^{1,j} + \frac{1}{N} q_1 \psi_\phi^j + q_j \delta^{1,j} + \frac{1}{N} q_1 \delta^j \right)$$

Inductive Step proof.

$$\sum_{j=2}^{N+1} \left( \frac{1}{N} q_1 \epsilon_{F_\phi}^1 + q_j \epsilon_{F_\phi}^j - q_j \epsilon_{CF_\phi}^{1,j} - \frac{1}{N} q_1 \epsilon_{CF_\phi}^j \right) =$$

$$\sum_{j=2}^{N} \left( \frac{1}{N} q_1 \epsilon_{F_\phi}^1 + q_j \epsilon_{F_\phi}^j - q_j \epsilon_{CF_\phi}^{1,j} - \frac{1}{N} q_1 \epsilon_{CF_\phi}^j \right) + \frac{1}{N} q_1 \epsilon_{F_\phi}^1 + q_{N+1} \epsilon_{F_\phi}^{N+1}$$

$$- q_{N+1} \epsilon_{CF_\phi}^{1,N+1} - \frac{1}{N} q_1 \epsilon_{CF_\phi}^{N+1}$$

$$\leq \sum_{j=2}^{N} \left( q_j \psi_\phi^{1,j} + \frac{1}{N} q_1 \psi_\phi^j + q_j \delta^{1,j} + \frac{1}{N} q_1 \delta^j \right)$$

$$\tag{15}$$

$$+ \frac{1}{N} q_1 \epsilon_{F_\phi}^1 + q_{N+1} \epsilon_{F_\phi}^{N+1}$$

$$- q_{N+1} \epsilon_{CF_\phi}^{1,N+1} - \frac{1}{N} q_1 \epsilon_{CF_\phi}^{N+1}$$

$$\leq \sum_{j=2}^{N} \left( q_j \psi_\phi^{1,j} + \frac{1}{N} q_1 \psi_\phi^j + q_j \delta^{1,j} + \frac{1}{N} q_1 \delta^j \right)$$

$$\tag{16}$$

$$+ q_{N+1} \psi_\phi^{1,N+1} + \frac{1}{N} q_1 \psi_\phi^{N+1}$$

$$+ q_{N+1} \delta^{1,N+1} + \frac{1}{N} q_1 \delta^{N+1}$$

$$= \sum_{j=2}^{N+1} \left( q_j \psi_\phi^{1,j} + \frac{1}{N} q_1 \psi_\phi^j + q_j \delta^{1,j} + \frac{1}{N} q_1 \delta^j \right)$$

We will get the following inequality:

$$\sum_{j=2}^{N+1} \left( \frac{1}{N} q_1 \epsilon_{F_\phi}^1 + q_j \epsilon_{F_\phi}^j \right) \leq \sum_{j=2}^{N+1} \left( q_j \epsilon_{CF_\phi}^{1,j} + \frac{1}{N} q_1 \epsilon_{CF_\phi}^j + q_j \psi_\phi^{1,j} + \frac{1}{N} q_1 \psi_\phi^j + q_j \delta^{1,j} + \frac{1}{N} q_1 \delta^j \right)$$

Finally, we sum $\frac{1}{N} q_1 \epsilon_{F_\phi}^1$ and get the causal bound for $N$ control treatments:

$$q_1 \epsilon_{F_\phi}^1 + \sum_{j=2}^{N+1} q_j \epsilon_{F_\phi}^j \leq \sum_{j=2}^{N+1} \left( q_j \epsilon_{CF_\phi}^{1,j} + \frac{1}{N} q_1 \epsilon_{CF_\phi}^j + q_j \psi_\phi^{1,j} + \frac{1}{N} q_1 \psi_\phi^j + q_j \delta^{1,j} + \frac{1}{N} q_1 \delta^j \right)$$

Figure 4: Illustration diagram that demonstrates the reduction from the causal inference framework (Left) to the DRL framework (Right).

Inequality 15 is by the inductive hypothesis, Inequality 16 is from the proof of Theorem B.12 where the control treatment is $t = N + 1$ instead of $t = 0$. $\square$

## B.4   Reduction from Causal to DRL Framework

We now formalize a reduction from a causal framework to a DRL framework.

In the DRL framework, we consider the treatment to be the agent's policy, where the target treatment matches the target policy, and the control treatments match the behavior policies.

Within the causal framework, the on-policy loss corresponds to the factual loss when the treatment being optimized matches the treatment applied for generating the observations. This is analogous to optimizing the target policy using experiences from an identical behavior policy. Similarly, the off-policy loss aligns with the counterfactual loss when the treatment being optimized differs from the one used to generate the observations. This is equivalent to optimizing the target policy using experiences from different behavior policies.

To formalize the corresponding definition in the DRL framework, consider a replay buffer $D_1$ containing samples produced by a behavior policy that matches the target policy. In this context, the on-policy loss is a special case of the expected factual loss.

Similarly, consider a replay buffer $D_0$ that stores experiences generated by behavior policies that differ from the target policy. In this context, the off-policy loss is a special case of the expected counterfactual loss.

**Q-Network Reduction:**   The definitions for the scenario of an agent using a Q-value network:

$\mathcal{X} := (\mathcal{S} \times \mathcal{A})$ is the state-action space. Potential outcome $y \sim P_{Y_t}^{(s,a)}$ is the accumulated reward over n time steps, starting from state $s$, taking the initial action $a$, and following policy $\pi_t$. And the hypothesis $\phi(x; \theta_t) := Q(s, a; \theta_t) : (\mathcal{S} \times \mathcal{A}) \to \mathcal{Y}$ is the Q-values network, which influences the policy $\pi_t$.

Definition B.4 in the DRL framework refers to the temporal difference loss between the accumulated rewards and the Q-value estimation, and it is defined as follows:

**Definition B.20.** The expected temporal difference loss:

$$\ell_Q(s, a, t) := \mathbb{E}_{y \sim P_{Y_t}^{(s,a)}}[\mathbf{L}(y, Q(s, a; \theta_t))].$$

Definitions B.5 and B.13 in the DRL framework assigns the temporal difference loss with experiences collected on-policy, with identical target and behavior policies from $D_1$, and it is defined as follows:

**Definition B.21.** The expected on-policy loss:

$$\epsilon_{\text{On-Policy}_Q} := \mathbb{E}_{(s,a)\sim D_1}[\ell_Q(s, a, t_{\text{target}})].$$

Definitions B.7 and B.15 in the DRL framework assigns the temporal difference loss with experiences gathered off-policy, with different target and behavior policies from $D_0$, and it is defined as follows:

**Definition B.22.** The expected off-policy loss:

$$\epsilon_{\text{Off-Policy}_Q} := \mathbb{E}_{(s,a)\sim D_0}[\ell_Q(s, a, t_{\text{target}})].$$

Definitions B.9 and B.17 interpretation in the DRL framework is an OPE term, and it is defined as follows:

**Definition B.23.** The SUFT OPE term:

$$\psi_{\text{SUFT}_Q} := \mathbb{E}_{(s,a)\sim D_0}[\mathbf{L}\left(Q(s, a; \theta_{\text{behavior}}), Q(s, a; \theta_{\text{target}})\right)].$$

Note that this term evaluates the target policy by measuring the loss between the Q-value estimations of the target policy and the behavior policies, given samples from $D_0$.

Theorems B.12 and B.19 interpretation in the DRL framework establishes an upper bound to the on-policy loss, combining the off-policy loss and the SUFT OPE term.

$$\epsilon_{\text{On-Policy}_Q} \leq \epsilon_{\text{Off-Policy}_Q} + \psi_{\text{SUFT}_Q} + \delta.$$

Note that since the constant term, $\delta$, is independent of $Q$, it is disregarded during the gradient-based optimization of $Q$, so the SUFT causal bound objective term is as follows:

$$\epsilon_{\text{Off-Policy}_Q} + \lambda_{\text{TF}} \cdot \psi_{\text{SUFT}_Q}.$$

**V-Network Reduction:**    Now we present the corresponding definitions for the scenario of an agent using a V network:

$\mathcal{X} := \mathcal{S}$ is the state space. Potential outcome $y \sim P_{Y_t}^s$ is the accumulated reward over n time steps, starting from state $s$, and following policy $\pi_t$. And the hypothesis $\phi(x; \theta_t) := V(s; \theta_t) : \mathcal{S} \to \mathcal{Y}$ is the V network, which influences the policy $\pi_t$.

Definition B.4 in the DRL framework refers to the temporal difference loss between the accumulated rewards and the V estimation, and it is defined as follows:

**Definition B.24.** The expected temporal difference loss:

$$\ell_V(s, t) := \mathbb{E}_{y\sim P_{Y_t}^s}[\mathbf{L}\left(y, V(s; \theta_t)\right)].$$

Definitions B.5 and B.13 in the DRL framework assigns the temporal difference loss with experiences collected on-policy, with identical target and behavior policies from $D_1$, and it is defined as follows:

**Definition B.25.** The expected on-policy loss:

$$\epsilon_{\text{On-Policy}_V} := \mathbb{E}_{s\sim D_1}[\ell_V(s, t_{\text{target}})].$$

Definitions B.7 and B.15 in the DRL framework assigns the temporal difference loss with experiences gathered off-policy, with different target and behavior policies from $D_0$, and it is defined as follows:

**Definition B.26.** The expected off-policy loss:

$$\epsilon_{\text{Off-Policy}_V} := \mathbb{E}_{s\sim D_0}[\ell_V(s, t_{\text{target}})].$$

Definitions B.9 and B.17 interpretation in the DRL framework is an OPE term, and it is defined as follows:

**Definition B.27.** The SUFT OPE term:

$$\psi_{\text{SUFT}_V} := \mathbb{E}_{s\sim D_0}[\mathbf{L}\left(V(s; \theta_{\text{behavior}}), V(s; \theta_{\text{target}})\right)].$$

Note that this term evaluates the target policy by measuring the loss between the V estimations of the target policy and the behavior policies, given samples from $D_0$.

Theorems B.12 and B.19 interpretation in the DRL framework establishes an upper bound to the on-policy loss, combining the off-policy loss and the SUFT OPE term.

$$\epsilon_{\text{On-Policy}_V} \leq \epsilon_{\text{Off-Policy}_V} + \psi_{\text{SUFT}_V} + \delta.$$

Note that since the constant term, $\delta$, is independent of $V$, it is disregarded during the gradient-based optimization of $V$. Thus, the SUFT causal bound objective term is as follows:

$$\epsilon_{\text{Off-Policy}_V} + \lambda_{\text{TF}} \cdot \psi_{\text{SUFT}_V}.$$

**Causal and DRL Framework Alignment:** An important alignment needs to be made to address a difference between the static observational data assumed by the causal framework and the dynamic experience replay buffer in the online DRL framework. In the causal framework, all observations are gathered once, so the data is fixed and irrevocably labeled either to be from the target treatment or from the control treatment in the binary treatment case. On the other hand, in the online DRL framework, the observational data changes over time, as we continuously populate the experience replay buffer and update the target policy. On each policy optimization step, all of the samples stored in the buffer that were labeled as the target policy are turned into samples that are labeled as a control policy. This phenomenon of changing the observational data labels when updating the policy in the DRL framework is a gap between the causal and DRL frameworks. In the on-policy case, the experience replay buffer contains only on-policy observations, meaning that $q_1 = 1$. When updating the target policy, this data becomes off-policy, and the data that was previously labeled as target policy turns into control policy, shifting the samples' labels together with the treatment probability $q_{0_{\text{new}}} = q_{1_{\text{old}}} = 1$.

The reduction we have made between the causal and the DRL framework above aligns the on-policy loss and off-policy loss terms with the corresponding terms in the causal framework. The on-policy loss is a special case of the factual loss where the observation data is filled with target treatment samples, corresponding to $\epsilon_F^1$ where $q_1 = 1$. The off-policy loss is a special case of the counterfactual loss where the observation data is filled with control treatment samples, corresponding to $\epsilon_{CF}^1$ where the labels and the treatment probability are shifted and $q_{0_{\text{new}}} = q_{1_{\text{old}}} = 1$.

## C SUFT Pseudocode

### C.1 SUFT DQN Pseudocode

---
**Algorithm 1** SUFT Deep Q-learning
---
Initialize replay memory $D_0$ to capacity $N$
Initialize action-value function $Q$ with random weights $\theta$
**for** episode $= 1, M$ **do**
    Initialize start state $s_1$
    **for** $t = 1, T$ **do**
        With probability $\rho$ select a random action $a_t$
        otherwise select $a_t = \arg\max_a Q(s_t, a; \theta_{\text{behavior}})$
        Execute action $a_t$ in the environment and observe reward $r_t$ and state $s_{t+1}$
        Store transition $(s_t, a_t, r_t, s_{t+1}, Q(s_t, a_t; \theta_{\text{behavior}}))$ in $D_0$
        Sample random mini-batch of transitions $(s_j, a_j, r_j, s_{j+1}, Q(s_j, a_j; \theta_{\text{behavior}}))$ from $D_0$
        Set $y_j = \begin{cases} r_j & \text{for terminal } s_{j+1} \\ r_j + \gamma \max_{a'} Q(s_{j+1}, a'; \theta_{\text{target}}) & \text{for non-terminal } s_{j+1} \end{cases}$
        Perform a gradient descent step with respect to the network parameters $\theta_{\text{target}}$ on:

$$\hat{\mathbb{E}}_j \left[ (y_j - Q(s_j, a_j; \theta_{\text{target}}))^2 + \lambda_{\text{TF}} \cdot (Q(s_j, a_j; \theta_{\text{behavior}}) - Q(s_j, a_j; \theta_{\text{target}}))^2 \right]$$

    **end for**
**end for**

---

Our method extends the standard DQN algorithm [80] in two main ways: The first is by storing the old Q values $Q(s, a; \theta_{\text{behavior}})$ in the replay buffer. The second is by incorporating the additional SUFT OPE term into the objective function $\lambda_{\text{TF}} \cdot (Q(s_j, a_j; \theta_{\text{behavior}}) - Q(s_j, a_j; \theta_{\text{target}}))^2$, which represents the estimated treatment effect. $\theta_{\text{behavior}}$ refers to the network weights used for action selection, the behavior policy, and $\theta_{\text{target}}$ represents the current network weights, which are optimized in the gradient descent step, the target policy.

## C.2 SUFT PPO Pseudocode

---

**Algorithm 2** SUFT Proximal Policy Optimization (PPO)

---

Initialize a rollout buffer $D_1$ with horizon $T$

Initialize policy function (actor) and value function (critic) with random weights $\theta$

**for** iteration $= 1, 2, \ldots$ **do**

    **for** $t = 1, T$ **do**

        Run policy $\pi_{\theta_{\text{behavior}}}(s_t)$

        Store transition $(s_t, a_t, r_t, \pi_{\theta_{\text{behavior}}}(a_t \mid s_t), V(s_t; \theta_{\text{behavior}}))$ in $D_1$

    **end for**

    Compute advantage estimates $\hat{A}_1, \ldots, \hat{A}_t$, and rewards to go $\hat{R}_1, \ldots, \hat{R}_t$

    based on the stored values $V(s_1; \theta_{\text{behavior}}), \ldots, V(s_t; \theta_{\text{behavior}})$

    **for** $k = 1, K$ epochs **do**

        Sample random mini-batch of transitions $(s_j, a_j, r_j, \pi_{\theta_{\text{behavior}}}(a_j \mid s_j), V(s_j; \theta_{\text{behavior}}))$

        from the rollout buffer $D_1$

        Compute probability ratio:

$$r_j(\theta) = \frac{\pi_{\theta_{\text{target}}}(a_j \mid s_j)}{\pi_{\theta_{\text{behavior}}}(a_j \mid s_j)}$$

        Optimize actor loss:

$$\hat{\mathbb{E}}_j \left[ \min \left( r_j(\theta)\hat{A}_j, \ \text{clip}\left(r_j(\theta), 1 - \epsilon, 1 + \epsilon\right) \hat{A}_j \right) \right]$$

        Optimize critic loss:

$$\hat{\mathbb{E}}_j \left[ (V(s_j; \theta_{\text{target}}) - \hat{R}_j)^2 + \lambda_{\text{TF}} \cdot (V(s_j; \theta_{\text{behavior}}) - V(s_j; \theta_{\text{target}}))^2 \right]$$

    **end for**

**end for**

---

Our method extends the standard PPO algorithm [20] by incorporating the additional SUFT OPE term into the PPO critic loss. Notably, PPO performs multiple epochs of optimization over the same rollout buffer $D_1$, leading to a divergence between the behavior and target policies. This divergence is the reason PPO benefits from our method, despite being an on-policy algorithm. $\theta_{\text{behavior}}$ refers to the network weights used to interact with the environment also referred to in [20] as $\theta_{\text{old}}$, the behavior policy, and $\theta_{\text{target}}$ represents the current network weights, which are optimized, also referred to in [20] as $\theta$, the target policy. During the first epoch, before optimizing the network, the behavior and target policies are identical, so the SUFT OPE term equals zero. After the first optimization step, the target policy diverges from the behavior policy, and the SUFT OPE term now quantifies the estimated treatment effect between the target and behavior policies. In addition, the $V(s_t; \theta_{\text{behavior}})$ values are already calculated during the calculation of the advantage estimates $\hat{A}_t$, and rewards to go $\hat{R}_t$. Thus, our method effectively reuses these stored network outputs for computing the SUFT OPE term, similar to the approach used in our SUFT DQN implementation.

# D  Full Experimental Results

In this section, we present detailed and comprehensive experimental results for both the Atari and the MuJoCo domains, as well as the formulas and criteria used to evaluate the results.

## D.1  Negligible Computational Overhead

We quantify the computational and space overhead introduced by computing the SUFT OPE term and storing old Q-values in the experience replay buffer. We run our experiments on a single NVIDIA L4 GPU, and for the Vanilla DQN agent, the additional computation takes approximately 23.77 seconds, compared to a total training time of 2,980 seconds. This corresponds to a 0.8% computational overhead. In terms of space, we append a single 64-bit scalar to each stored experience. For a 4K experience replay buffer, this results in an additional 0.0305 MB, compared to 972.85 MB without it. This is equivalent to a 0.003% space overhead.

## D.2  Tuning the $\lambda_{TF}$ Coefficient

In order to evaluate the impact of the $\lambda_{TF}$ coefficient on improving the loss accuracy, we conducted experiments using various $\lambda_{TF}$ values. Adding the $\lambda_{TF}$ demonstrates a consistent improvement across a range of $\lambda_{TF}$ values. This indicates that precise hyperparameter tuning is unnecessary, and as long as the SUFT OPE term is added with a coefficient within a valid range of $\lambda_{TF}$, the loss optimization improves. The $\lambda_{TF}$ value can be easily tuned through random search hyperparameter optimization within a valid range. Specifically, we encounter that a sufficient range to search from is between $\lambda_{TF} = 0.5$ and $\lambda_{TF} = 1.5$. The $\lambda_{TF}$ values that yield the best performance are $\lambda_{TF} = 1$ for all the DQN agents, $\lambda_{TF} = 2.5$ for the PPO agent using the Using $L_1$ Loss and $\lambda_{TF} = 5$ for the PPO agent using the Using $L_2$ Loss in the Atari domain. In the MuJoCo domain, SAC performs best at $\lambda_{TF} = 0.6$ and PPO at $\lambda_{TF} = 1.8$.

## D.3  Evaluation Formulas

To evaluate our method, we train every agent-environment configuration for 10 different runs and report the median result for each setup. Since this is an even-numbered list, there are two central values, and we use the higher of the two as the median. The valid environments used for calculating the improvement percentage and mean reward ratio percentage formulas are those where the numerator and denominator are greater than one. This restriction allows us to exclude cases where the reward is only marginally higher than the random reward, which could otherwise yield arbitrarily large ratios. By doing so, we ensure that the metric remains stable and effectively reflects meaningful improvements.

**Improvement Percentage Formula:**  The improvement percentage for comparing our approach to an identical agent without the additional term is calculated with the following formula:

$$\frac{\text{Higher reward approach} - \text{Lower reward approach}}{\text{Lower reward approach} - \text{Random reward}} \cdot 100\%$$

For better visualization of the results, we display them in log scale with the following formula:

$$\log_{10}\left[\text{Improvement Percentage Formula} + 1\right]$$

We add 1 to the improvement percentage formula to avoid the log of 0 and fractions, which result in a negative log. This addition has no impact on the results.

**Mean Reward Ratio Percentage Formula:**  The mean reward ratio percentage for comparing our approach to an identical agent without the additional term is calculated with the following formula:

$$\frac{1}{|\text{ Valid Envs }|}\sum_{\forall\text{env}}\frac{\text{SUFT reward} - \text{Random reward}}{\text{Baseline reward} - \text{Random reward}} \cdot 100\%.$$

Note that for the Atari domain, the human normalized results are calculated using the above formula, where the baseline reward is replaced with the human reward.

## D.4  Assumption 4.1 Empirical Analysis with $L_2$

DRL agents are often trained using $L_2$ loss, for which Assumption 4.1 does not hold. To provide intuition for the observed $L_2$ performance, we empirically examine the validity of the $L_2$ loss for Assumption 4.1 using synthetic and domain-specific data. For the synthetic data, we generate one billion independent samples from the standard normal distribution, $x, y, x', y' \sim \mathcal{N}(0, 1)$, and evaluate whether:

$$(x - y)^2 - (x' - y')^2 \le (x - x')^2 + (y - y')^2 \tag{17}$$

This experiment corresponds directly to the inequality form of Assumption 4.1 when applied to the $L_2$ loss. We find that inequality 17 holds in approximately 82% of random samples, demonstrating that while Assumption 4.1 is not guaranteed theoretically, it is satisfied in the vast majority of cases. To further validate this observation in a domain-specific dataset, we conduct an experiment in the Atari 57 benchmark using a DQN agent trained for 400K steps with a batch size of 32. Since our method operates in an off-policy setting, the left side of the causal bound in Theorem 4.10 is not computable from the observed data, as we don't observe the on-policy experiences (this is the reason we optimize on the causal bound in our method). For this reason, we evaluate only inequality 11 used in the proof of Theorem B.12, considering the off-policy terms on both sides (with $q_0 = 1$):

$$\left(\mathbb{E}_{x \sim P_X^0}\left[\mathbb{E}_{y_0 \sim P_{Y_0}^x}\left[\mathbb{E}_{y_1 \sim P_{Y_1}^x}\left[\mathbf{L}(y_0, \phi(x; \theta_0)) - \mathbf{L}(y_1, \phi(x; \theta_1))\right]\right]\right]\right) \le \tag{18}$$
$$\left(\mathbb{E}_{x \sim P_X^0}\left[\mathbb{E}_{y_0 \sim P_{Y_0}^x}\left[\mathbb{E}_{y_1 \sim P_{Y_1}^x}\left[\mathbf{L}(y_0, y_1) + \mathbf{L}(\phi(x; \theta_0), \phi(x; \theta_1))\right]\right]\right]\right)$$

Because the true values of $y_0$ and $y_1$ are not directly observable, since they are the true values from the given state until the end of the episode, we approximate them using the standard Bellman equation for Q-values. Under this setup, we find that inequality 18 holds in approximately 79% of cases across the 57 Atari games. These empirical findings indicate that although Assumption 4.1 is not theoretically guaranteed for $L_2$, it holds in the vast majority of the optimization steps in practice, which may explain the strong empirical performance of our method using the $L_2$ loss.

## D.5  Empirical Evaluation of the $\delta$ Term Magnitude

In Theorem 4.10 the $\delta$ term is ignored during optimization since it is independent of $Q$. To assess whether this omission introduces significant looseness in the causal bound, we conduct an empirical analysis on the Atari 57 benchmark using a DQN agent trained for 400K steps. Specifically, we plotted the magnitudes of the standard DQN loss, the $\delta$ term, and the SUFT OPE term throughout training. As discussed in Appendix D.4, the $\delta$ term involves the true values of $y_0$ and $y_1$, which are not directly observable. Thus, we approximate them using the standard Bellman equation for Q-values. The results, shown in Figure 5, indicate that across all of the environments, the $\delta$ term remains consistently an order of magnitude lower than the standard DQN loss, and on the same order of magnitude as the SUFT OPE term. These findings support the decision to ignore the $\delta$ term during optimization, as it does not meaningfully loosen the causal bound.

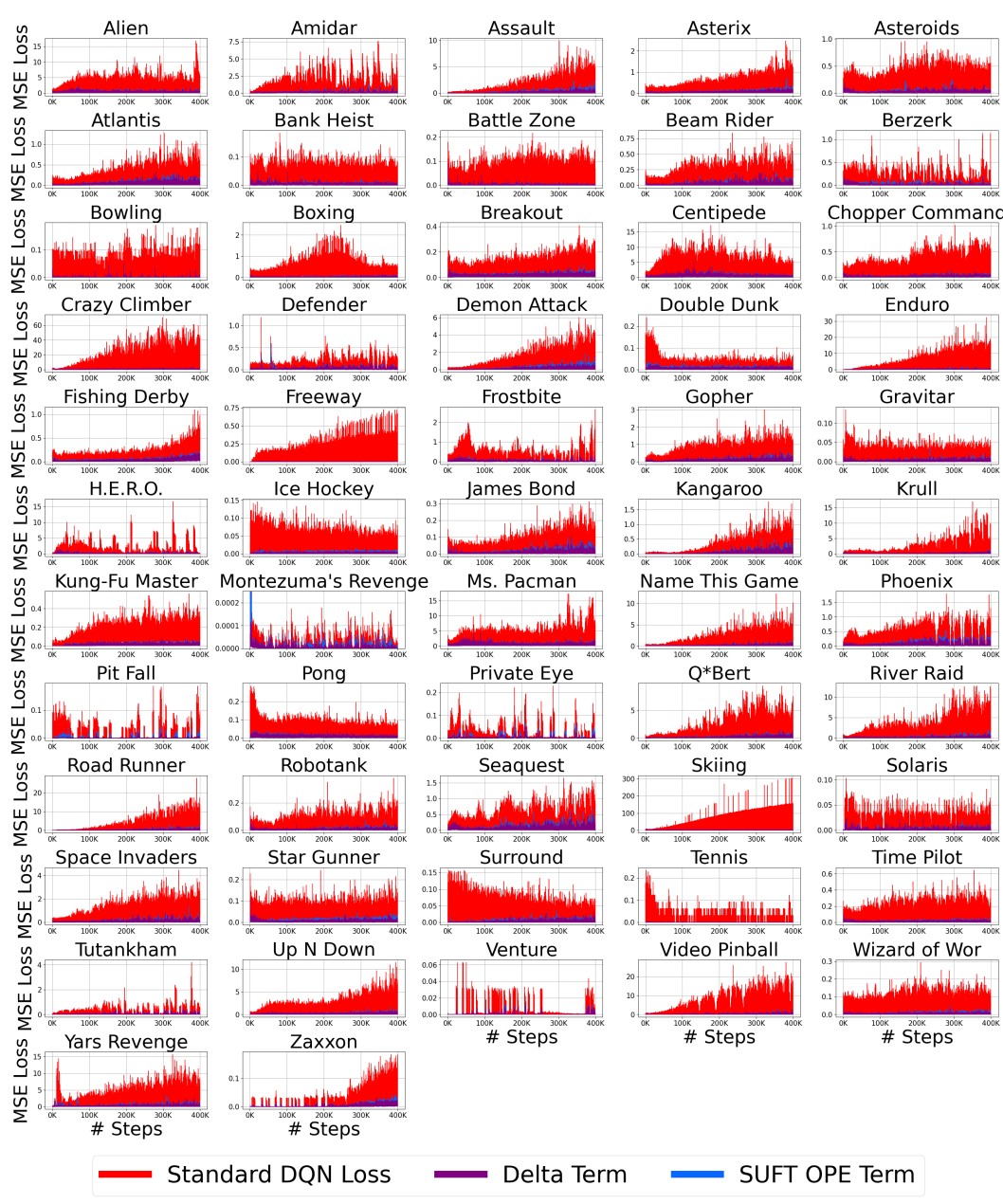

Figure 5: Magnitude of the $\delta$ term, SUFT OPE term, and the standard DQN loss across the Atari 57 benchmark.

## D.6 Atari 2600 57 Games Benchmark

Table 2: Atari 2600 random and average human rewards. The scores are sourced from [79].

| Game | Random | Human |
|------|--------|-------|
| Alien | 227.8 | 6371.3 |
| Amidar | 5.8 | 1540.4 |
| Assault | 222.4 | 628.9 |
| Asterix | 164.5 | 7536 |
| Asteroids | 871.3 | 36517.3 |
| Atlantis | 13463 | 26575 |
| Bank Heist | 14.2 | 644.5 |
| Battle Zone | 3560 | 33030 |
| Beam Rider | 254.6 | 14961 |
| Berzerk | 196.1 | 2237.5 |
| Bowling | 35.2 | 146.5 |
| Boxing | -1.5 | 9.6 |
| Breakout | 1.6 | 27.9 |
| Centipede | 2090.9 | 10321.9 |
| Chopper Command | 811 | 7387.8 |
| Crazy Climber | 10780.5 | 32667 |
| Defender | 2874.5 | 14296 |
| Demon Attack | 208.3 | 1971 |
| Double Dunk | -18.6 | -16.4 |
| Enduro | 0 | 740.2 |
| Fishing Derby | -77.1 | -38.7 |
| Freeway | 0.1 | 25.6 |
| Frostbite | 65.2 | 4202.8 |
| Gopher | 257.6 | 2311 |
| Gravitar | 173 | 3116 |
| H.E.R.O. | 1580.3 | 25839.4 |
| Ice Hockey | -11.2 | 0.5 |
| James Bond | 33.5 | 302.8 |
| Kangaroo | 100 | 2739 |
| Krull | 1598 | 2109.1 |
| Kung-Fu Master | 258.5 | 20786.8 |
| Montezuma's Revenge | 25 | 4182 |
| Ms. Pacman | 307.3 | 6951.6 |
| Name This Game | 1747.8 | 6796 |
| Phoenix | 761.4 | 6686.2 |
| Pit Fall | -348.8 | 5998.9 |
| Pong | -20.7 | 14.6 |
| Private Eye | 662.8 | 64169.1 |
| Q*Bert | 163.9 | 12085 |
| River Raid | 1338.5 | 14382.2 |
| Road Runner | 200 | 6878 |
| Robotank | 2.2 | 8.9 |
| Seaquest | 215.5 | 40425.8 |
| Skiing | -15287.4 | -4336.9 |
| Solaris | 2047.2 | 11032.6 |
| Space Invaders | 182.6 | 1464.9 |
| Star Gunner | 697 | 9528 |
| Surround | -10 | 5.4 |
| Tennis | -23.8 | -8.3 |
| Time Pilot | 3568 | 5229.2 |
| Tutankham | 12.7 | 138.3 |
| Up N Down | 707.2 | 9896.1 |
| Venture | 18 | 1039 |
| Video Pinball | 20452 | 15641.1 |
| Wizard of Wor | 804 | 4556 |
| Yars Revenge | 3092.9 | 47135.2 |
| Zaxxon | 32.5 | 8443 |

**Atari PPO Results (Using $L_1$ Loss):**

Table 3: Training rewards comparison between the PPO agent using the additional SUFT OPE term and the baseline agent without it across 57 Atari games. SUFT surpasses the baseline agent in 40 games, with statistically significant gains in 31 games. Both agents are using $L_1$ loss.

| Game | Baseline PPO | SUFT PPO | p-value |
|---|---|---|---|
| Alien | **860.0** | 780.0 | N/A |
| Amidar | **184.0** | 133.0 | N/A |
| Assault | 631.0 | **692.0** | 9.15e-02 |
| Asterix | 346.0 | **568.0** | 1.17e-08 |
| Asteroids | 483.0 | **644.0** | 3.19e-07 |
| Atlantis | 36100.0 | **51300.0** | 4.50e-06 |
| Bank Heist | 37.5 | **51.5** | 6.14e-02 |
| Battle Zone | 8380.0 | **8880.0** | 6.07e-01 |
| Beam Rider | 450.0 | **526.0** | 2.34e-05 |
| Berzerk | 661.0 | **676.0** | 4.70e-01 |
| Bowling | **41.7** | 33.8 | N/A |
| Boxing | -10.0 | **15.1** | 1.35e-13 |
| Breakout | **14.4** | 10.4 | N/A |
| Centipede | 2860.0 | **3390.0** | 2.23e-06 |
| Chopper Command | 1350.0 | **1990.0** | 2.84e-05 |
| Crazy Climber | 26700.0 | **37800.0** | 4.67e-11 |
| Defender | 2850.0 | **4280.0** | 8.90e-08 |
| Demon Attack | 295.0 | **306.0** | 5.37e-01 |
| Double Dunk | -21.7 | **-18.1** | 2.19e-03 |
| Enduro | 13.1 | **67.8** | 3.18e-06 |
| Fishing Derby | -82.9 | **-79.5** | 2.95e-01 |
| Freeway | 20.2 | **24.6** | 6.60e-06 |
| Frostbite | **1610.0** | 261.0 | N/A |
| Gopher | 381.0 | **494.0** | 2.38e-05 |
| Gravitar | **318.0** | 211.0 | N/A |
| H.E.R.O. | **9810.0** | 8840.0 | N/A |
| Ice Hockey | -9.05 | **-5.86** | 2.65e-05 |
| James Bond | 324.0 | **466.0** | 3.54e-08 |
| Kangaroo | **3000.0** | 1080.0 | N/A |
| Krull | **4770.0** | 4360.0 | N/A |
| Kung-Fu Master | 1990.0 | **2130.0** | 1.05e-01 |
| Montezuma's Revenge | **0.0** | **0.0** | N/A |
| Ms. Pacman | 1210.0 | **1770.0** | 1.55e-06 |
| Name This Game | 2480.0 | **4520.0** | 2.10e-13 |
| Phoenix | 734.0 | **1700.0** | 2.19e-11 |
| Pit Fall | **-1.28** | -4.73 | N/A |
| Pong | -16.5 | **-12.0** | 4.95e-03 |
| Private Eye | **84.3** | 25.7 | N/A |
| Q*Bert | **2080.0** | 689.0 | N/A |
| River Raid | 2560.0 | **2590.0** | 5.53e-01 |
| Road Runner | 2330.0 | 8600.0 | 4.66e-03 |
| Robotank | **4.88** | 4.1 | N/A |
| Seaquest | 576.0 | **689.0** | 4.14e-03 |
| Skiing | -30000.0 | **-18000.0** | 2.19e-04 |
| Solaris | 1850.0 | **1880.0** | 6.68e-01 |
| Space Invaders | 292.0 | **406.0** | 7.12e-06 |
| Star Gunner | 1030.0 | **1090.0** | 7.81e-04 |
| Surround | -9.7 | **-9.04** | 1.59e-02 |
| Tennis | **-13.3** | -15.6 | N/A |
| Time Pilot | 2790.0 | **3610.0** | 4.88e-05 |
| Tutankham | 82.1 | **102.0** | 2.85e-04 |
| Up N Down | 3090.0 | **19100.0** | 7.83e-06 |
| Venture | **16.0** | 8.0 | N/A |
| Video Pinball | 9030.0 | **14600.0** | 9.20e-06 |
| Wizard of Wor | 508.0 | **1060.0** | 2.09e-09 |
| Yars Revenge | 5230.0 | **10700.0** | 2.87e-12 |
| Zaxxon | **1040.0** | 271.0 | N/A |
| Mean (%) | 100.0 | **164.7** | N/A |

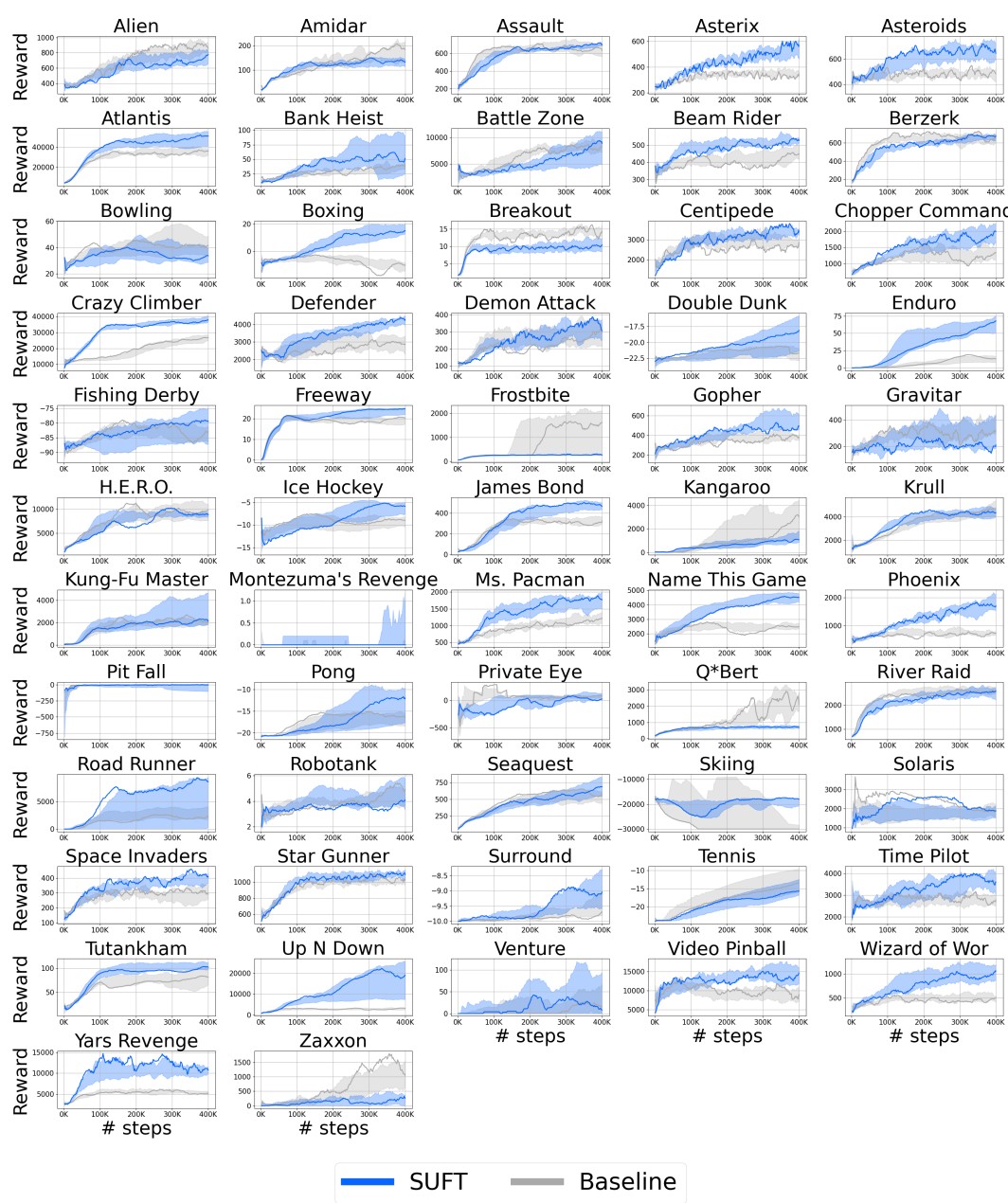

Figure 6: Learning curves comparison between the PPO agent using the additional SUFT OPE term and the baseline agent without it across 57 Atari games. The shaded area is the 10% and 90% percentiles with linear interpolation from the 10 different seeds' runs. Both agents are using $L_1$ loss.

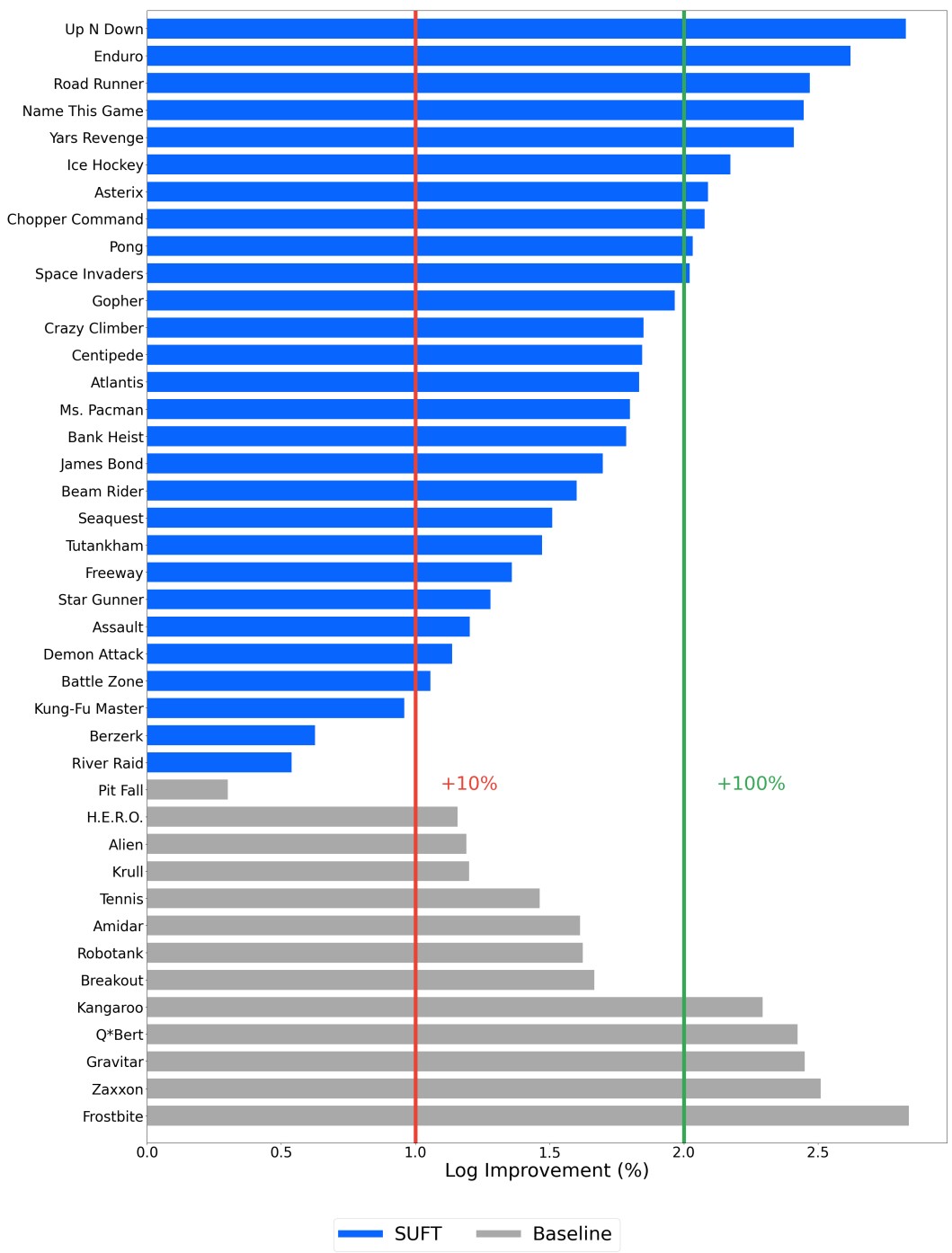

Figure 7: Log-scaled reward improvements comparison between the PPO agent using the additional SUFT OPE term and the baseline agent without it across 41 valid Atari games. SUFT outperforms the baseline agent in 28 out of the 41 valid games, achieving a reward improvement of over 100% in 24.4% of them and over 10% in 61% of them. The red line indicates a 10% improvement, and the green line represents a 100% improvement. Both agents are using $L_1$ loss.

**Atari PPO Results (Using $L_2$ Loss):**

Table 4: Training rewards comparison between the PPO agent using the additional SUFT OPE term and the baseline agent without it across 57 Atari games. SUFT surpasses the baseline agent in 50 games, with statistically significant gains in 40 games. Both agents are using $L_2$ loss.

| Game | Baseline PPO | SUFT PPO | p-value |
|---|---|---|---|
| Alien | 860.0 | **861.0** | 3.48e-01 |
| Amidar | **184.0** | **184.0** | N/A |
| Assault | 631.0 | **1150.0** | 3.24e-07 |
| Asterix | 346.0 | **651.0** | 7.43e-08 |
| Asteroids | 483.0 | **586.0** | 9.18e-05 |
| Atlantis | 36100.0 | **59100.0** | 2.57e-09 |
| Bank Heist | 37.5 | **38.8** | 3.47e-01 |
| Battle Zone | 8380.0 | **11000.0** | 5.02e-05 |
| Beam Rider | 450.0 | **497.0** | 7.57e-03 |
| Berzerk | 661.0 | **782.0** | 3.34e-05 |
| Bowling | 41.7 | **43.2** | 3.73e-01 |
| Boxing | -10.0 | **1.75** | 2.10e-09 |
| Breakout | 14.4 | **17.3** | 1.90e-04 |
| Centipede | 2860.0 | **4120.0** | 3.28e-09 |
| Chopper Command | 1350.0 | **2940.0** | 4.37e-07 |
| Crazy Climber | 26700.0 | **40200.0** | 1.99e-10 |
| Defender | 2850.0 | **4070.0** | 3.08e-07 |
| Demon Attack | 295.0 | **634.0** | 2.31e-06 |
| Double Dunk | -21.7 | **-21.5** | 8.58e-01 |
| Enduro | 13.1 | **70.1** | 1.30e-10 |
| Fishing Derby | -82.9 | **-73.9** | 4.32e-06 |
| Freeway | 20.2 | **21.2** | 1.74e-02 |
| Frostbite | 1610.0 | **2750.0** | 5.59e-02 |
| Gopher | 381.0 | **547.0** | 1.71e-06 |
| Gravitar | 318.0 | **370.0** | 3.63e-01 |
| H.E.R.O. | 9810.0 | **11700.0** | 1.32e-01 |
| Ice Hockey | -9.05 | **-7.45** | 4.09e-04 |
| James Bond | 324.0 | **380.0** | 6.50e-05 |
| Kangaroo | 3000.0 | **5830.0** | 1.04e-01 |
| Krull | 4770.0 | **8980.0** | 1.36e-03 |
| Kung-Fu Master | 1990.0 | **2610.0** | 3.76e-03 |
| Montezuma's Revenge | **0.0** | **0.0** | N/A |
| Ms. Pacman | 1210.0 | **1710.0** | 3.61e-03 |
| Name This Game | 2480.0 | **4410.0** | 7.45e-16 |
| Phoenix | 734.0 | **1720.0** | 1.05e-13 |
| Pit Fall | -1.28 | **-1.23** | 2.23e-01 |
| Pong | -16.5 | **-15.0** | 6.31e-03 |
| Private Eye | **84.3** | 76.0 | N/A |
| Q*Bert | 2080.0 | **5100.0** | 1.72e-03 |
| River Raid | 2560.0 | **3540.0** | 2.84e-11 |
| Road Runner | 2330.0 | **6320.0** | 1.51e-06 |
| Robotank | **4.88** | 4.29 | N/A |
| Seaquest | 576.0 | **1000.0** | 1.19e-07 |
| Skiing | **-30000.0** | **-30000.0** | N/A |
| Solaris | 1850.0 | **2200.0** | 3.81e-01 |
| Space Invaders | 292.0 | **387.0** | 1.31e-04 |
| Star Gunner | 1030.0 | **1130.0** | 1.60e-08 |
| Surround | -9.7 | **-8.65** | 7.33e-04 |
| Tennis | **-13.3** | -13.6 | N/A |
| Time Pilot | 2790.0 | **3190.0** | 1.24e-02 |
| Tutankham | 82.1 | **129.0** | 1.79e-02 |
| Up N Down | 3090.0 | **6020.0** | 3.83e-03 |
| Venture | **16.0** | 5.0 | N/A |
| Video Pinball | 9030.0 | **11500.0** | 3.22e-03 |
| Wizard of Wor | 508.0 | **734.0** | 6.22e-04 |
| Yars Revenge | 5230.0 | **10100.0** | 1.36e-04 |
| Zaxxon | 1040.0 | **2310.0** | 7.88e-03 |
| Mean (%) | 100.0 | **197.34** | N/A |

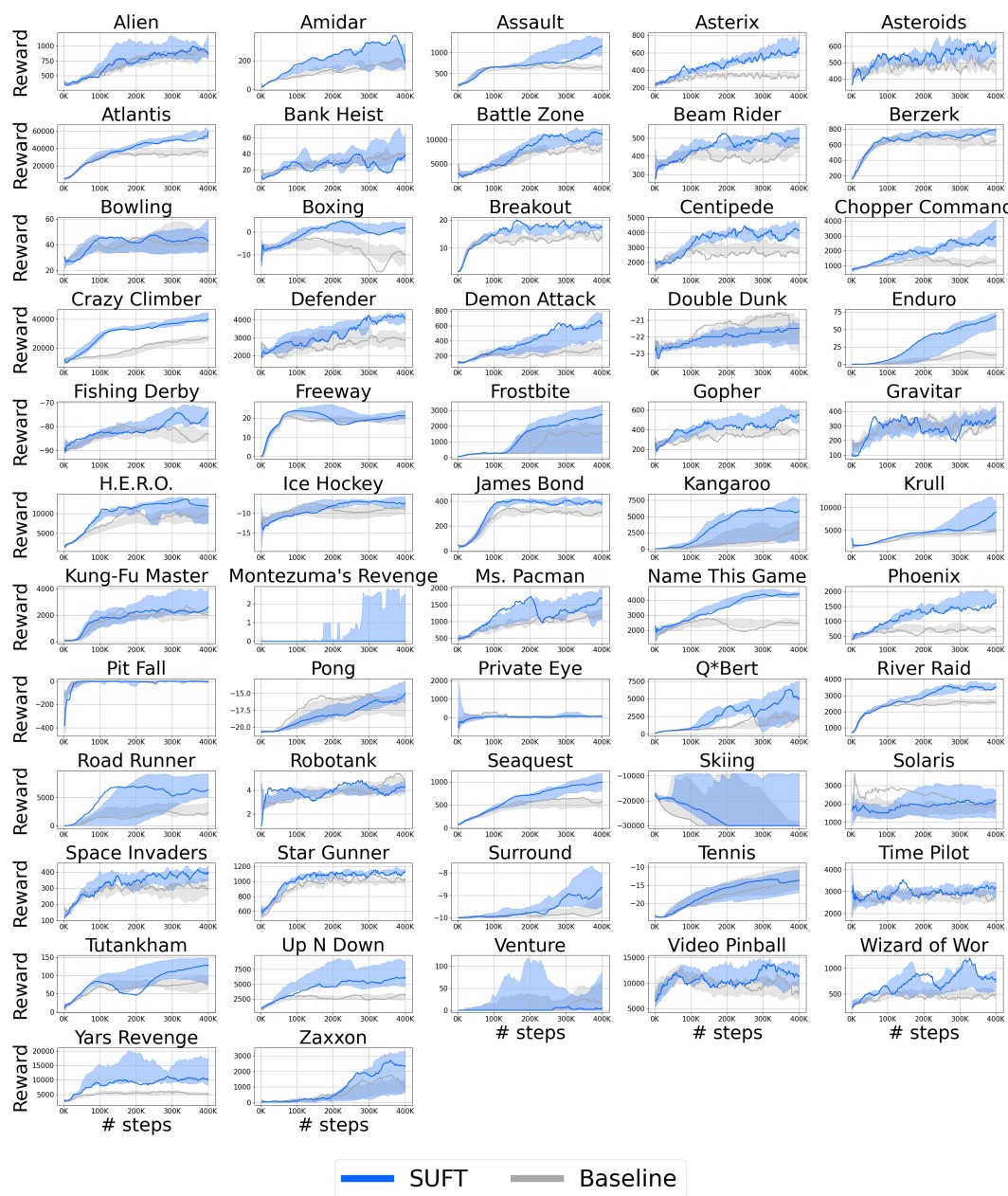

Figure 8: Learning curves comparison between the PPO agent using the additional SUFT OPE term and the baseline agent without it across 57 Atari games. The shaded area is the 10% and 90% percentiles with linear interpolation from the 10 different seeds' runs. Both agents are using $L_2$ loss.

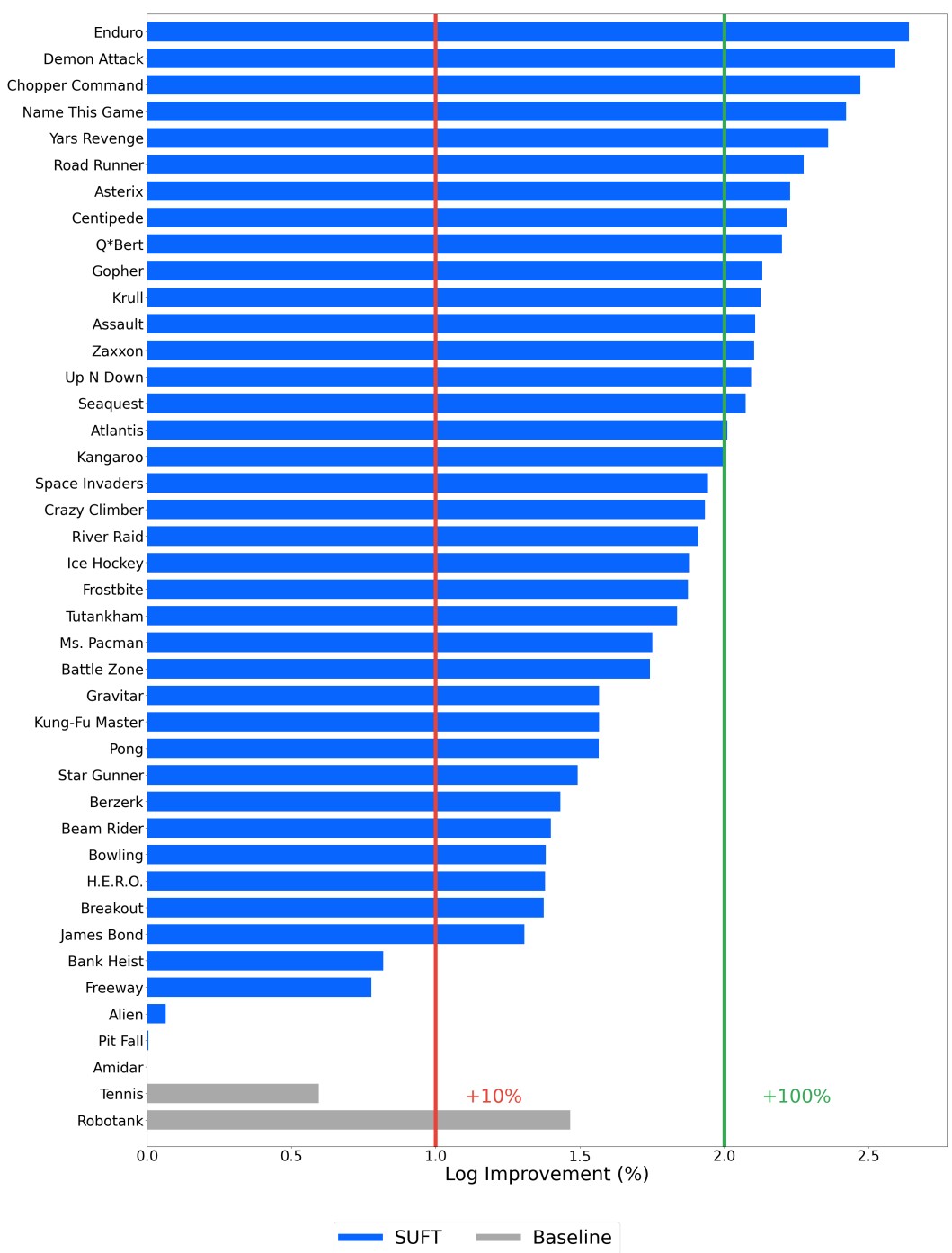

Figure 9: Log-scaled reward improvements comparison between the PPO agent using the additional SUFT OPE term and the baseline agent without it across 42 valid Atari games. SUFT outperforms the baseline agent in 39 out of the 42 valid games, achieving a reward improvement of over 100% in 38.1% of and over 10% in 83.3% of them. The red line indicates a 10% improvement, and the green line represents a 100% improvement. Both agents are using $L_2$ loss.

Table 5: Training rewards comparison between the PPO agent using the additional SUFT OPE term with different $\lambda_{TF}$ values and the baseline agent without it across 57 Atari games. All agents are using $L_2$ loss.

| Agent | $\lambda_{TF}$ | Mean (%) |
|---|---|---|
| PPO | 0.0 | 100.0% |
| SUFT PPO | 0.1 | 105.0% |
| | 0.2 | 116.05% |
| | 0.3 | 124.0% |
| | 0.4 | 126.99% |
| | 0.5 | 128.79% |
| | 0.6 | 137.06% |
| | 0.7 | 143.12% |
| | 0.8 | 146.56% |
| | 0.9 | 145.4% |
| | 1.0 | 151.72% |
| | 1.5 | 170.4% |
| | 2.0 | 167.33% |
| | 2.5 | 179.01% |
| | 3.0 | 184.66% |
| | 5.0 | 197.34% |
| | 10.0 | **198.28%** |

**Atari DQN Results:**

Table 6: Training rewards comparison between DQN agent using the additional SUFT OPE term and the baseline agent without it across 57 Atari games. SUFT surpasses the baseline agent in 52 games, with statistically significant gains in 49 games. Both agents are using $L_2$ loss.

| Game | Baseline DQN, buffer 4K | SUFT DQN, buffer 4K | p-value |
|------|------|------|------|
| Alien | 518.0 | **717.0** | 3.58e-07 |
| Amidar | 121.0 | **198.0** | 5.34e-06 |
| Assault | 1280.0 | **2250.0** | 4.10e-10 |
| Asterix | 872.0 | **1130.0** | 2.82e-03 |
| Asteroids | 294.0 | **450.0** | 2.31e-05 |
| Atlantis | 14000.0 | **19300.0** | 9.56e-05 |
| Bank Heist | 12.8 | **34.4** | 1.04e-05 |
| Battle Zone | 3350.0 | **4630.0** | 2.97e-04 |
| Beam Rider | 514.0 | **1050.0** | 6.13e-11 |
| Berzerk | 507.0 | **622.0** | 3.44e-06 |
| Bowling | 21.9 | **31.1** | 4.56e-02 |
| Boxing | -29.9 | **17.4** | 3.70e-15 |
| Breakout | 12.8 | **18.3** | 5.80e-03 |
| Centipede | 3160.0 | **3220.0** | 8.00e-01 |
| Chopper Command | 1150.0 | **1380.0** | 1.91e-03 |
| Crazy Climber | 21700.0 | **36000.0** | 3.05e-11 |
| Defender | 2510.0 | **2530.0** | 3.16e-01 |
| Demon Attack | 2240.0 | **4430.0** | 9.42e-11 |
| Double Dunk | -23.8 | **-22.9** | 1.96e-06 |
| Enduro | 48.2 | **169.0** | 3.24e-16 |
| Fishing Derby | -88.9 | **-39.8** | 2.55e-08 |
| Freeway | 19.0 | **21.2** | 1.39e-03 |
| Frostbite | 128.0 | **607.0** | 1.03e-03 |
| Gopher | 288.0 | **614.0** | 1.72e-06 |
| Gravitar | 136.0 | **284.0** | 2.10e-06 |
| H.E.R.O. | 2010.0 | **4110.0** | 1.56e-05 |
| Ice Hockey | -13.0 | **-6.62** | 1.19e-07 |
| James Bond | 34.0 | **363.0** | 1.24e-10 |
| Kangaroo | 460.0 | **5810.0** | 5.23e-07 |
| Krull | 315.0 | **3670.0** | 1.96e-14 |
| Kung-Fu Master | 6.0 | **2920.0** | 2.42e-05 |
| Montezuma's Revenge | **0.0** | **0.0** | N/A |
| Ms. Pacman | 1060.0 | **1270.0** | 6.08e-05 |
| Name This Game | 2540.0 | **4250.0** | 6.56e-11 |
| Phoenix | 608.0 | **3120.0** | 3.82e-16 |
| Pit Fall | -163.0 | **-78.1** | 2.47e-03 |
| Pong | -20.2 | **-17.9** | 2.01e-07 |
| Private Eye | 133.0 | **154.0** | 7.51e-01 |
| Q*Bert | 908.0 | **2320.0** | 5.64e-06 |
| River Raid | 2410.0 | **3160.0** | 2.25e-08 |
| Road Runner | 991.0 | **10400.0** | 1.59e-10 |
| Robotank | 4.9 | **8.71** | 2.55e-07 |
| Seaquest | 393.0 | **911.0** | 2.56e-08 |
| Skiing | -30800.0 | **-30200.0** | 8.17e-04 |
| Solaris | 565.0 | **696.0** | 2.62e-04 |
| Space Invaders | 317.0 | **371.0** | 2.53e-04 |
| Star Gunner | 519.0 | **953.0** | 7.92e-11 |
| Surround | **-9.11** | -9.17 | N/A |
| Tennis | **-18.9** | -19.6 | N/A |
| Time Pilot | 1210.0 | **3350.0** | 4.85e-14 |
| Tutankham | 11.2 | **71.4** | 7.98e-12 |
| Up N Down | 3180.0 | **5340.0** | 1.27e-03 |
| Venture | **4.0** | 0.0 | N/A |
| Video Pinball | **9360.0** | 8890.0 | N/A |
| Wizard of Wor | 235.0 | **421.0** | 4.94e-09 |
| Yars Revenge | 3060.0 | **8280.0** | 7.44e-08 |
| Zaxxon | 0.0 | **2710.0** | 1.23e-07 |
| Mean (%) | 100.0 | **383.19** | N/A |

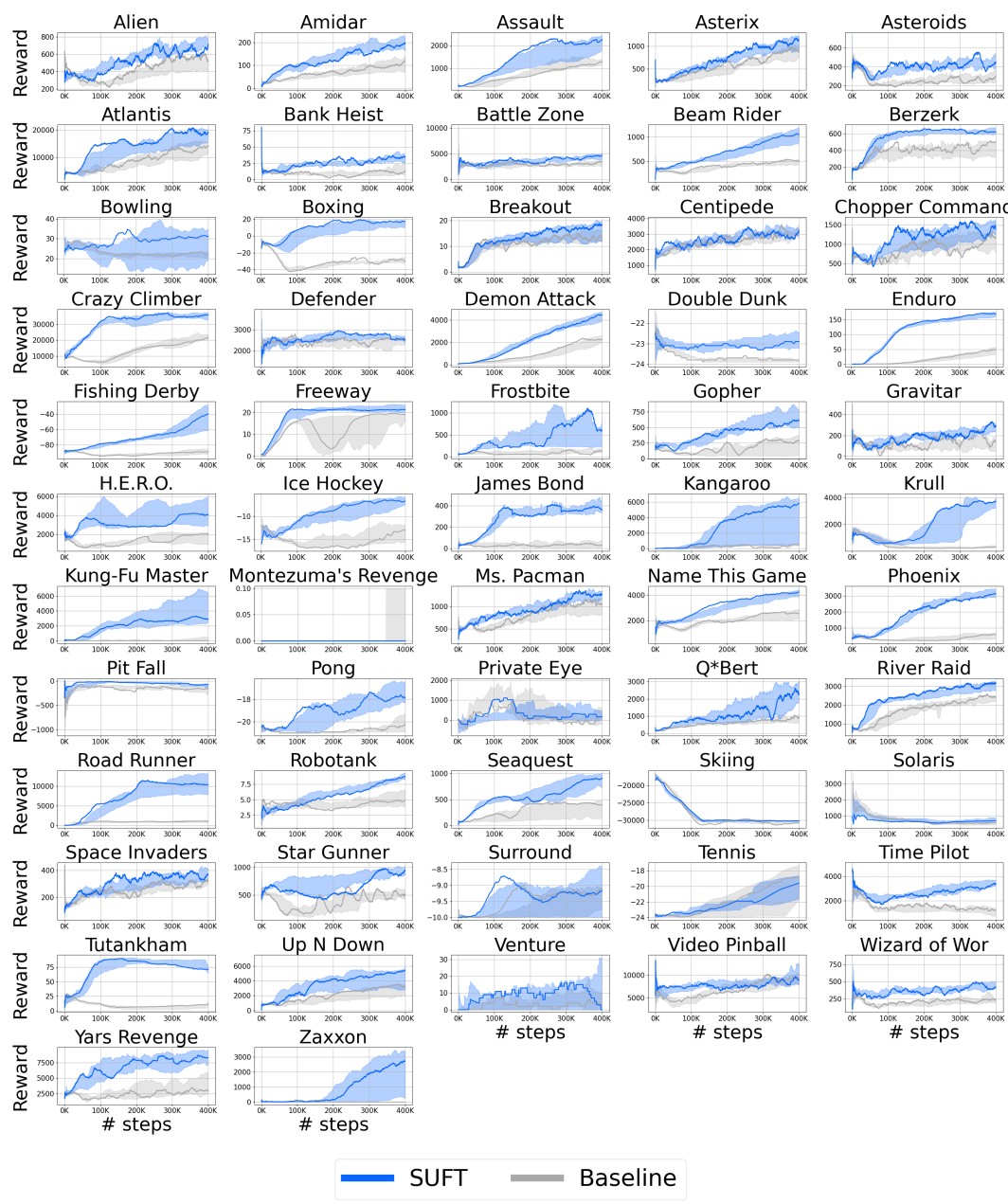

Figure 10: Learning curves comparison between DQN agent using the additional SUFT OPE term and the baseline agent without it across 57 Atari games. The shaded area is the 10% and 90% percentiles with linear interpolation from the 10 different seeds' runs. Both agents are using $L_2$ loss.

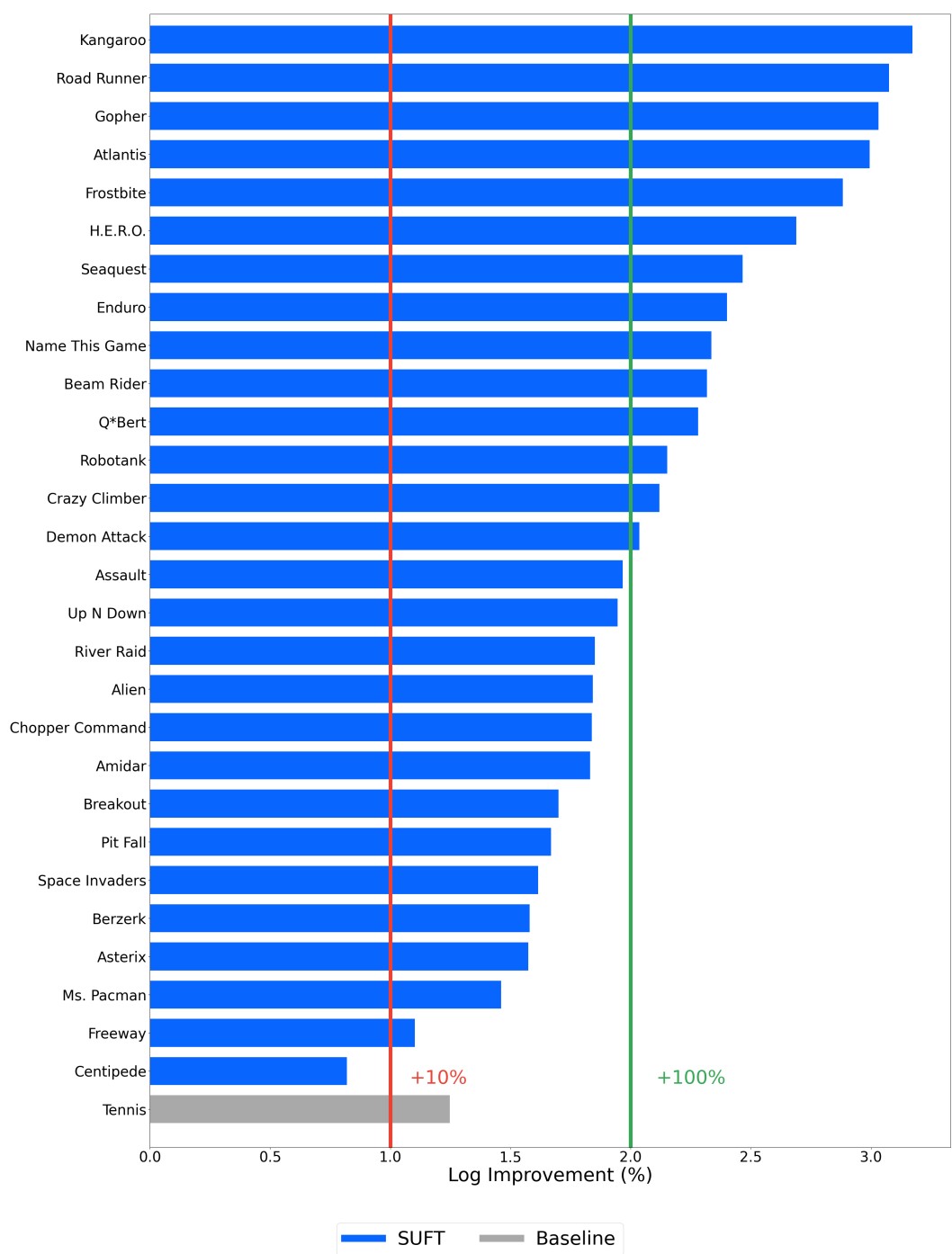

Figure 11: Log-scaled reward improvements comparison between DQN agent using the additional SUFT OPE term and the baseline agent without it across 29 valid Atari games. SUFT outperforms the baseline agent in 28 out of the 29 valid games, achieving a reward improvement of over 100% in 48.3% of them and over 10% in 93.1% of them. The red line indicates a 10% improvement, and the green line represents a 100% improvement. Both agents are using $L_2$ loss.

Table 7: Training rewards comparison between DQN agent using the additional SUFT OPE term with a small 4K buffer and the baseline agent without the SUFT OPE term with a larger 100K buffer across 57 Atari games. SUFT surpasses the baseline agent in 52 games, with statistically significant gains in 47 games. Both agents are using $L_2$ loss.

| Game | Baseline DQN, buffer 100K | SUFT DQN, buffer 4K | p-value |
|------|---------------------------|---------------------|---------|
| Alien | 438.0 | **717.0** | 3.14e-09 |
| Amidar | 118.0 | **198.0** | 2.20e-07 |
| Assault | 1120.0 | **2250.0** | 6.54e-10 |
| Asterix | 914.0 | **1130.0** | 4.06e-03 |
| Asteroids | 275.0 | **450.0** | 4.69e-06 |
| Atlantis | 7970.0 | **19300.0** | 7.59e-09 |
| Bank Heist | 9.9 | **34.4** | 1.01e-11 |
| Battle Zone | 3100.0 | **4630.0** | 9.71e-04 |
| Beam Rider | 561.0 | **1050.0** | 7.44e-09 |
| Berzerk | 458.0 | **622.0** | 1.55e-05 |
| Bowling | 21.8 | **31.1** | 4.58e-02 |
| Boxing | -28.7 | **17.4** | 2.25e-15 |
| Breakout | 17.7 | **18.3** | 6.94e-01 |
| Centipede | 3130.0 | **3220.0** | 7.99e-01 |
| Chopper Command | 648.0 | **1380.0** | 1.30e-09 |
| Crazy Climber | 30400.0 | **36000.0** | 1.38e-04 |
| Defender | 2440.0 | **2530.0** | 4.85e-02 |
| Demon Attack | 3220.0 | **4430.0** | 8.29e-08 |
| Double Dunk | -23.9 | **-22.9** | 8.65e-07 |
| Enduro | 66.8 | **169.0** | 5.22e-16 |
| Fishing Derby | -84.1 | **-39.8** | 4.29e-07 |
| Freeway | 18.9 | **21.2** | 2.32e-03 |
| Frostbite | 159.0 | **607.0** | 1.95e-03 |
| Gopher | 119.0 | **614.0** | 3.64e-09 |
| Gravitar | 107.0 | **284.0** | 4.17e-09 |
| H.E.R.O. | 2320.0 | **4110.0** | 8.37e-05 |
| Ice Hockey | -13.2 | **-6.62** | 2.20e-08 |
| James Bond | 43.5 | **363.0** | 1.41e-10 |
| Kangaroo | 534.0 | **5810.0** | 5.25e-07 |
| Krull | 336.0 | **3670.0** | 2.71e-14 |
| Kung-Fu Master | 13.0 | **2920.0** | 2.17e-05 |
| Montezuma's Revenge | **0.0** | **0.0** | N/A |
| Ms. Pacman | 1060.0 | **1270.0** | 3.69e-03 |
| Name This Game | 2990.0 | **4250.0** | 3.66e-10 |
| Phoenix | 732.0 | **3120.0** | 1.12e-15 |
| Pit Fall | -114.0 | **-78.1** | 5.41e-01 |
| Pong | -20.6 | **-17.9** | 7.89e-09 |
| Private Eye | 110.0 | **154.0** | 6.69e-01 |
| Q*Bert | 1130.0 | **2320.0** | 3.81e-05 |
| River Raid | 2200.0 | **3160.0** | 3.60e-10 |
| Road Runner | 909.0 | **10400.0** | 1.04e-10 |
| Robotank | 4.44 | **8.71** | 4.26e-05 |
| Seaquest | 391.0 | **911.0** | 1.08e-08 |
| Skiing | -30700.0 | **-30200.0** | 8.64e-03 |
| Solaris | 515.0 | **696.0** | 2.80e-05 |
| Space Invaders | 366.0 | **371.0** | 3.07e-01 |
| Star Gunner | 351.0 | **953.0** | 1.21e-12 |
| Surround | **-8.69** | -9.17 | N/A |
| Tennis | **-17.5** | -19.6 | N/A |
| Time Pilot | 1100.0 | **3350.0** | 1.47e-14 |
| Tutankham | 9.4 | **71.4** | 4.66e-18 |
| Up N Down | 2160.0 | **5340.0** | 6.09e-05 |
| Venture | **4.0** | 0.0 | N/A |
| Video Pinball | **9890.0** | 8890.0 | N/A |
| Wizard of Wor | 292.0 | **421.0** | 2.39e-07 |
| Yars Revenge | 3530.0 | **8280.0** | 9.75e-08 |
| Zaxxon | 0.0 | **2710.0** | 1.19e-07 |
| Mean (%) | 100.0 | **437.05** | N/A |

Table 8: Training rewards comparison between DQN agent using the additional SUFT OPE term and the baseline agent without it across 57 Atari games. SUFT surpasses the baseline agent in 49 games, with statistically significant gains in 44 games. Both agents are using a 100k buffer and $L_2$ loss.

| Game | Baseline DQN, buffer 100K | SUFT DQN, buffer 100K | p-value |
|---|---|---|---|
| Alien | 438.0 | **686.0** | 1.45e-08 |
| Amidar | 118.0 | **235.0** | 6.49e-11 |
| Assault | 1120.0 | **2170.0** | 3.18e-10 |
| Asterix | 914.0 | **1240.0** | 3.19e-05 |
| Asteroids | 275.0 | **430.0** | 1.03e-05 |
| Atlantis | 7970.0 | **12400.0** | 5.88e-05 |
| Bank Heist | 9.9 | **27.7** | 8.87e-04 |
| Battle Zone | 3100.0 | **4270.0** | 5.45e-05 |
| Beam Rider | 561.0 | **1010.0** | 1.61e-09 |
| Berzerk | 458.0 | **625.0** | 2.38e-05 |
| Bowling | **21.8** | 21.5 | N/A |
| Boxing | -28.7 | **-1.34** | 1.10e-10 |
| Breakout | 17.7 | **18.1** | 5.37e-01 |
| Centipede | 3130.0 | **4820.0** | 1.47e-02 |
| Chopper Command | 648.0 | **957.0** | 9.89e-04 |
| Crazy Climber | 30400.0 | **35500.0** | 6.89e-05 |
| Defender | 2440.0 | **2590.0** | 3.53e-02 |
| Demon Attack | 3220.0 | **5220.0** | 1.85e-10 |
| Double Dunk | -23.9 | **-22.5** | 1.34e-11 |
| Enduro | 66.8 | **153.0** | 2.45e-15 |
| Fishing Derby | -84.1 | **-43.9** | 1.87e-08 |
| Freeway | 18.9 | **21.1** | 4.23e-03 |
| Frostbite | **159.0** | 33.3 | N/A |
| Gopher | 119.0 | **666.0** | 9.22e-07 |
| Gravitar | 107.0 | **286.0** | 2.30e-04 |
| H.E.R.O. | **2320.0** | 431.0 | N/A |
| Ice Hockey | -13.2 | **-5.29** | 6.87e-10 |
| James Bond | 43.5 | **372.0** | 2.39e-06 |
| Kangaroo | 534.0 | **5670.0** | 4.11e-14 |
| Krull | 336.0 | **1780.0** | 2.90e-05 |
| Kung-Fu Master | 13.0 | **3190.0** | 5.18e-09 |
| Montezuma's Revenge | **0.0** | **0.0** | N/A |
| Ms. Pacman | 1060.0 | **1350.0** | 9.98e-04 |
| Name This Game | 2990.0 | **3680.0** | 9.95e-06 |
| Phoenix | 732.0 | **2700.0** | 4.01e-15 |
| Pit Fall | **-114.0** | -355.0 | N/A |
| Pong | -20.6 | **-18.3** | 4.76e-06 |
| Private Eye | **110.0** | 82.4 | N/A |
| Q*Bert | 1130.0 | **2850.0** | 1.01e-05 |
| River Raid | 2200.0 | **2950.0** | 4.95e-11 |
| Road Runner | 909.0 | **9240.0** | 1.54e-13 |
| Robotank | 4.44 | **8.95** | 3.82e-05 |
| Seaquest | 391.0 | **574.0** | 5.20e-04 |
| Skiing | -30700.0 | **-30200.0** | 1.39e-02 |
| Solaris | 515.0 | **653.0** | 1.07e-03 |
| Space Invaders | 366.0 | **421.0** | 4.64e-06 |
| Star Gunner | 351.0 | **387.0** | 3.01e-01 |
| Surround | -8.69 | **-8.55** | 4.83e-01 |
| Tennis | **-17.5** | -18.5 | N/A |
| Time Pilot | 1100.0 | **3570.0** | 1.43e-15 |
| Tutankham | 9.4 | **15.0** | 2.77e-01 |
| Up N Down | 2160.0 | **3020.0** | 2.25e-03 |
| Venture | 4.0 | **10.0** | 1.46e-02 |
| Video Pinball | **9890.0** | 8970.0 | N/A |
| Wizard of Wor | 292.0 | **423.0** | 4.12e-03 |
| Yars Revenge | 3530.0 | **6900.0** | 7.48e-02 |
| Zaxxon | 0.0 | **1020.0** | 1.08e-02 |
| Mean (%) | 100.0 | **413.64** | N/A |

**Atari Vanilla DQN Results:**

Table 9: Training rewards comparison between Vanilla DQN agent using the additional SUFT OPE term and the baseline agent without it across 57 Atari games. SUFT surpasses the baseline agent in 49 games, with statistically significant gains in 33 games. Both agents are using $L_2$ loss.

| Game | Baseline Vanilla DQN, buffer 4K | SUFT Vanilla DQN, buffer 4K | p-value |
|---|---|---|---|
| Alien | 657.0 | **736.0** | 1.37e-02 |
| Amidar | 163.0 | **201.0** | 1.95e-04 |
| Assault | 904.0 | **1600.0** | 7.33e-11 |
| Asterix | 680.0 | **1030.0** | 8.30e-09 |
| Asteroids | 408.0 | **423.0** | 5.36e-01 |
| Atlantis | 10300.0 | **19000.0** | 6.54e-08 |
| Bank Heist | 29.4 | **35.2** | 8.01e-02 |
| Battle Zone | 11600.0 | **12700.0** | 1.10e-01 |
| Beam Rider | 909.0 | **1440.0** | 8.25e-08 |
| Berzerk | 613.0 | **653.0** | 2.35e-03 |
| Bowling | **26.5** | 25.1 | N/A |
| Boxing | 0.31 | **17.8** | 1.10e-10 |
| Breakout | 16.0 | **18.0** | 5.85e-01 |
| Centipede | 3430.0 | **3680.0** | 2.13e-02 |
| Chopper Command | 1390.0 | **1700.0** | 1.84e-04 |
| Crazy Climber | 38700.0 | **43500.0** | 1.76e-05 |
| Defender | 3290.0 | **4000.0** | 5.24e-04 |
| Demon Attack | 1490.0 | **2790.0** | 6.15e-12 |
| Double Dunk | **-22.9** | -23.0 | N/A |
| Enduro | 156.0 | **207.0** | 3.40e-08 |
| Fishing Derby | -76.0 | **-60.8** | 4.17e-05 |
| Freeway | 24.9 | **25.2** | 9.93e-01 |
| Frostbite | 272.0 | **348.0** | 1.32e-01 |
| Gopher | 362.0 | **654.0** | 8.02e-06 |
| Gravitar | 221.0 | **224.0** | 5.56e-01 |
| H.E.R.O. | 4590.0 | **8700.0** | 1.55e-02 |
| Ice Hockey | -10.8 | **-9.41** | 2.94e-02 |
| James Bond | 402.0 | **428.0** | 1.80e-01 |
| Kangaroo | 1780.0 | **4360.0** | 1.95e-02 |
| Krull | 3650.0 | **4440.0** | 8.03e-05 |
| Kung-Fu Master | **2880.0** | 2660.0 | N/A |
| Montezuma's Revenge | **0.0** | **0.0** | N/A |
| Ms. Pacman | **1560.0** | 1520.0 | N/A |
| Name This Game | 3060.0 | **4170.0** | 2.79e-09 |
| Phoenix | 2110.0 | **2720.0** | 7.66e-07 |
| Pit Fall | **-35.2** | -42.2 | N/A |
| Pong | **-15.6** | -16.9 | N/A |
| Private Eye | 66.4 | **72.6** | 5.77e-01 |
| Q*Bert | 1510.0 | **2100.0** | 1.69e-03 |
| River Raid | 2970.0 | **3420.0** | 4.87e-06 |
| Road Runner | 16900.0 | **17700.0** | 5.00e-01 |
| Robotank | 5.62 | **7.02** | 1.02e-04 |
| Seaquest | 751.0 | **1120.0** | 1.08e-07 |
| Skiing | -30300.0 | **-30200.0** | 7.50e-02 |
| Solaris | **915.0** | 683.0 | N/A |
| Space Invaders | 374.0 | **424.0** | 5.24e-04 |
| Star Gunner | 829.0 | **978.0** | 1.53e-05 |
| Surround | -9.42 | **-9.33** | 5.56e-01 |
| Tennis | -19.5 | **-18.0** | 2.67e-02 |
| Time Pilot | 2880.0 | **2910.0** | 5.70e-01 |
| Tutankham | 53.5 | **106.0** | 1.03e-07 |
| Up N Down | 4640.0 | **6970.0** | 8.16e-05 |
| Venture | 9.0 | **18.0** | 1.29e-01 |
| Video Pinball | 13700.0 | **14200.0** | 2.84e-01 |
| Wizard of Wor | 365.0 | **424.0** | 1.79e-03 |
| Yars Revenge | 7260.0 | **10100.0** | 1.42e-07 |
| Zaxxon | 112.0 | **932.0** | 1.96e-01 |
| Mean (%) | 100.0 | **228.12** | N/A |

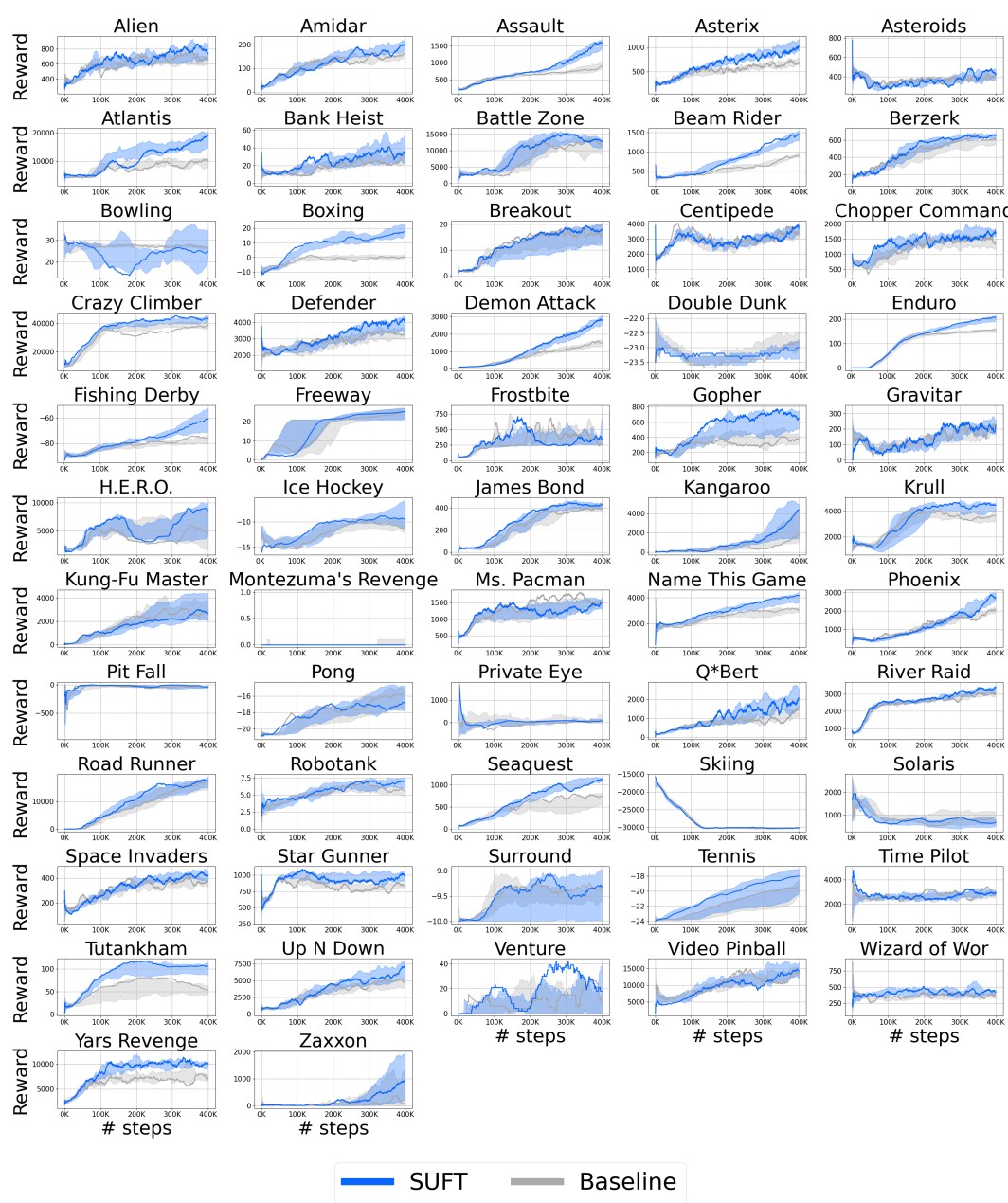

Figure 12: Learning curves comparison between Vanilla DQN agent using the additional SUFT OPE term and the baseline agent without it across 57 Atari games. The shaded area is the 10% and 90% percentiles with linear interpolation from the 10 different seeds' runs. Both agents are using $L_2$ loss.

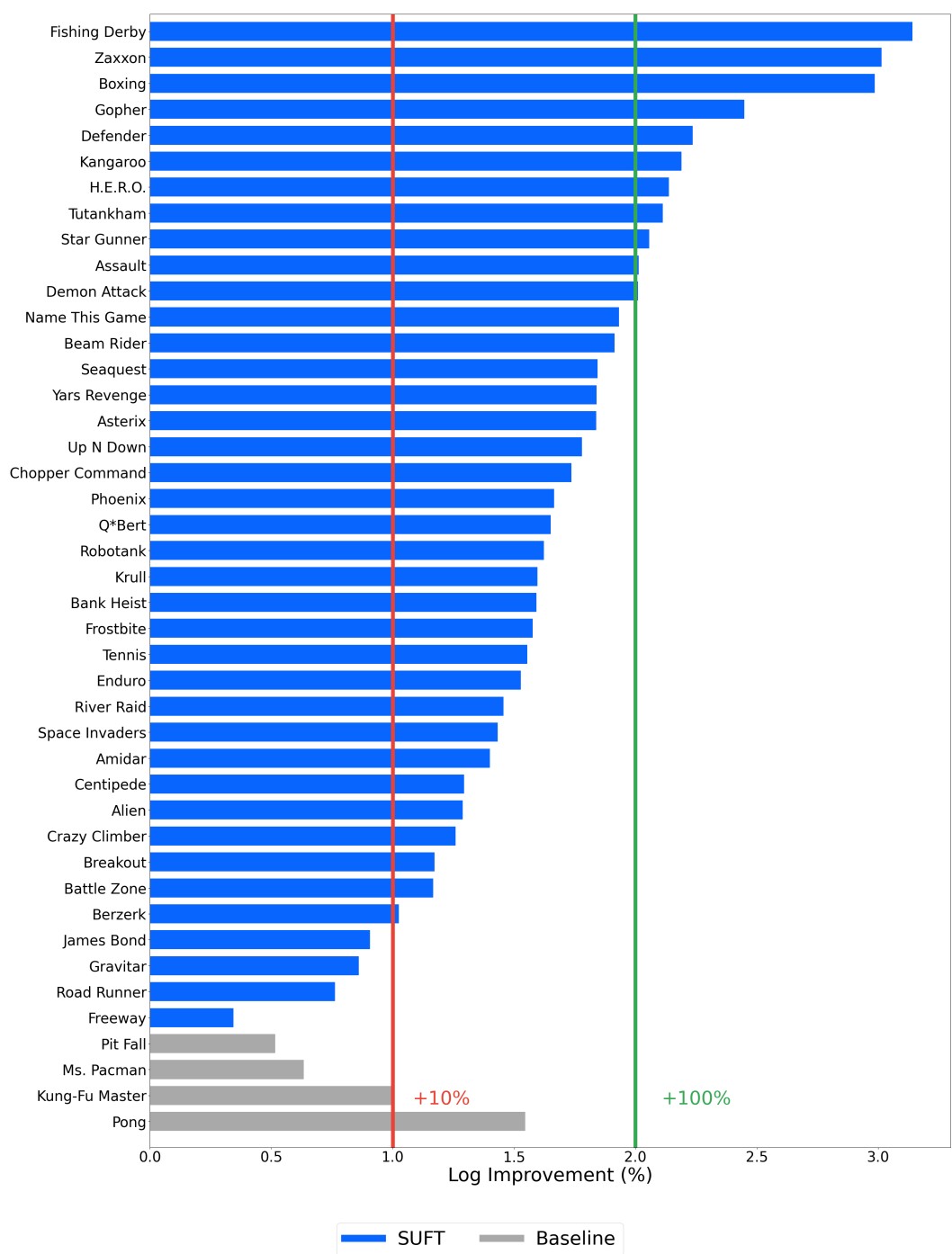

Figure 13: Log-scaled reward improvements comparison between Vanilla DQN agent using the additional SUFT OPE term and the baseline agent without it across 43 valid Atari games. SUFT outperforms the baseline agent in 39 out of the 43 valid games, achieving a reward improvement of over 100% in 25.6% of them and over 10% in 79.1% of them. The red line indicates a 10% improvement, and the green line represents a 100% improvement. Both agents are using $L_2$ loss.

**Atari Double DQN Results:**

Table 10: Training rewards comparison between the Double DQN agent using the additional SUFT OPE term and the baseline agent without it across 57 Atari games. SUFT surpasses the baseline agent in 48 games, with statistically significant gains in 37 games. Both agents are using $L_2$ loss.

| Game | Baseline DDQN, buffer 4K | SUFT DDQN, buffer 4K | p-value |
|------|--------------------------|----------------------|---------|
| Alien | 773.0 | **806.0** | 7.88e-01 |
| Amidar | 152.0 | **177.0** | 8.27e-03 |
| Assault | 781.0 | **1440.0** | 3.84e-12 |
| Asterix | 676.0 | **939.0** | 1.03e-08 |
| Asteroids | 357.0 | **365.0** | 1.30e-01 |
| Atlantis | 13900.0 | **18100.0** | 1.14e-03 |
| Bank Heist | 29.2 | **36.1** | 1.30e-03 |
| Battle Zone | 10900.0 | **14000.0** | 1.16e-04 |
| Beam Rider | 719.0 | **1230.0** | 1.03e-12 |
| Berzerk | 567.0 | **668.0** | 5.01e-08 |
| Bowling | **27.8** | 27.0 | N/A |
| Boxing | -1.73 | **18.4** | 8.10e-11 |
| Breakout | **19.3** | 16.3 | N/A |
| Centipede | 3390.0 | **3430.0** | 6.14e-01 |
| Chopper Command | 1460.0 | **1870.0** | 4.47e-05 |
| Crazy Climber | 41200.0 | **42200.0** | 3.93e-01 |
| Defender | 2980.0 | **4140.0** | 8.61e-10 |
| Demon Attack | 369.0 | **2240.0** | 3.09e-15 |
| Double Dunk | -23.6 | **-23.4** | 2.41e-02 |
| Enduro | 143.0 | **195.0** | 3.43e-13 |
| Fishing Derby | -81.4 | **-67.8** | 4.39e-07 |
| Freeway | **25.2** | 21.3 | N/A |
| Frostbite | **344.0** | 265.0 | N/A |
| Gopher | 595.0 | **658.0** | 4.17e-01 |
| Gravitar | 100.0 | **134.0** | 2.00e-02 |
| H.E.R.O. | 4020.0 | **8870.0** | 2.06e-03 |
| Ice Hockey | **-7.7** | -8.15 | N/A |
| James Bond | 316.0 | **416.0** | 7.84e-04 |
| Kangaroo | 2430.0 | **3280.0** | 1.94e-01 |
| Krull | 3830.0 | **4400.0** | 9.09e-05 |
| Kung-Fu Master | 2470.0 | **3680.0** | 3.30e-02 |
| Montezuma's Revenge | **0.0** | **0.0** | N/A |
| Ms. Pacman | 1560.0 | **1650.0** | 3.26e-02 |
| Name This Game | 3140.0 | **4140.0** | 7.71e-08 |
| Phoenix | 1080.0 | **2220.0** | 1.05e-10 |
| Pit Fall | **-14.2** | -27.5 | N/A |
| Pong | -15.9 | **-15.8** | 6.54e-01 |
| Private Eye | -45.2 | **49.4** | 8.98e-01 |
| Q*Bert | 1260.0 | **1960.0** | 6.31e-05 |
| River Raid | 2960.0 | **3420.0** | 1.56e-07 |
| Road Runner | 15700.0 | **17000.0** | 4.29e-01 |
| Robotank | 6.43 | **7.47** | 6.65e-03 |
| Seaquest | 474.0 | **938.0** | 9.91e-12 |
| Skiing | -30300.0 | **-30200.0** | 2.87e-01 |
| Solaris | **1620.0** | 599.0 | N/A |
| Space Invaders | 377.0 | **446.0** | 3.52e-05 |
| Star Gunner | 970.0 | **1100.0** | 5.04e-06 |
| Surround | **-9.62** | -9.64 | N/A |
| Tennis | -23.6 | **-20.3** | 7.08e-05 |
| Time Pilot | 2130.0 | **2640.0** | 3.75e-05 |
| Tutankham | 64.1 | **107.0** | 1.56e-05 |
| Up N Down | 3940.0 | **6950.0** | 1.23e-04 |
| Venture | 6.0 | **18.0** | 4.54e-02 |
| Video Pinball | 13400.0 | **15400.0** | 1.23e-02 |
| Wizard of Wor | 349.0 | **447.0** | 3.76e-05 |
| Yars Revenge | 8610.0 | **10400.0** | 2.31e-06 |
| Zaxxon | 4.0 | **10.0** | 1.71e-01 |
| Mean (%) | 100.0 | **226.91** | N/A |

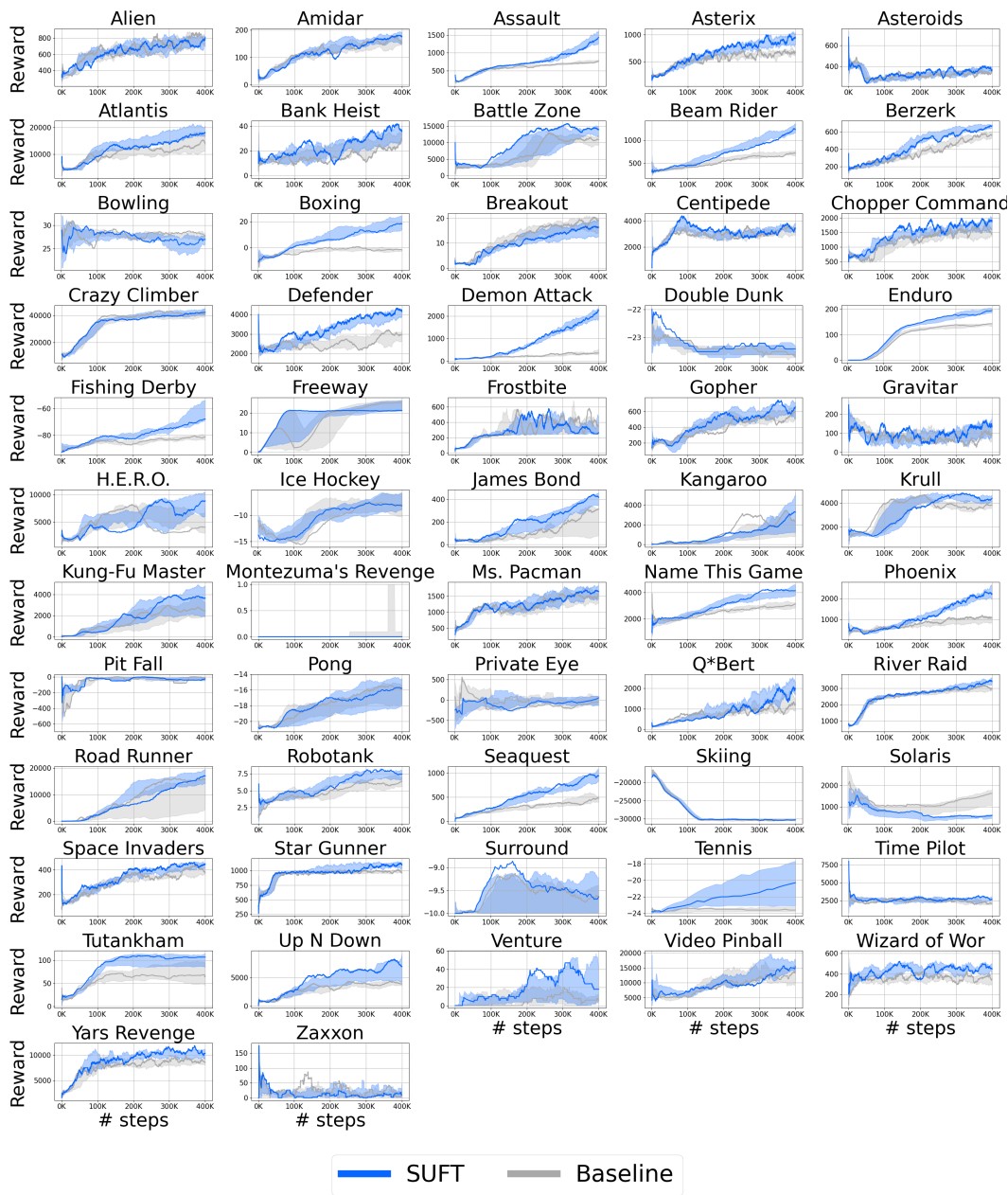

Figure 14: Learning curves comparison between the Double DQN agent using the additional SUFT OPE term and the baseline agent without it across 57 Atari games. The shaded area is the 10% and 90% percentiles with linear interpolation from the 10 different seeds' runs. Both agents are using $L_2$ loss.

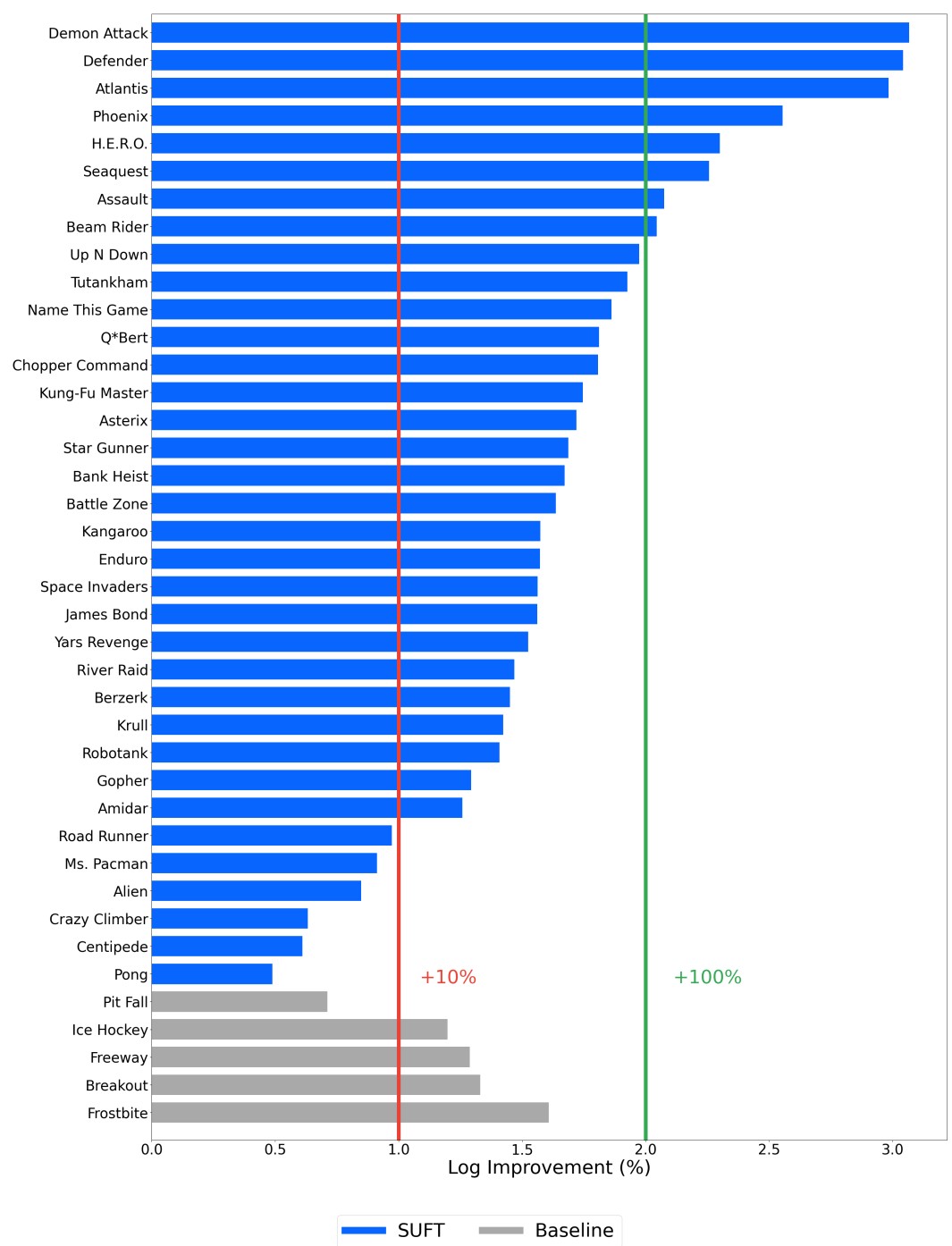

Figure 15: Log-scaled reward improvements comparison between the Double DQN agent using the additional SUFT OPE term and the baseline agent without it across 40 valid Atari games. SUFT outperforms the baseline agent in 35 out of the 40 valid games, achieving a reward improvement of over 100% in 20% of them and over 10% in 72.5% of them. The red line indicates a 10% improvement, and the green line represents a 100% improvement. Both agents are using $L_2$ loss.

Table 11: Training rewards comparison between our causal bound optimization method and an inertia regularizer. We compare a Double DQN agent with the additional SUFT OPE term to one using an inertia regularizer across 57 Atari games. While SUFT significantly improves the baseline agent's performance with 226.91% mean reward ratio, the inertia regularizer harms the baseline agent's performance with only 83.21% mean reward ratio. All agents are using $L_2$ loss and a 4K buffer size.

| Game | Baseline DDQN | Inertia Regularizer DDQN | SUFT DDQN |
|---|---|---|---|
| Alien | 773.0 | 762.0 | **806.0** |
| Amidar | 152.0 | 144.0 | **177.0** |
| Assault | 781.0 | 670.0 | **1440.0** |
| Asterix | 676.0 | 666.0 | **939.0** |
| Asteroids | 357.0 | 278.0 | **365.0** |
| Atlantis | 13900.0 | 3970.0 | **18100.0** |
| Bank Heist | 29.2 | 21.8 | **36.1** |
| Battle Zone | 10900.0 | 11200.0 | **14000.0** |
| Beam Rider | 719.0 | 707.0 | **1230.0** |
| Berzerk | 567.0 | 628.0 | **668.0** |
| Bowling | 27.8 | **28.8** | 27.0 |
| Boxing | -1.73 | -3.91 | **18.4** |
| Breakout | **19.3** | 8.85 | 16.3 |
| Centipede | 3390.0 | **3840.0** | 3430.0 |
| Chopper Command | 1460.0 | 1120.0 | **1870.0** |
| Crazy Climber | 41200.0 | **42800.0** | 42200.0 |
| Defender | 2980.0 | 2650.0 | **4140.0** |
| Demon Attack | 369.0 | 191.0 | **2240.0** |
| Double Dunk | -23.6 | -23.6 | **-23.4** |
| Enduro | 143.0 | 107.0 | **195.0** |
| Fishing Derby | -81.4 | -84.9 | **-67.8** |
| Freeway | **25.2** | 21.2 | 21.3 |
| Frostbite | **344.0** | 273.0 | 265.0 |
| Gopher | 595.0 | 376.0 | **658.0** |
| Gravitar | 100.0 | **138.0** | 134.0 |
| H.E.R.O. | 4020.0 | 5060.0 | **8870.0** |
| Ice Hockey | -7.7 | **-6.6** | -8.15 |
| James Bond | 316.0 | 186.0 | **416.0** |
| Kangaroo | 2430.0 | 1110.0 | **3280.0** |
| Krull | 3830.0 | 2700.0 | **4400.0** |
| Kung-Fu Master | 2470.0 | 1520.0 | **3680.0** |
| Montezuma's Revenge | **0.0** | **0.0** | 0.0 |
| Ms. Pacman | 1560.0 | 1510.0 | **1650.0** |
| Name This Game | 3140.0 | 2420.0 | **4140.0** |
| Phoenix | 1080.0 | 640.0 | **2220.0** |
| Pit Fall | **-14.2** | -25.5 | -27.5 |
| Pong | -15.9 | -17.3 | **-15.8** |
| Private Eye | -45.2 | -11.9 | **49.4** |
| Q*Bert | 1260.0 | 1180.0 | **1960.0** |
| River Raid | 2960.0 | 2550.0 | **3420.0** |
| Road Runner | 15700.0 | 11000.0 | **17000.0** |
| Robotank | 6.43 | 6.51 | **7.47** |
| Seaquest | 474.0 | 306.0 | **938.0** |
| Skiing | -30300.0 | **-26500.0** | -30200.0 |
| Solaris | 1620.0 | **1860.0** | 599.0 |
| Space Invaders | 377.0 | 400.0 | **446.0** |
| Star Gunner | 970.0 | 980.0 | **1100.0** |
| Surround | -9.62 | **-9.37** | -9.64 |
| Tennis | -23.6 | **-15.1** | -20.3 |
| Time Pilot | 2130.0 | 1920.0 | **2640.0** |
| Tutankham | 64.1 | 75.8 | **107.0** |
| Up N Down | 3940.0 | 2930.0 | **6950.0** |
| Venture | 6.0 | 4.0 | **18.0** |
| Video Pinball | 13400.0 | 12400.0 | **15400.0** |
| Wizard of Wor | 349.0 | 215.0 | **447.0** |
| Yars Revenge | 8610.0 | 7990.0 | **10400.0** |
| Zaxxon | 4.0 | **12.0** | 10.0 |
| Mean (%) | 100.0 | 83.21 | **226.91** |

## D.7 MuJoCo Five Environments Benchmark

**MuJoCo PPO Results:**

Table 12: Training rewards comparison between the PPO agent using the additional SUFT OPE term and the baseline agent without it across 5 MuJoCo environments. SUFT surpasses the baseline agent in 4 out of 5 environments, with statistically significant gains in 2 environments. Both agents are using $L_2$ loss.

| Game | Baseline PPO | SUFT PPO | p-value |
|---|---|---|---|
| Ant | 1110.0 | **1570.0** | 1.65e-02 |
| Half Cheetah | 1520.0 | **1540.0** | 7.28e-01 |
| Hopper | 1880.0 | **2120.0** | 3.16e-03 |
| Humanoid | **464.0** | **464.0** | N/A |
| Walker2d | 1850.0 | **1860.0** | 8.33e-01 |
| Mean (%) | 100.0 | **111.21** | N/A |

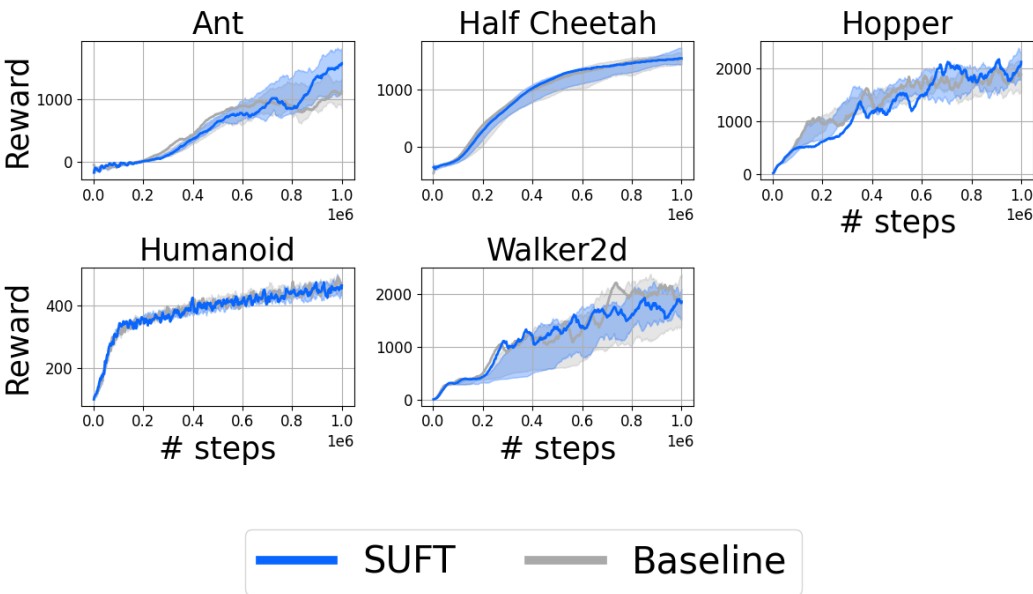

Figure 16: Learning curves comparison between the PPO agent using the additional SUFT OPE term and the baseline agent without it across 5 MuJoCo environments. The shaded area is the 10% and 90% percentiles with linear interpolation from the 10 different seeds' runs. Both agents are using $L_2$ loss.

**MuJoCo SAC Results:**

Table 13: Training rewards comparison between the SAC agent using the additional SUFT OPE term and the baseline agent without it across 5 MuJoCo environments. SUFT surpasses the baseline agent in 4 out of 5 environments, with statistically significant gains in 3 environments. Both agents are using $L_2$ loss.

| Game | Baseline SAC, buffer 4K | SUFT SAC, buffer 4K | p-value |
|------|------|------|------|
| Ant | **1250.0** | 1020.0 | N/A |
| Half Cheetah | 4440.0 | **6290.0** | 3.94e-01 |
| Hopper | 978.0 | **1910.0** | 1.46e-06 |
| Humanoid | 638.0 | **1100.0** | 2.16e-02 |
| Walker2d | 316.0 | **1680.0** | 8.16e-11 |
| Mean (%) | 100.0 | **224.53** | N/A |

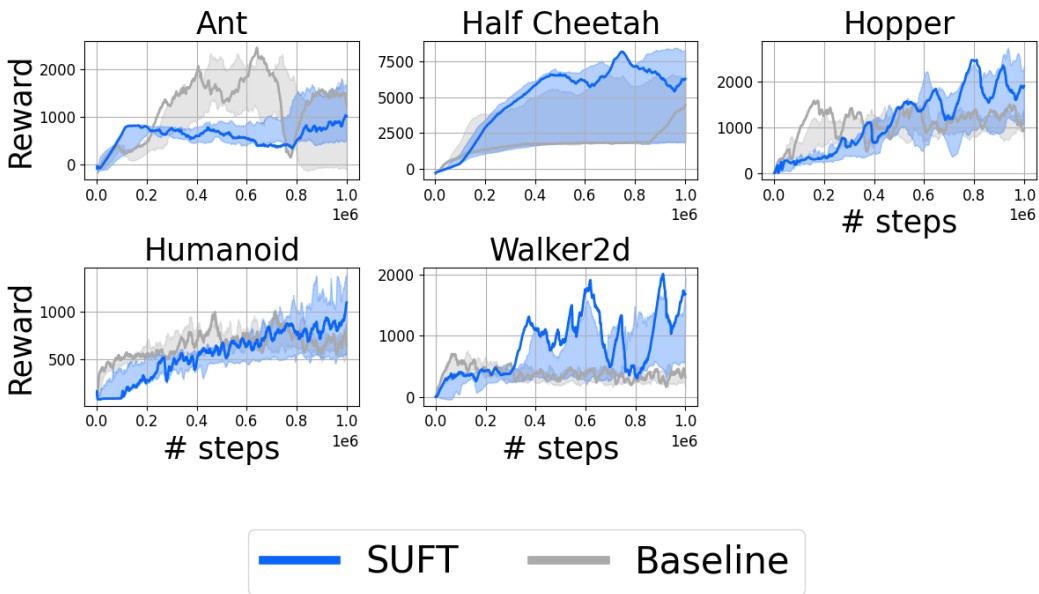

Figure 17: Learning curves comparison between the SAC agent using the additional SUFT OPE term and the baseline agent without it across 5 MuJoCo environments. The shaded area is the 10% and 90% percentiles with linear interpolation from the 10 different seeds' runs. Both agents are using $L_2$ loss.

