# OpenReview forum: "Turning Sand to Gold: Recycling Data to Bridge On-Policy and Off-Policy Learning via Causal Bound"
_NeurIPS.cc/2025/Conference — NeurIPS 2025 poster_

### Official Review · Reviewer_KRKD · 2025-06-16

**Clarity:** 4
**Significance:** 3
**Originality:** 4
**Rating:** 5
**Confidence:** 3

**Summary:**

The authors present a novel theoretical result according to which the number of training steps and the experience replay buffer in deep reinforcement learning (DRL) can be significantly reduced.
This is very interesting, given the fact that most of DRL papers present ad hoc heuristics.

**Questions:**

Page 1:
Can you please elaborate a bit on the comparison between on and off policy agents? :)

Page 2:
Please elaborate on the caption of Figure 1.

Page 4:
In Preliminaries, can you please also add some general RL background?

Page 5:
Can you please comment on Assumption 4.1?

Page 6:
Why is $a$ included in the arguments of $\ell$?
Could it be defined on $s$ and $t_{\rm target}$ only?

Page 7:
Can you please further explain the ramifications of Theorem 4.10? :)

Page 8:
The experimental setup looks OK!
Figure 3 needs more comments though, in the caption.

Page 9:
Please expand the Conclusion section and add more future work directions.

**Ethical Concerns:**

["NO or VERY MINOR ethics concerns only"]

**Final Justification:**

The authors have effectively addressed my comments and concerns.

**Limitations:**

None.

**Paper Formatting Concerns:**

None.

**Quality:**

3

**Strengths And Weaknesses:**

Strengths:
The main strength of the paper lies at its theoretical scope, on such an important topic.

In more detail:

The paper introduces a new causal upper bound on the factual loss in reinforcement learning, adapting the Neyman-Rubin potential outcomes framework in a nice way.
This conceptual contribution might open new avenues for bridging on-policy and off-policy methods.

The SUFT OPE term that the authors introduce, leverages computed value network outputs by recycling them in a "replay buffer."
This clever reuse of data leads to improvements in sample complexity and buffer size requirements, while achieving good rewards.

The method the authors introduce is evaluated across a large and diverse set of environments and agents, showing statistically good performance.

Weaknesses:

While the causal bound is theoretically novel, it would be nice if the authors could comment more on its tightness and generalizability.
Moreover, it would be nice to have more comparisons to existing theoretical bounds in OPE :)

Also, please see Questions below.

My Impression:
The paper merits acceptance :)

---

> ### Author Rebuttal · Authors · 2025-07-31
>
> Dear reviewer,
>
> Thank you for taking the time to review our paper.
> Below we address all your questions and suggestions:
>
> $\underline{\text{Bound tightness:}}$
>
> Following the suggestion from reviewer MtKE, we performed the following experiment to test the bound tightness when ignoring the $\delta$ term in optimization.
>
> Firstly, the $\delta$ term involves the true values $y_0, y_1$ (as in Theorem B.12 line 592), which are not observable during training. However, we can approximate these values using the standard Bellman equation for $Q$.
> We conducted an experiment on 20 Atari games using the DQN agent, plotting the values of the standard off-policy loss, the SUFT OPE term, and the approximation for the $\delta$ term over the optimization.
>
> Following this experiment, we observed that in all of the environments, the $\delta$ term remained consistently an order of magnitude **lower** than the off-policy loss, and on the same order of magnitude as the SUFT OPE term.
> This supports the choice of ignoring the $\delta$ term during optimization, and our significant improvements indicate that the bound is tight enough.
> We will conduct this experiment on the entire 57 games benchmark, and add the plots showing the $\delta$ term and the off-policy loss magnitude, along with a short discussion about this term, to the appendix.
>
> Q: Can you please elaborate a bit on the comparison between on and off-policy agents?
>
> A:
>
> Regarding the comparison between on-policy and off-policy agents.
> RL agents interact with the environment using the behavior policy and optimize the target policy.
> On-policy agents maintain an identical behavior and target policy. They interact with the environment, collect experiences, and once they optimize the target policy, they discard all previously collected experiences to maintain the behavior and target policies identical. This results in a more stable learning convergence, but has a major disadvantage of not being sample efficient since they require new experiences for each optimization step.
> On the other hand, off-policy agents are a compromise that allows the agent to act off-policy, meaning it can have a different behavior and target polices. This methodology improves sample efficiency by leveraging past experiences stored in the replay buffer, but it can lead to reduced learning stability due to a mismatch between the behavior and target policies. Our method addresses this divergence and bounds the on-policy loss using the standard off-policy loss, along with an additional OPE term, which is the estimated treatment effect.
>
> Thank you for your suggestion. We will adjust the introduction (line 23) to elaborate more about the comparison between on-policy and off-policy agents to improve clarity.
>
> Q: Please elaborate on the caption of Figure 1:
>
> A:
>
> Regarding Figure 1, it demonstrates our method's reward improvements in comparison to baseline agents that don’t use our method (not including the additional SUFT OPE term in their loss function).
> The blue results indicate that our method outperforms the baseline in those environments, while the grey results show the baseline outperforming our method in those environments. We present the improvement percentage for the higher reward method, as calculated in the improvement percentage formula (line 750).
> We will adjust Figure 1 to include further explanation on this.
>
> Q: In Preliminaries, can you please also add some general RL background?
>
> A:
>
> We appreciate your suggestion to include additional general RL background in the preliminaries, but we prefer not to expand this section and instead use the space for Section 4, as it is now.
>
> Q: Can you please comment on Assumption 4.1?
>
> A:
>
> Regarding assumption 4.1, it is a variation of the inverse triangle inequality, and it is needed for our causal bound proof. We will include this comment in the revised paper.
>
> Q: Why is $a$ included in the arguments of $\ell$? Could it be defined on $s$ and $t_{target}$ only?
>
> A:
>
> In Section 4.2, we present a detailed reduction from the causal framework to the DRL framework using a $Q$-value network for the DQN agent.
> This agent's stored experiences include the states and the performed actions $(s, a)$. The combination of $Q(s, a, \theta_t)$ outputs the estimated outcome for this experience, which is analogous to the definitions in the causal framework, as we are interested in potential outcomes. This is why it is needed in the arguments for $\ell$. For agents that are using a $V$ network, we only use $s$ for $\ell$. For the full reduction between the causal framework and the DRL framework for both $Q$ and $V$ networks, please see Appendix C, Section B.4.
>
> Q: Can you please further explain the ramifications of Theorem 4.10?
>
> A:
>
> Theorem 4.10 is our causal bound, where we bound the factual using the counterfactual loss, an additional term of the estimated treatment effect, and a delta term, which is independent of the network. This bound interpretation in the DRL framework is of bounding the on-policy loss using the off-policy loss, an additional OPE term, and a delta term, which is independent of the $Q$ network.
> The ramification of Theorem 4.10 is the implementation of it into any DRL agent with a $V$ or $Q$-value network by adding the SUFT OPE term to the standard off-policy loss, as shown in Section 4.5.
>
> Q: The experimental setup looks OK! Figure 3 needs more comments, though, in the caption.
>
> A:
>
> We will add additional comments in the caption of Figure 3, further emphasizing that this graph indicates our method's potential to boost agents’ performance even above the human level, while keeping constrained environment interactions and resource demands.
>
> Q: Please expand the Conclusion section and add more future work directions.
>
> A:
>
> We will expand the conclusion section and provide more future work directions, such as the implementation of our method with physical robotics reinforcement learning, where the need for our reduced resource method is crucial.
>
> We would like to thank you for your positive feedback and suggestions. We believe that we have addressed all your questions and suggestions and that our paper is now even better.

---

### Official Review · Reviewer_1ska · 2025-06-23

**Clarity:** 2
**Significance:** 3
**Originality:** 3
**Rating:** 4
**Confidence:** 3

**Summary:**

This paper introduces a causal bound on the factual loss, derived from the potential outcomes framework, to enhance DRL sample efficiency. The method yields substantially higher rewards while drastically reducing the required size of the experience replay buffer at a negligible cost.

**Questions:**

- What is the main difficulty in establishing the theoretical result (Theorem 4.10) for the L2 loss?
- Could you discuss the soundness of the metric used? Specifically, when the baseline performs only marginally better than a random guess, how can the effectiveness of the ratio be guaranteed?

**Ethical Concerns:**

["NO or VERY MINOR ethics concerns only"]

**Final Justification:**

While the empirical results still suggest that the theorem might not hold under the $L_2$ setting, most of my concerns have been mitigated, and I am willing to adjust my score accordingly.

**Limitations:**

yes

**Quality:**

2

**Strengths And Weaknesses:**

**Strengths:**
- Leveraging causal information to enhance sample efficiency in DRL is a significant research direction.
- The experimental results are extensive and validated across various standard benchmarks.
- Theoretical results are provided to ground the proposed approach and verify its effectiveness.

**Weaknesses:**
- The paper is **not well-structured**, which negatively impacts readability. For example, the term "SUFT" should be explained when it first appears, not pages later. Similarly, the explanation for Figure 1 is located far from the figure itself (Figure on pp. 2, explanation on pp. 9). Meanwhile, the introduction lacks detail on how the causal bound is incorporated into the DRL framework. It also fails to provide sufficient insight into why adding the proposed term to the loss function yields such significant improvements over conventional methods.
- Several figures and equations **require formatting adjustments**. For instance, the font size in Figures 4 and 10 is too large, and the equation in line 752 is too long and runs outside the margin. Pages 5 and 6 are overly dense with definitions, many of which could be moved to an appendix. This space in the main body would be better used to detail why the challenge is difficult for previous work to handle and to provide more intuition about the proposed approach.
- **Theory-Experiment Disconnect**. The main theoretical result (Theorem 4.10) relies on Assumption 4.1, which the authors admit does not hold for the L2 loss—the loss used in their main experiments. This undermines the validity of their theoretical justification. Without the assumption, the proposed method reduces to a heuristic regularization technique, and the claim of optimizing a causal bound does not hold. The brief L1-loss results in the appendix do not adequately support the main claims. What's the main difficulty of providing the theoretical result for the L2 loss?
- **Fragile and Potentially Misleading Metric**. The reported “2,427% improvement” is based on a reward ratio formula, i.e., (SUFT - Random) / (Baseline - Random) in line 755, which is highly sensitive *when the baseline performs only slightly better than random*. This can produce arbitrarily large values, exaggerating the method’s effectiveness and obscuring its true impact compared to more standard metrics like absolute or human-normalized scores. The soundness of the used metric should be discussed.

---

> ### Author Rebuttal · Authors · 2025-07-30
>
> Dear reviewer,
>
> Thank you for taking the time to review our paper.
> Below we address all your questions and concerns:
>
> $\underline{L_2 \text{ loss:}}$
>
> The difficulty in establishing the theoretical results for the $L_2$ loss is that in our proof, we rely on Assumption 4.1, which is not satisfied in the $L_2$ case. Having said that, the following analysis shows that in practice, it is satisfied in the vast majority of the optimization process.
>
> Following your review and to provide an intuitive explanation for the significant improvements in our results, we empirically tested the validity of the $L_2$ loss for Assumption 4.1 using synthetic and domain-specific data.
>
> For the synthetic data, we created a short Python script (see the code below) that randomly generates 1 billion samples for $x, y, x', y'$, and checks if the inequality holds for $L_2$ loss.
> The results show that Assumption 4.1 is valid for 82% of randomly sampled values using the $L_2$ loss.
>
> In addition, we tested Assumption 4.1 using the $L_2$ loss for the Atari games domain using the DQN agent.
> Since we are using an off-policy agent, the left side of our causal bound (Theorem 4.10) is not computable, as we cannot sample observations on-policy (this is why we optimize on the upper bound in our method).
> Nevertheless, we return to the part of our proof where we apply Assumption 4.1 (line 595), considering only the parts where the samples are off-policy for both the left and right sides of the inequality in Equations 10 and 11 (using $q_0=1$). Additionally, to approximate the true values of $y_0$ and $y_1$, we use the standard Bellman equation for Q.
>
> With this setup, we ran the 57 Atari games benchmark using 400K timesteps and a batch size of 32 samples, and calculated the elements of the inequality.
> For a sanity check, we use the $L_1$ loss to verify that it holds 100% of the time, which it does indeed.
> The results show that while the assumption may not theoretically hold for $L_2$, it holds in 79% of the cases.
>
> To summarize, although Assumption 4.1 does not hold for $L_2$ loss, in practice, it holds for the vast majority of the optimization process, which explains the performance improvements of our method in the $L_2$ case.
>
> Additionally, you claimed that due to the fact that Assumption 4.1 doesn’t hold for $L_2$, “the proposed method reduces to a heuristic regularization technique, and the claim of optimizing a causal bound does not hold”.
>
> Reviewer MtKE suggested a controlled experiment to isolate the effect of our causal mechanism, which will prove that our method's significant improvements are due to the causal bound optimization, and not just a regularization technique.
> He suggested adding a regularizer to the standard temporal difference loss function in the form of $(Q_{current}-Q_{target})^2$, where $Q_{target}$ is the slowly moving target network, as done in double DQN.
>
> We conducted this experiment on 5 Atari games using the DDQN agent on 10 different random seeds.
> The results are below:
>
> | GAME | DDQN | DDQN with target network regularizer | DDQN using SUFT |
> |-------------|-------|---------------------------|-------------|
> | Assault | 781 | 670 | **1,440** |
> | Asterix | 676 | 666 | **939** |
> | Boxing | -1.73 | -2.41 | **18.4** |
> | Demon Attack | 369 |-17.3 | **2240** |
> | James Bond | 316 | 186 | **416** |
>
> The results above are conclusive and demonstrate that this kind of regularization harms the actual DDQN performance. It proves that our method has a significant impact due to the causal bound optimization, where the old Q-network values used for the OPE term are of the behavior policy used to generate the experiences, therefore mitigating the divergence between the on-policy and off-policy methods. This shows that our method is not just a regularizer that promotes stability, but a causal bound optimization.
> We will conduct this experiment on the entire 57 games benchmark and add the results to the appendix.
>
> $\underline{\text{Metric:}}$
>
> In our paper, we used several evaluation metrics and techniques to ensure the robustness and reliability of our results.
> We conducted controlled experiments with different agents that include our proposed term in comparison to identical baseline agents without it. These controlled experiments were intentionally made in order to isolate and measure the direct impact of our method. To measure our method’s improvement, we compared our agent's reward to the baseline reward using the ratio formula, similar to the one used in [1], which is a highly cited work from a top-tier venue (see Figure 4 in [1]).
>
> When calculating the mean reward ratio formula, we have considered only the valid environments where both the numerator and denominator are greater than zero, ensuring the validity of the ratio formula.
> Following your review, we are willing to adopt your suggestion, which will enhance our method's performance analysis and demonstrate its effectiveness and true impact.
> Therefore, in our revised version of the paper, we will address the areas where the baseline reward is only slightly better than the random reward, specifically when the denominator is lower than 1, which leads to large values. We will adjust the valid environment restriction to consider only environments where both the numerator and denominator are greater than 1, thereby preventing arbitrarily large values.
> The new mean reward ratios for the different agents are:
>
> | Agent | New reward mean ratio | Old reward mean ratio |
> |------|---------------------------|-------------|
> | DQN | 383.19% |  2,427.06% |
> | Vanilla DQN | 228.12% | 230.49% |
> | DDQN | 226.91% | 260.03% |
> | PPO | 197.34% | 203.21% |
>
> Thank you for bringing this to our attention, and your intuition about this metric was correct.
> With our new adjustment to this metric, we are excluding the impact of cases where baseline rewards are only slightly better than the random rewards, and guarantee the effectiveness of this metric.
> The results with the new adjustment still demonstrate that our method continues to have a significant impact on performance while excluding the slightly better than random environments from the mean calculation, thereby strengthening the effectiveness of the evaluation metric and providing a more accurate assessment of our method’s impact.
>
> Furthermore, please note that the original manuscript contains evaluated results using the standard human-normalized score in an ablation study, as shown in Table 5. In the revised version of the paper, we will relocate this analysis from the appendix to the main body of the paper to strengthen our results.
>
> Regarding your modification suggestions, we are willing to accept any suggestions for improvement, and in the revised version of the paper, we will modify the following:
>
> 1.	We will adjust the section in line 40 to add more details to the introduction on how to incorporate the causal bound to the DRL framework, and to provide additional insight into why adding the proposed term to the loss function yields such significant improvements over conventional methods.
> 2.	We will add further explanation to Figure 1 in the caption.
> 3.	We will move footnote number 2 that explains "SUFT" to page 2.
> 4.	We will use a smaller font on Figures 4 and 10.
> 5.	We will adjust the equation on line 752 to fit the margin.
>
> Overall, we would like to thank you for your constructive feedback. We believe that we have addressed all your questions and concerns and that our paper is now even better.
>
> Python code for the synthetic data:
>
> `import numpy as np`
>
> `size = 1_000_000_000`
>
> `x = np.random.normal(loc=0, scale=1, size=size)`
>
> `y = np.random.normal(loc=0, scale=1, size=size)`
>
> `x_tag = np.random.normal(loc=0, scale=1, size=size)`
>
> `y_tag = np.random.normal(loc=0, scale=1, size=size)`
>
> `lhs = (x - y)**2 - (x_tag - y_tag)**2`
>
> `rhs = (x - x_tag)**2 + (y - y_tag)**2`
>
> `false_counter = np.sum(lhs > rhs)`
>
> `print("Success Percentage:", 100 - false_counter / size * 100)`
>
> References:
>
> [1] Kaiser, Ł., Babaeizadeh, M., Miłos, P., Osiński, B., Campbell, R. H., Czechowski, K., ... & Michalewski, H. (2020). MODEL BASED REINFORCEMENT LEARNING FOR ATARI. In 8th International Conference on Learning Representations, ICLR 2020.‏

---

> > ### Comment · Reviewer_1ska · 2025-08-05
> >
> > Thanks for your response. Most of my concerns have been mitigated, and I am willing to adjust my score accordingly.

---

### Official Review · Reviewer_MtKE · 2025-06-26

**Clarity:** 3
**Significance:** 4
**Originality:** 4
**Rating:** 5
**Confidence:** 3

**Summary:**

This paper presents a new algorithm for improving the sample efficiency and performance of DRL agents. The algorithm is based on a theoretical result derived from the causal inference literature. Unlike most work that bounds the counterfactual (off-policy) loss, the authors here propose bounding the unobserved factual (on-policy) loss instead, using off-policy samples. This bound is made of the usual off-policy loss plus a term that measures the treatment effect between the target policy and past policies. In practice, the authors suggest to cleverly recycle the value network outputs, which are normally just thrown away after action selection, and instead store them in the replay buffer. The authors conducts extensive experiments on DQN, PPO, and SAC agents across many Atari and MuJoCo environments, demonstrating clear improvements in reward and convergence speed, and a significant reduction in the required replay buffer size.

**Questions:**

* The bound has the  $\delta$ term which is correctly ignored during optimization, since it doesn't depend on the agent's parameters. However it would be good to add a short discussion on potential looseness of the bound. If  $\delta$ is very large, the bound might be too loose to be useful. Your strong results suggest this isn't a problem, but it would be good to comment on it. In particular, it would be interesting to plot the magnitude of this term over the optimization.

* Coming back to my last point in the "Weaknesses": the SUFT loss term, $(V_{behavior} − V_{target} )^2$ , is functionally very similar to known regularization techniques that performs some form of value function smoothing (e.g., in PPO v2). To isolate the effect of your proposed causal mechanism, would it make sense adding a control experiment? For example, it would be interesting to check whether the benefit comes from simply adding inertia to the value function, preventing it from changing too rapidly, rather than tracking a specific behavioral policy's value. One could e.g. add a term ot the TD loss, $L_{total} = L_{TD} + \lambda_{inertia} \cdot (Q_{current} - Q_{target})^2$ where $Q_{target}$ is the slowly moving target Q network used to compute the target value in the TD loss, as is usually done in DQN.

**Ethical Concerns:**

["NO or VERY MINOR ethics concerns only"]

**Final Justification:**

The performance gain is impressive, and the method simple. The authors addressed my concerns and the proposed revision is satisfactory.

**Limitations:**

yes

**Paper Formatting Concerns:**

no concern

**Quality:**

4

**Strengths And Weaknesses:**

# Strengths

* The paper tackles the crucial problem of sample efficiency in DRL. The method is surprisingly simple, can be used orthogonally with many other algorithms, and its empirical results are exceptionally strong across many different agents and benchmarks. If these results hold up, the method's ability to improve performance and cut down on resource needs would be a major step for the field.

* I also found the main theoretical idea of looking at the problem as a causal bound on the factual loss, instead of the counterfactual loss, to be really original.

* The experimental validation is high quality and very thorough. Using 10 random seeds, all 57 Atari games and 5 MuJoCo environments, with several different DRL agents (DQN, PPO, SAC) really convincingly show the robustness of the advantage of the algorithm.

# Weaknesses

* I really think the paper needs some pseudocode for how to apply the bound to DQN and PPO. Currently, the reader has to guess some of the key implementation details. For instance, it is not clear to me how the existing PPO critique loss would map to the proposed upper bound.

* There's a bit of a gap between the formal theory and the experiments. The proof for the causal bound needs an L1 loss, while the main experiments with the most impressive results are all using an L2 loss. The authors acknowledge this, but this raises the question of why the method is so effective with L2 loss, e.g. it is possible that the additional term act as a regularizer that promotes stability, which benefits the optimization independently of the bound itself

---

> ### Author Rebuttal · Authors · 2025-07-30
>
> Dear reviewer,
>
> Thank you for taking the time to review our paper.
> Below we address all your questions and suggestions:
>
> Following your review, you suggested performing a controlled experiment to isolate the effect of our causal mechanism, which will prove that our method's significant improvements are due to the causal bound optimization, and not just a regularization technique.
>
> As you suggested, we conducted such an experiment and added a regularizer to the standard temporal difference loss function in the form of $(Q_{current}-Q_{target})^2$, where $Q_{target}$ is the slowly moving target network, as done in double DQN.
> We conducted this experiment on 5 Atari games using the DDQN agent on 10 different random seeds. The results are below:
>
> | GAME | DDQN | DDQN with target network regularizer | DDQN using SUFT |
> |-------------|-------|---------------------------|-------------|
> | Assault | 781 | 670 | **1,440** |
> | Asterix | 676 | 666 | **939** |
> | Boxing | -1.73 | -2.41 | **18.4** |
> | Demon Attack | 369 |-17.3 | **2240** |
> | James Bond | 316 | 186 | **416** |
>
> The results above are conclusive and demonstrate that this kind of regularization harms the actual DDQN performance. It proves that our method has a significant impact due to the causal bound optimization, where the old Q-network values used for the OPE term are of the behavior policy used to generate the experiences, therefore mitigating the divergence between the on-policy and off-policy methods.
> This shows that our method is not just a regularizer that promotes stability, but a causal bound optimization. We will conduct this experiment on the entire 57 games benchmark and add the results to the appendix.
>
> Having said that we also performed an analysis that explains the success using the $L_2$ loss.
> As mentioned in the paper, the theoretical result relies on Assumption 4.1, which doesn't hold for the $L_2$ loss.
> We used the $L_2$  loss in our experiments to compare our method with standard DRL agents.
>
> While the $L_2$ loss does not theoretically hold this assumption, in order to provide an intuitive explanation for the significant improvements in our results, we empirically tested the validity of the $L_2$ loss for Assumption 4.1 using synthetic and domain-specific data.
> For the synthetic data, we created a short Python script (see the code below) that randomly generates 1 billion samples for $x, y, x', y'$, and checks if the inequality holds for $L_2$ loss.
> The results show that Assumption 4.1 is valid for 82% of randomly sampled values using the $L_2$ loss.
>
> In addition, we tested Assumption 4.1 using the $L_2$ loss for the Atari games domain using the DQN agent.
> Since we are using an off-policy agent, the left side of our causal bound (Theorem 4.10) is not computable, as we cannot sample observations on-policy (this is why we optimize on the upper bound in our method).
> Nevertheless, we return to the part of our proof where we apply Assumption 4.1 (line 595), considering only the parts where the samples are off-policy for both the left and right sides of the inequality in Equations 10 and 11 (using $q_0=1$). Additionally, to approximate the true values of $y_0$ and $y_1$, we use the standard Bellman equation for Q.
>
> With this setup, we ran the 57 Atari games benchmark using 400K timesteps and a batch size of 32 samples, and calculated the elements of the inequality.
> For a sanity check, we use the $L_1$ loss to verify that it holds 100% of the time, which it does indeed.
> The results show that while the assumption may not theoretically hold for $L_2$, it holds in 79% of the cases.
>
> To summarize, although Assumption 4.1 does not hold for $L_2$ loss, in practice, it holds for the vast majority of the optimization process, which explains the performance improvements of our method in the $L_2$ case.
>
> $\underline{\delta \text{ term:}}$
>
> Thank you for your suggestion to include a discussion on the potential looseness of the bound, specifically regarding the $\delta$ term, which we ignore during optimization since it is independent of $Q$. You also proposed plotting the magnitude of this term over the optimization.
>
> The $\delta$ term involves the true values $y_0, y_1$ (as in Theorem B.12 line 592), which are not observable during training. However, we can approximate these values using the standard Bellman equation for $Q$.
> We conducted an experiment on 20 Atari games using the DQN agent, plotting the values of the standard off-policy loss, the SUFT OPE term, and the approximation for the $\delta$ term over the optimization.
>
> Following this experiment, we observed that in all of the environments, the $\delta$ term remained consistently an order of magnitude **lower** than the off-policy loss, and on the same order of magnitude as the SUFT OPE term.
> This supports the choice of ignoring the $\delta$ term during optimization, as it does not made the bound too loosen to be useful.
> We will conduct this experiment on the entire 57 games benchmark, and add the plots showing the $\delta$ term and the off-policy loss magnitude, along with a short discussion about this term, to the appendix.
>
> In addition, we accept your suggestion to add pseudocode for how to apply the causal bound in both DQN and PPO, and we will add this to the appendix.
>
> Regarding the integration of the SUFT term into PPO’s critic loss, although PPO is an on-policy algorithm, it performs multiple epochs of optimization over the same buffer of data, leading to a divergence between the behavior and target policies. This divergence is the reason PPO benefits from our method.
>
> In the PPO paper [1], equation 9 defines the overall PPO loss, where the critic component is expressed as: $L_t^{VF}(\theta)=(V_{\theta}(s_t)-V_t^{targ})^2$.
> Our additional SUFT term is implemented to the critic loss in the form of $L_t^{VF}(\theta)=(V_{\theta}(s_t)-V_t^{targ})^2 + \lambda_{TF}(V_{\theta}(s_t)-V_{\theta_{old}}(s_t))^2$.
> Where $V_{\theta}(s_t)$ is the value outputs of the target policy, and $V_{\theta_{old}}(s_t)$ is the value outputs of the behavior policy used to generate the experiences.
>
> Notably, during the first epoch, before optimizing the network, the behavior and target policies are identical, meaning that our additional term will be 0.
> After the first optimization step, the target policy network has been optimized, and now it differs from the behavior policy, meaning that our additional OPE term now quantifies the estimated treatment effect between the target and behavior policies.
>
> In addition, the $V_{\theta_{old}}(s_t)$ is already calculated during the interaction with the environment and the calculation of the $V_t^{targ}$ for each sample collection. This means that our method reuses the old network outputs and stores them along with the sample collection to use them in the calculation of the SUFT OPE term in a similar manner to the DQN implementation.
>
> We would like to thank you for your positive feedback and suggestions. We believe that we have addressed all your questions and suggestions and that our paper is now even better.
>
>
> Python code for the synthetic data:
>
> `import numpy as np`
>
> `size = 1_000_000_000`
>
> `x = np.random.normal(loc=0, scale=1, size=size)`
>
> `y = np.random.normal(loc=0, scale=1, size=size)`
>
> `x_tag = np.random.normal(loc=0, scale=1, size=size)`
>
> `y_tag = np.random.normal(loc=0, scale=1, size=size)`
>
> `lhs = (x - y)**2 - (x_tag - y_tag)**2`
>
> `rhs = (x - x_tag)**2 + (y - y_tag)**2`
>
> `false_counter = np.sum(lhs > rhs)`
>
> `print("Success Percentage:", 100 - false_counter / size * 100)`
>
> References:
>
> [1] Schulman, J., Wolski, F., Dhariwal, P., Radford, A., and Klimov, O. Proximal policy optimization algorithms. 2017

---

### Official Review · Reviewer_zGCa · 2025-07-01

**Clarity:** 3
**Significance:** 3
**Originality:** 3
**Rating:** 4
**Confidence:** 2

**Summary:**

This paper proposes a causal approach to an upper bound on-policy loss in RL with off-policy data. The resulting upper bound can be integrated into any on-policy RL algorithm. The authors use empirical evaluations to show that the resulting SUFT version of RL algorithms can achieve better performance with low sample/computational overhead.

**Questions:**

Since Q-learning is inherently an off-policy method, the arguments presented in Section 4.1 are unclear to me. In particular, it is surprising that the SUFT variant leads to such a substantial performance improvement. Could the authors clarify this discrepancy and provide further explanation?

The theoretical analysis relies on a form of the triangle inequality for the loss function. While the $L_1$ loss satisfies this assumption, the more commonly used $L_2$ loss does not. Nevertheless, SUFT appears to perform even better under the $L_2$ loss in the empirical results, which seems inconsistent with the theoretical framework. I encourage the authors to provide intuitive explanations for this observation.

**Ethical Concerns:**

["NO or VERY MINOR ethics concerns only"]

**Final Justification:**

After reviewing the other reviewers’ comments and the authors’ rebuttal, I find that my two primary concerns—evaluation metrics and the inconsistency between theory and empirical findings—have been satisfactorily addressed. I therefore maintain my positive assessment of this paper.

**Limitations:**

The authors do not explicitly state the limitations of their work.

**Quality:**

3

**Strengths And Weaknesses:**

Strength:
* The paper is overall well-written and easy to follow.
* The paper is well-motivated, addressing the long-standing challenge in reinforcement learning of using inexpensive off-policy data in place of costly on-policy data.
* The authors introduce tools from causal inference to RL. The proposed methodology looks natural and intuitive.
* The empirical evaluations look sound.

Weaknesses:
* The empirical evaluation introduces a metric termed the reward improvement ratio. This metric appears unconventional and may potentially overstate the improvements achieved by the proposed method.
* I think the author should try more recent baselines on Atari games and see whether the proposed method works for them. For example the baselines in https://arxiv.org/abs/2112.04145.

Since I am not very familiar with the literature on causal inference, I choose to assign a low confidence score to my evaluation.

---

> ### Author Rebuttal · Authors · 2025-07-30
>
> Dear reviewer,
>
> Thank you for taking the time to review our paper.
> Below we address all your questions and concerns:
>
> Firstly, we want to address a misunderstanding in your review.
> In the summary, you wrote that you understand that our method is applicable to any **on-policy** RL algorithm. Then, in your first question, you asked how our method leads to a substantial performance improvement in **off-policy** Q-learning algorithms.
> In our paper, we presented a method to enhance any DRL **off-policy** agent with a V or a Q-value network.
>
> Since the off-policy method is a compromise of allowing the agent to act off-policy to improve sample efficiency by leveraging past experiences stored in the replay buffer, it can lead to reduced learning stability due to a mismatch between the behavior and target policies.
> Our method addresses this divergence and bounds the on-policy loss using the standard off-policy loss, along with an additional OPE term, which is the estimated treatment effect.
> Therefore, our method demonstrates significant improvements for off-policy agents, as it bridges the gap between on-policy and off-policy methods using our causal bound.
> Note that our method is also applicable to PPO, an on-policy agent, which benefits from our method due to the divergence between the behavior and target policies during its epoch optimization.
>
>
> $\underline{L_2 \text{ loss:}}$
>
> As mentioned in the paper, the theoretical result relies on Assumption 4.1, which doesn't hold for the $L_2$ loss. We used the $L_2$ loss in our experiments to compare our method with standard DRL agents.
>
> While the $L_2$ loss does not theoretically hold this assumption, following your review and to provide an intuitive explanation for the significant improvements in our results, we empirically tested the validity of the $L_2$ loss for Assumption 4.1 using synthetic and domain-specific data.
>
> For the synthetic data, we created a short Python script (see the code below) that randomly generates 1 billion samples for $x, y, x', y'$, and checks if the inequality holds for $L_2$ loss.
> The results show that Assumption 4.1 is valid for 82% of randomly sampled values using the $L_2$ loss.
>
> In addition, we tested Assumption 4.1 using the $L_2$ loss for the Atari games domain using the DQN agent.
> Since we are using an off-policy agent, the left side of our causal bound (Theorem 4.10) is not computable, as we cannot sample observations on-policy (this is why we optimize on the upper bound in our method).
> Nevertheless, we return to the part of our proof where we apply Assumption 4.1 (line 595), considering only the parts where the samples are off-policy for both the left and right sides of the inequality in Equations 10 and 11 (using $q_0=1$). Additionally, to approximate the true values of $y_0$ and $y_1$, we use the standard Bellman equation for Q.
>
> With this setup, we ran the 57 Atari games benchmark using 400K timesteps and a batch size of 32 samples, and calculated the elements of the inequality.
> For a sanity check, we use the $L_1$ loss to verify that it holds 100% of the time, which it does indeed.
> The results show that while the assumption may not theoretically hold for $L_2$, it holds in 79% of the cases.
>
> To summarize, although Assumption 4.1 does not hold for $L_2$ loss, in practice, it holds for the vast majority of the optimization process, which explains the performance improvements of our method in the $L_2$ case.
>
> Additionally, reviewer MtKE suggested a controlled experiment to isolate the effect of our causal mechanism, which will prove that our method's significant improvements are due to the causal bound optimization, and not just a regularization technique.
> He suggested adding a regularizer to the standard temporal difference loss function in the form of $(Q_{current}-Q_{target})^2$, where $Q_{target}$ is the slowly moving target network, as done in double DQN.
>
> We conducted this experiment on 5 Atari games using the DDQN agent on 10 different random seeds.
> The results are below:
>
> | GAME | DDQN | DDQN with target network regularizer | DDQN using SUFT |
> |-------------|-------|---------------------------|-------------|
> | Assault | 781 | 670 | **1,440** |
> | Asterix | 676 | 666 | **939** |
> | Boxing | -1.73 | -2.41 | **18.4** |
> | Demon Attack | 369 |-17.3 | **2240** |
> | James Bond | 316 | 186 | **416** |
>
> The results above are conclusive and demonstrate that this kind of regularization harms the actual DDQN performance. It proves that our method has a significant impact due to the causal bound optimization, where the old Q-network values used for the OPE term are of the behavior policy used to generate the experiences, therefore mitigating the divergence between the on-policy and off-policy methods. This shows that our method is not just a regularizer that promotes stability, but a causal bound optimization.
> We will conduct this experiment on the entire 57 games benchmark and add the results to the appendix.
>
> $\underline{\text{Metric:}}$
>
> In our paper, we used several evaluation metrics and techniques to ensure the robustness and reliability of our results.
> We conducted controlled experiments with different agents that include our proposed term in comparison to identical baseline agents without it. These controlled experiments were intentionally made in order to isolate and measure the direct impact of our method. To measure our method’s improvement, we compared our agent's reward to the baseline reward using the ratio formula, similar to the one used in [1], which is a highly cited work from a top-tier venue (see Figure 4 in [1]).
>
> When calculating the mean reward ratio formula, we have considered only the valid environments where both the numerator and denominator are greater than zero, ensuring the validity of the ratio formula.
> Following your review, we will address the areas where the denominator is lower than 1, which leads to large values. We will adjust the valid environment restriction to consider only environments where both the numerator and denominator are greater than 1, thereby preventing arbitrarily large values.
> The new mean reward ratios for the different agents are:
>
> | Agent | New reward mean ratio | Old reward mean ratio |
> |------|---------------------------|-------------|
> | DQN | 383.19% |  2,427.06% |
> | Vanilla DQN | 228.12% | 230.49% |
> | DDQN | 226.91% | 260.03% |
> | PPO | 197.34% | 203.21% |
>
> Thank you for bringing this to our attention, and your intuition about this metric was correct.
> With our new adjustment to this metric, we are excluding the impact of cases where baseline rewards are only slightly better than the random rewards, and guarantee the effectiveness of this metric.
> The results with the new adjustment still demonstrate that our method continues to have a significant impact on performance while excluding the slightly better than random environments from the mean calculation, thereby strengthening the effectiveness of the evaluation metric and providing a more accurate assessment of our method’s impact.
>
> Furthermore, please note that the original manuscript contains evaluated results using the standard human-normalized score in an ablation study, as shown in Table 5. In the revised version of the paper, we will relocate this analysis from the appendix to the main body of the paper to strengthen our results.
>
> Overall, we would like to thank you for your feedback. We believe that we have addressed all your questions and concerns and that our paper is now even better.
>
> Python code for the synthetic data:
>
> `import numpy as np`
>
> `size = 1_000_000_000`
>
> `x = np.random.normal(loc=0, scale=1, size=size)`
>
> `y = np.random.normal(loc=0, scale=1, size=size)`
>
> `x_tag = np.random.normal(loc=0, scale=1, size=size)`
>
> `y_tag = np.random.normal(loc=0, scale=1, size=size)`
>
> `lhs = (x - y)**2 - (x_tag - y_tag)**2`
>
> `rhs = (x - x_tag)**2 + (y - y_tag)**2`
>
> `false_counter = np.sum(lhs > rhs)`
>
> `print("Success Percentage:", 100 - false_counter / size * 100)`
>
> References:
>
> [1] Kaiser, Ł., Babaeizadeh, M., Miłos, P., Osiński, B., Campbell, R. H., Czechowski, K., ... & Michalewski, H. (2020). MODEL BASED REINFORCEMENT LEARNING FOR ATARI. In 8th International Conference on Learning Representations, ICLR 2020.‏

---

### Note · Authors · 2025-08-13

We thank the reviewers for their insightful and constructive feedback, as well as for recognizing the novelty and significant impact of our method.
In our rebuttal, as mentioned by the reviewers, we have mitigated all raised concerns.
Specifically, we conducted additional experiments to provide an intuitive and empirical explanation for the L2 improvements. We have also conducted experiments that prove that our method has a significant impact due to the causal bound optimization, and that it is not just a regularizer that promotes stability.
In addition, we mitigated the concerns regarding the used evaluation metric by excluding the impact of cases where baseline rewards are only slightly better than the random rewards, and guarantee the effectiveness of this metric.
We also provided expanded explanations and clarified technical details.
Overall, we thank the reviewers again, and we believe that our paper is now even better.

---

### Decision · Program_Chairs · 2025-09-17

**Decision:**

Accept (poster)

**Comment:**

The paper proposes a new objective for off-policy RL based on a novel causal bound on the factual (on-policy) loss, which can be computed using past value network outputs. (As these outputs are usually discarded, the approach turns "sand into gold.") The bound consists of an off-policy component and a term that measures the treatment effect between the current policy and past policies. It's a clever idea, and it appears to actually work too. The paper reports improvements up to 2400% over baselines.

All of the reviews said that the paper was novel and technically sound. They also highlighted two main weaknesses:
1. There is a slight gap between the theory (Assumption 1 says the loss must obey the triangle inequality) and the application (the experiments use $L_2$ loss, which doesn't obey the triangle inequality).
2. The "reward improvement" metric is non-standard, and sensitive to the performance of the baseline.
The authors' rebuttal appears to have mitigated these concerns. They conducted additional analysis of the $L_2$ loss, and updated their reward improvement metric to make it less sensitive. This satisfied the reviewers. I therefore recommend accepting the paper.